# Optimal approximation using complex-valued neural networks

**Paul Geuchen**
MIDS,
KU Eichstätt-Ingolstadt,
Auf der Schanz 49,
85049 Ingolstadt, Germany
`paul.geuchen@ku.de`

**Felix Voigtlaender**
MIDS,
KU Eichstätt-Ingolstadt,
Auf der Schanz 49,
85049 Ingolstadt, Germany
`felix.voigtlaender@ku.de`

## Abstract

Complex-valued neural networks (CVNNs) have recently shown promising empirical success, for instance for increasing the stability of recurrent neural networks and for improving the performance in tasks with complex-valued inputs, such as in MRI fingerprinting. While the overwhelming success of Deep Learning in the real-valued case is supported by a growing mathematical foundation, such a foundation is still largely lacking in the complex-valued case. We thus analyze the expressivity of CVNNs by studying their approximation properties. Our results yield the first quantitative approximation bounds for CVNNs that apply to a wide class of activation functions including the popular modReLU and complex cardioid activation functions. Precisely, our results apply to any activation function that is smooth but not polyharmonic on some non-empty open set; this is the natural generalization of the class of smooth and non-polynomial activation functions to the complex setting. Our main result shows that the error for the approximation of $C^k$-functions scales as $m^{-k/(2n)}$ for $m \to \infty$ where $m$ is the number of neurons, $k$ the smoothness of the target function and $n$ is the (complex) input dimension. Under a natural continuity assumption, we show that this rate is optimal; we further discuss the optimality when dropping this assumption. Moreover, we prove that the problem of approximating $C^k$-functions using continuous approximation methods unavoidably suffers from the curse of dimensionality.

## 1 Introduction

Deep Learning currently predominantly relies on real-valued neural networks, which have led to breakthroughs in fields like image classification or speech recognition [23, 27, 43]. However, recent work has uncovered several application areas in which the use of complex-valued neural networks (CVNNs) leads to better results than the use of their real-valued counterparts. These application areas mainly include tasks where complex numbers inherently occur as inputs of a machine learning model such as Magnetic Resonance Imaging (MRI) [16, 38, 44] and Polarimetric Synthetic Aperture Radar (PolSAR) Imaging [8, 9, 48]. Moreover, CVNNs have been used to improve the stability of recurrent neural networks [5] and have been successfully applied in various other fields [32, 37]. The mathematical theory of these complex-valued neural networks, however, is still in its infancy. There is therefore a great interest in studying CVNNs and in particular in uncovering the differences and commonalities between CVNNs and their real-valued counterparts. A prominent example highlighting the unexpected differences between both network classes is the *universal approximation theorem* for neural networks, whose most general real-valued version was proven in 1993 [28] (with a more restricted version appearing earlier [17]) and which was recently generalized to the case of CVNNs [45]. The two theorems describe necessary and sufficient conditions on

37th Conference on Neural Information Processing Systems (NeurIPS 2023).

| | Condition on activation function | Continuity of weight selection | Approximation Error |
|---|---|---|---|
| Theorem 3.2 | smooth & non-polyharmonic | possible | $\mathcal{O}(m^{-k/(2n)})$ |
| Consequence of Theorem 4.1 | continuous | assumed | $\Omega(m^{-k/(2n)})$ |
| Theorem 4.2 | very special activation function | not assumed | $\mathcal{O}(m^{-k/(2n-1)})$ |
| Theorem 4.3 | $\dfrac{1}{1+e^{-\mathrm{Re}(z)}}$ | not assumed | $\widetilde{\Omega}(m^{-k/(2n)})$ |

Table 1: Overview of the proven approximation bounds. $k$ is the regularity of the approximated functions (which are assumed to be $C^k$), $n$ the (complex) input dimension and $m$ the number of neurons in the hidden layer of the network. The notation $\widetilde{\Omega}$ is similar to $\Omega$, but ignoring log factors.

an activation function which guarantee that arbitrarily wide neural networks of a fixed depth can approximate any continuous function on any compact set arbitrarily well (with respect to the uniform norm). Already here it was shown that complex-valued networks behave significantly different from real-valued networks: While real-valued networks are universal if and only if the activation function is non-polynomial, complex-valued networks with a single hidden layer are universal if and only if the activation function is non-polyharmonic (see below for a definition). Furthermore, there exist continuous activation functions for which deep CVNNs are universal but shallow CVNNs are not, whereas the same cannot happen for real-valued neural networks. This example shows that it is highly relevant to study the properties of CVNNs and to examine which of the fundamental properties of real-valued networks extend to complex-valued networks.

Essentially the only existing *quantitative* result regarding the approximation-theoretical properties of CVNNs is [14], which provides results for approximating $C^k$-functions by *deep* CVNNs using the modReLU activation function. However, for real-valued NNs it is known that already *shallow* NNs can approximate $C^k$-functions at the optimal rate. Precisely, Mhaskar showed in [33] that one can approximate $C^k$-functions on $[-1,1]^n$ with an error of order $m^{-k/n}$ as $m \to \infty$, where $m$ is the number of neurons in the hidden layer. Here he assumed that the activation function is smooth on an open interval and that at some point of this interval no derivative vanishes. This is equivalent to the activation function being smooth and non-polynomial on that interval, cf. [20, p. 53].

The present paper shows that a comparable result holds in the setting of complex-valued networks, by proving that one can approximate every function in $C^k(\Omega_n; \mathbb{C})$ (where differentiability is understood in the sense of real variables) with an error of the order $m^{-k/(2n)}$ (as $m \to \infty$) using shallow complex-valued neural networks with $m$ neurons in the hidden layer. Here we define the cube $\Omega_n := [-1,1]^n + i[-1,1]^n$. The result holds whenever the activation function $\phi : \mathbb{C} \to \mathbb{C}$ is smooth and non-polyharmonic on some non-empty open set. This is a very natural condition, since for polyharmonic activation functions there exist $C^k$-functions that cannot be approximated at all below some error threshold using shallow neural networks with this activation function [45].

Furthermore, the present paper studies in how far the approximation order of $m^{-k/(2n)}$ is *optimal*, meaning that an order of $m^{-(k/2n)-\alpha}$ (where $\alpha > 0$) cannot be achieved. Here it turns out that the derived order of approximation is indeed optimal (even in the class of CVNNs with possibly more than one hidden layer) in the setting that the weight selection is *continuous*, meaning that the map that assigns to a function $f \in C^k(\Omega_n; \mathbb{C})$ the weights of the approximating network is continuous with respect to some norm on $C^k(\Omega_n; \mathbb{C})$. This continuity assumption is natural since typical learning algorithms such as (stochastic) gradient descent use samples $f(x_j)$ of the target function and then apply continuous operations to them to update the network weights.

We investigate this optimality result further by dropping the continuity assumption and constructing two special smooth and non-polyharmonic activation functions with the first one having the property that the order of approximation can indeed be strictly improved via a *discontinuous* selection of

the related weights. For the second activation function we show that the order of $m^{-k/(2n)}$ *cannot* be improved, even if one allows a discontinuous weight selection. This in particular shows that in the given generality of arbitrary smooth, non-polyharmonic activation functions, the upper bound $\mathcal{O}\left(m^{-k/(2n)}\right)$ cannot be improved, even for a possibly discontinuous choice of the weights. An overview of the approximation bounds proven in this paper can be found in Table 1.

Moreover, we analyze the *tractability* (in terms of the input dimension $n$) of the considered problem of approximating $C^k$-functions using neural networks. Theorem 5.1 shows that one necessarily needs a number of parameters *exponential* in $n$ to obtain a non-trivial approximation error. To the best of our knowledge, Theorem 5.1 is the first result showing that the problem of approximating $C^k$-functions using continuous approximation methods is intractable (in terms of the input dimension $n$).

## 1.1 Related Work

**Real-valued neural networks.** By now, the approximation properties of real-valued neural networks are quite well-studied (cf. [10, 28, 33, 40, 41, 46, 47] and the references therein). We here only discuss a few papers that are most closely related to the present work.

In [33], Mhaskar analyzes the rate of approximation of shallow real-valued neural networks for target functions of regularity $C^k$. Our results can be seen as the generalization of [33] to the complex setting. Our proofs rely on several techniques from [33]; however, significant modifications are required to make the proofs work for general smooth non-polyharmonic functions.

One of the first papers to observe that neural networks with general (smooth) activation function can be surprisingly expressive is [31] where it was shown that a neural network of *constant size* can be universal. One of the activation functions in Section 4 is based on a similar idea.

The importance of distinguishing between continuous and discontinuous weight selection (which in our setting is discussed in Section 4) was observed for ReLU-networks in [47].

The paper [25] shows that neural network approximation is *not* continuous in the following sense: The best approximating neural network $\Phi(f)$ of a given size does not depend continuously on $f \in C^k$. This result, however, is not in conflict with our results: We want to assign to any $C^k$-function $f$ a network $\widetilde{\Phi}(f)$ that approximates $f$ below the error threshold $m^{-k/(2n)}$. The network $\widetilde{\Phi}(f)$, however, does *not* have to coincide with the best approximating network $\Phi(f)$.

**Complex-valued neural networks.** When it comes to general literature about mathematical properties of complex-valued neural networks, surprisingly little work can be found. The *Universal Approximation Theorem for Complex-Valued Neural Networks* [45] has already been mentioned above. In particular, it has been shown that shallow CVNNs are universal if and only if the activation function $\phi$ is not polyharmonic. Thus, the condition assumed in the present paper (that $\phi$ should be smooth, but not polyharmonic) is quite natural.

Regarding *quantitative* approximation results for CVNNs, the only existing work of which we are aware is [14], analyzing the approximation capabilities of *deep* CVNNs where the modReLU is used as activation function. Since the modReLU satisfies our condition regarding the activation function, the present work can be seen as an improvement to [14]. Precisely, (i) we consider general activation functions, including but not limited to the modReLU, (ii) we improve the order of approximation by a log factor, and (iii) we show that this order of approximation can be achieved using shallow networks instead of the deep networks used in [14]. We stress that our proof techniques differ significantly from the ones applied in [14]: The arguments in [14] take their main ideas from [46] making heavy use of the specific definition of the modReLU. In contrast, since we consider quite general activation functions, we necessarily follow a much more general approach following the ideas from [33].

## 2 Preliminaries

**Shallow complex-valued neural networks.** In this paper we mainly consider so-called shallow complex-valued neural networks, meaning complex-valued neural networks with a single hidden layer. Precisely, we consider functions of the form

$$\mathbb{C}^n \ni z \mapsto \sum_{j=1}^{m} \sigma_j \phi \left( \rho_j^T \cdot z + \eta_j \right) \in \mathbb{C},$$

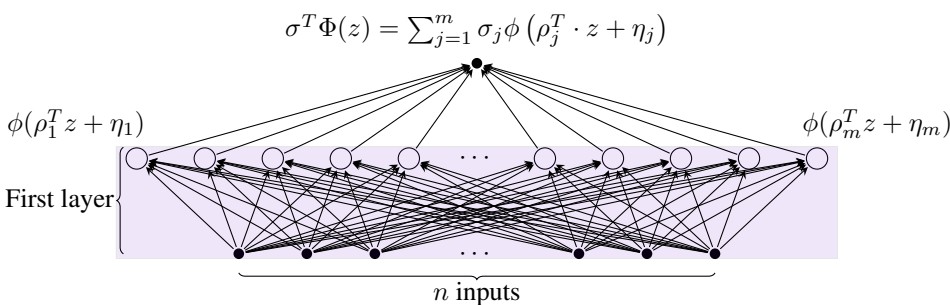

$$\sigma^T \Phi(z) = \sum_{j=1}^m \sigma_j \phi\left(\rho_j^T \cdot z + \eta_j\right)$$

$\phi(\rho_1^T z + \eta_1)$ $\phi(\rho_m^T z + \eta_m)$

First layer

$\cdots$

$n$ inputs

Figure 1: Graphical representation of a shallow neural network. Input and output neurons are depicted as dots, hidden neurons are depicted as circles. The term *first layer* describes the transformation from the input to the hidden neurons, including the application of the activation function.

with $\rho_1, ..., \rho_m \in \mathbb{C}^n$, $\sigma_1, ..., \sigma_m, \eta_1, ..., \eta_m \in \mathbb{C}$ and an *activation function* $\phi : \mathbb{C} \to \mathbb{C}$. Here, $m \in \mathbb{N}$ denotes the number of neurons of the network and we write $v^T$ for the transpose of a vector $v$.

To simplify the formulation of the results, we introduce the following notation: We write $\mathcal{F}_{n,m}^\phi$ for the set of *first layers* of shallow complex-valued neural networks with activation function $\phi$, with $n$ input neurons and $m$ hidden neurons, meaning

$$\mathcal{F}_{n,m}^\phi := \left\{ z \mapsto \left(\phi\left(\rho_j^T \cdot z + \eta_j\right)\right)_{j=1}^m : \rho_j \in \mathbb{C}^n, \ \eta_j \in \mathbb{C} \right\} \subseteq \{F : \mathbb{C}^n \to \mathbb{C}^m\}.$$

Hence, each shallow CVNN can be written as $\sigma^T \Phi$ with $\sigma \in \mathbb{C}^m$ and $\Phi \in \mathcal{F}_{n,m}^\phi$; see Figure 1 for a graphical representation of a shallow CVNN.

**Approximation.** The paper aims to analyze the approximation of $C^k$-functions on the complex cube

$$\Omega_n := [-1, 1]^n + i[-1, 1]^n$$

using shallow CVNNs. Here, we say that a function $f : \Omega_n \to \mathbb{C}$ is in $C^k(\Omega_n; \mathbb{C})$ if and only if $f$ is $k$ times continuously differentiable on $\Omega_n$, where the derivative is to be understood in the sense of real variables, i.e., in the sense of interpreting $f$ as a function $[-1, 1]^{2n} \to \mathbb{R}^2$ and taking usual real derivatives. We further define the $C^k$-*norm* of a function $f \in C^k(\Omega_n; \mathbb{C})$ as

$$\|f\|_{C^k(\Omega_n;\mathbb{C})} := \sup_{\substack{\mathbf{k} \in \mathbb{N}_0^{2n} \\ |\mathbf{k}| \leq k}} \|\partial^{\mathbf{k}} f\|_{L^\infty(\Omega_n;\mathbb{C})}, \qquad \text{where} \qquad \|g\|_{L^\infty(\Omega_n;\mathbb{C})} := \sup_{z \in \Omega_n} |g(z)| \qquad (2.1)$$

for any function $g : \Omega_n \to \mathbb{C}$. Note that we write $\mathbb{N} = \{1, 2, 3, ...\}$ and $\mathbb{N}_0 = \{0\} \cup \mathbb{N}$. Using the previously introduced notation, we thus seek to bound the worst-case approximation error, i.e.,

$$\sup_{\substack{f \in C^k(\Omega_n;\mathbb{C}) \\ \|f\|_{C^k(\Omega_n;\mathbb{C})} \leq 1}} \inf_{\substack{\Phi \in \mathcal{F}_{n,m}^\phi \\ \sigma \in \mathbb{C}^m}} \|f - \sigma^T \Phi\|_{L^\infty(\Omega_n;\mathbb{C})}.$$

**Wirtinger calculus and polyharmonic functions.** For a function $f : \mathbb{C} \to \mathbb{C}$ which is differentiable in the sense of real variables at a point $z_0 \in \mathbb{C}$ we define its *Wirtinger derivatives* at $z_0$ as

$$\partial_{\text{wirt}} f(z_0) := \frac{1}{2}\left(\frac{\partial f}{\partial x}(z_0) - i \cdot \frac{\partial f}{\partial y}(z_0)\right) \quad \text{and} \quad \overline{\partial}_{\text{wirt}} f(z_0) := \frac{1}{2}\left(\frac{\partial f}{\partial x}(z_0) + i \cdot \frac{\partial f}{\partial y}(z_0)\right).$$

Here, $\frac{\partial}{\partial x}$ and $\frac{\partial}{\partial y}$ denote the usual partial derivatives in the sense of real variables. We extend this definition to multivariate functions defined on open subsets of $\mathbb{C}^n$ by considering *coordinatewise* Wirtinger derivatives.

A function $f : U \to \mathbb{C}$, where $U \subseteq \mathbb{C}$ is an open set, is called *smooth* if it is differentiable arbitrarily many times (in the sense of real variables). We write $f \in C^\infty(U; \mathbb{C})$ in that case. Moreover, $f$ is called *polyharmonic* (on $U$) if it is smooth and if there exists $m \in \mathbb{N}_0$ satisfying

$$\Delta^m f \equiv 0 \quad \text{on } U.$$

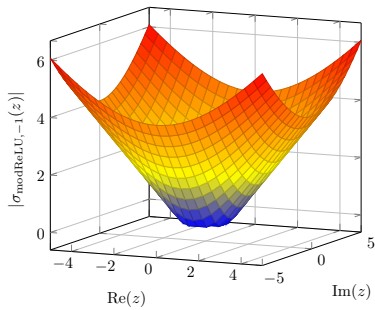 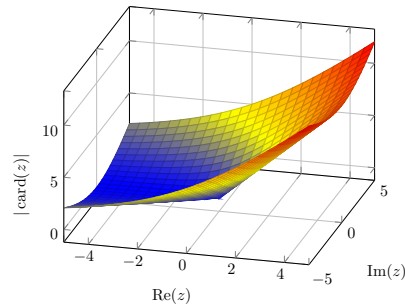

Figure 2: Absolute value of the activation functions $\sigma_{\mathrm{modReLU},-1}$ (left) and $\mathrm{card}$ (right).

Here, $\Delta := \frac{\partial^2}{\partial x^2} + \frac{\partial^2}{\partial y^2} = 4\partial_{\mathrm{wirt}}\overline{\partial}_{\mathrm{wirt}}$ denotes the usual Laplace-Operator.

The following Proposition 2.1 describes a property of non-polyharmonic functions which is crucial for proving the approximation results of this paper.

**Proposition 2.1.** *Let $\varnothing \neq U \subseteq \mathbb{C}$ be an open set and let $\phi \in C^\infty(U;\mathbb{C})$ be non-polyharmonic. Then for every $M \in \mathbb{N}_0$ there exists a point $z_M \in U$ satisfying*

$$\partial_{\mathrm{wirt}}^m \overline{\partial}_{\mathrm{wirt}}^\ell \phi(z_M) \neq 0 \quad \text{for all } m, \ell \in \mathbb{N}_0 \text{ with } m, \ell \leq M.$$

The proof of Proposition 2.1 is an application of the Baire category theorem; see Appendix B.2.

**Important complex activation functions.** We briefly discuss in how far two commonly used complex activation functions satisfy our assumptions: The *modReLU* proposed in [5] and the *complex cardioid* used in [44] for MRI fingerprinting where the performance could be significantly improved using complex-valued neural networks. The modReLU is defined as

$$\sigma_{\mathrm{modReLU},b}: \quad \mathbb{C} \to \mathbb{C}, \quad \sigma_{\mathrm{modReLU},b}(z) := \begin{cases} (|z| + b)\frac{z}{|z|}, & \text{if } |z| + b \geq 0, \\ 0, & \text{otherwise,} \end{cases}$$

where $b < 0$ is a fixed parameter. The complex cardioid is defined as

$$\mathrm{card}: \quad \mathbb{C} \to \mathbb{C}, \quad \mathrm{card}(z) := \frac{1}{2}(1 + \cos(\sphericalangle z))z.$$

Here, $\sphericalangle z = \theta \in \mathbb{R}$ denotes the polar angle of a complex number $z = re^{i\theta}$, where we define $\sphericalangle 0 := 0$; see Figure 2 for plots of the absolute value of the two functions.

Both functions are smooth and non-polyharmonic on a non-empty open subset of $\mathbb{C}$, which is proven in Appendix A.2. Furthermore, they are both continuous on $\mathbb{C}$. Therefore, our approximation bounds established in Theorems 3.1 and 3.2 in particular apply to those two functions.

## 3 Main results

In this section we state the main results of this paper and provide proof sketches for them. Detailed proofs of the two statements can be found in Appendices B.3 and C.3.

First we show in Theorem 3.1 that it is possible to approximate any complex polynomial in $z$ and $\overline{z}$ arbitrarily well using shallow CVNNs with the size of the networks only depending on the degree of the polynomial (not on the desired approximation accuracy). Using this result we can prove the main approximation bound, Theorem 3.2, by first approximating a given $C^k$-function using a polynomial in $z$ and $\overline{z}$ and then approximating this polynomial using Theorem 3.1.

For $m, n \in \mathbb{N}$ let

$$\mathcal{P}_m^n := \left\{ \mathbb{C}^n \to \mathbb{C}, \ z \mapsto \sum_{\mathbf{m} \leq m} \sum_{\boldsymbol{\ell} \leq m} a_{\mathbf{m},\boldsymbol{\ell}} z^{\mathbf{m}} \overline{z}^{\boldsymbol{\ell}} : \ a_{\mathbf{m},\boldsymbol{\ell}} \in \mathbb{C} \right\}$$

denote the space of all complex polynomials on $\mathbb{C}^n$ in $z$ and $\overline{z}$ of componentwise degree at most $m$. Here, we are summing over all multi-indices $\mathbf{m}, \boldsymbol{\ell} \in \mathbb{N}_0^n$ with $\mathbf{m}_j, \boldsymbol{\ell}_j \leq m$ for every $j \in \{1, ..., n\}$ and use the notation

$$z^{\mathbf{m}} := \prod_{j=1}^n z_j^{\mathbf{m}_j} \quad \text{and} \quad \overline{z}^{\boldsymbol{\ell}} := \prod_{j=1}^n \overline{z_j}^{\boldsymbol{\ell}_j}.$$

The space $\mathcal{P}_m^n$ is finite-dimensional; hence, it makes sense to talk about bounded subsets of $\mathcal{P}_m^n$ without specifying a norm.

**Theorem 3.1.** *Let $m, n \in \mathbb{N}$, $\varepsilon > 0$ and $\phi : \mathbb{C} \to \mathbb{C}$ be smooth and non-polyharmonic on an open set $\varnothing \neq U \subseteq \mathbb{C}$. Let $\mathcal{P}' \subseteq \mathcal{P}_m^n$ be bounded and set $N := (4m+1)^{2n}$. Then there exists a first layer $\Phi \in \mathcal{F}_{n,N}^\phi$ with the following property: For each polynomial $p \in \mathcal{P}'$ there exists $\sigma \in \mathbb{C}^N$, such that*

$$\left\| p - \sigma^T \Phi \right\|_{L^\infty(\Omega_n; \mathbb{C})} \leq \varepsilon.$$

*Sketch of proof.* For any multi-indices $\mathbf{m}, \boldsymbol{\ell} \in \mathbb{N}_0^n$ an inductive argument shows for every fixed $z \in \Omega_n$ and $b \in \mathbb{C}$ that

$$\partial_{\text{wirt}}^{\mathbf{m}} \overline{\partial}_{\text{wirt}}^{\boldsymbol{\ell}} \left[ w \mapsto \phi(w^T z + b) \right] = z^{\mathbf{m}} \overline{z}^{\boldsymbol{\ell}} \cdot \left( \partial_{\text{wirt}}^{|\mathbf{m}|} \overline{\partial}_{\text{wirt}}^{|\boldsymbol{\ell}|} \phi \right) (w^T z + b).$$

Here, $\partial_{\text{wirt}}^{\mathbf{m}}$ and $\overline{\partial}_{\text{wirt}}^{\boldsymbol{\ell}}$ denote the multivariate Wirtinger derivatives with respect to $w$ according to the multi-indices $\mathbf{m}$ and $\boldsymbol{\ell}$, respectively. Evaluating this at $w = 0$ and taking $b \in \mathbb{C}$ such that none of the mixed Wirtinger derivatives of $\phi$ at $b$ up to a sufficiently high order vanish (where such a $b$ exists by Proposition 2.1) shows that we can rewrite

$$z^{\mathbf{m}} \overline{z}^{\boldsymbol{\ell}} = \left( \partial_{\text{wirt}}^{\mathbf{m}} \overline{\partial}_{\text{wirt}}^{\boldsymbol{\ell}} \left[ w \mapsto \phi(w^T z + b) \right] \right) \Big|_{w=0} \cdot \left( \left( \partial_{\text{wirt}}^{|\mathbf{m}|} \overline{\partial}_{\text{wirt}}^{|\boldsymbol{\ell}|} \phi \right)(b) \right)^{-1}. \tag{3.1}$$

The mixed Wirtinger derivatives can by definition be expressed as linear combinations of usual partial derivatives. Those partial derivatives can be approximated using a generalized version of *divided differences*: If $g \in C^k((-r, r)^s; \mathbb{R})$ and $\mathbf{p} \in \mathbb{N}_0^s$ with $|\mathbf{p}| \leq k$ we have

$$\partial^{\mathbf{p}} g(0) \approx (2h)^{-|\mathbf{p}|} \sum_{0 \leq \mathbf{r} \leq \mathbf{p}} (-1)^{|\mathbf{p}| - |\mathbf{r}|} \binom{\mathbf{p}}{\mathbf{r}} g\left( h(2\mathbf{r} - \mathbf{p}) \right) \quad \text{for } h \searrow 0. \tag{3.2}$$

See Appendix B.1 for a proof of this approximation. Note that when one takes $g(w) = \phi(w^T z + b)$, the right-hand side of (3.2) is a shallow neural network, as a function of $z$.

Combining (3.1) and (3.2) yields the desired result; see Appendix B.3 for the details. $\qquad\square$

It is crucial that the size of the networks considered in Theorem 3.1 is independent of the approximation accuracy $\varepsilon$. Moreover, the first layer $\Phi$ can be chosen independently of the particular polynomial $p$. Only the weights $\sigma$ connecting hidden layer and output neuron have to be adapted to $p$.

The final approximation result is as follows. Its full proof can be found in Appendix C.3.

**Theorem 3.2.** *Let $n, k \in \mathbb{N}$. Then there exists a constant $c = c(n, k) > 0$ with the following property: For any activation function $\phi : \mathbb{C} \to \mathbb{C}$ that is smooth and non-polyharmonic on an open set $\varnothing \neq U \subseteq \mathbb{C}$ and for any $m \in \mathbb{N}$ there exists a first layer $\Phi \in \mathcal{F}_{n,m}^\phi$ with the following property: For any $f \in C^k(\Omega_n; \mathbb{C})$ there exist coefficients $\sigma = \sigma(f) \in \mathbb{C}^m$, such that*

$$\left\| f - \sigma^T \Phi \right\|_{L^\infty(\Omega_n; \mathbb{C})} \leq c \cdot m^{-k/(2n)} \cdot \| f \|_{C^k(\Omega_n; \mathbb{C})}.$$

*Furthermore, the map $f \mapsto \sigma(f)$ is a continuous linear operator with respect to the $L^\infty$-norm.*

*Sketch of proof.* By splitting $f$ into real and imaginary part we may assume that $f$ is real-valued. Fourier-analytical results (recalled in Appendix C.1) imply that each $C^k$-function $f$ can be well approximated by a linear combination of (multivariate) *Chebyshev polynomials*. Precisely, we have

$$\| f - P \|_{L^\infty(\Omega_n; \mathbb{R})} \leq \frac{c}{m^k} \cdot \| f \|_{C^k(\Omega_n; \mathbb{R})},$$

where $P$ is given via the formula

$$P(z) = \sum_{0 \le \mathbf{k} \le 2m-1} \mathcal{V}_{\mathbf{k}}^m(f) \cdot T_{\mathbf{k}}(z), \quad z \in \Omega_n.$$

Here, the functions $T_{\mathbf{k}}$ are multivariate versions of Chebyshev polynomials and $\mathcal{V}_{\mathbf{k}}^m(f)$ are continuous linear functionals in $f$. The constant $c > 0$ only depends on $n$ and $k$. See Appendix C.1 for a rigorous proof of this approximation property. Approximating the polynomials $T_{\mathbf{k}}$ by neural networks according to Theorem 3.1 yields the desired claim; see Appendix C.3 for the details. $\qquad\square$

*Remark* 3.3. Theorem 3.1 can not only be used to derive approximation rates for the approximation of $C^k$-functions but can be applied to any function class that is well approximable by algebraic polynomials. For example, it can be used to prove an order of approximation of $\nu^{-m^{1/(2n)}/17}$ for the approximation of functions $f : \Omega_n \to \mathbb{C}$ that can be *holomorphically extended* onto some polyellipse in $\mathbb{C}^{2n}$. The parameter $\nu > 1$ specifies the size of this polyellipse. We refer the interested reader to Appendix D for detailed definitions, statements and proofs for this fact.

## 4 Optimality of the derived approximation rate

In this section we discuss the optimality of the approximation rate proven in Theorem 3.2. We first deduce from general results by DeVore et al. [19] that the rate is optimal in the setting that the map which assigns to a function $f \in C^k(\Omega_n; \mathbb{C})$ the weights of the approximating network is continuous, as is the case in Theorem 3.2. However, it might be possible to achieve a better degree of approximation if this map is *not* required to be continuous, which is discussed in the second part of this section. Proofs for all the statements in this section are given in Appendices E.1, F.2 and F.3.

**Continuous weight selection.** We begin by considering the case of continuous weight selection. As mentioned in the introduction, this is a natural assumption, since in classical training algorithms such as (S)GD, continuous operations based on samples $f(x_j)$ are used to adjust the weights.

In [19, Theorem 4.2] a lower bound of $m^{-k/s}$ is established for the rate of approximating functions $f$ of Sobolev regularity $W^{k,\infty}$ in the following very general setting: The set of functions that is used for approximation can be parametrized using $m$ (real) parameters and the map that assigns to $f$ the parameters of the approximating function is continuous with respect to *some* norm on $W^{k,\infty}([-1,1]^s; \mathbb{R})$. A detailed version of the proof of that statement (for $C^k$ instead of $W^{k,\infty}$) is contained in Appendix E.1. A careful transfer of this result to the complex-valued setting yields the following theorem (see also Appendix E.1).

**Theorem 4.1.** *Let $n, k \in \mathbb{N}$. Then there exists a constant $c = c(n,k) > 0$ with the following property: For any $m \in \mathbb{N}$, any map $\overline{a} : C^k(\Omega_n; \mathbb{C}) \to \mathbb{C}^m$ that is continuous with respect to some norm on $C^k(\Omega_n; \mathbb{C})$ and any map $M : \mathbb{C}^m \to C(\Omega_n; \mathbb{C})$ we have*

$$\sup_{f \in C^k(\Omega_n; \mathbb{C}), \|f\|_{C^k(\Omega_n; \mathbb{C})} \le 1} \|f - M(\overline{a}(f))\|_{L^\infty(\Omega_n; \mathbb{C})} \ge c \cdot m^{-k/(2n)}.$$

The interpretation of Theorem 4.1 is as follows: If one approximates $C^k$-functions on $\Omega_n$ using a set of functions that can be parametrized using $m$ (complex) parameters and one assumes that the weight assignment is continuous, one cannot achieve a better rate of approximation than $m^{-k/(2n)}$. As a special case it can be deduced that the approximation rate is at most $m^{-k/(2n)}$ when approximating $C^k$-functions using shallow CVNNs with $m$ parameters under continuous weight assignment (see Corollary E.3). One can show that this even holds for *deep* CVNNs. Hence, for continuous weight selection the rate proven in Theorem 3.2 is optimal, even in the class of (possibly) deep networks.

**Discontinuous weight selection.** When we drop the continuity assumption on the selection of the weights, the behavior of the optimal approximation rate is more subtle. Precisely, we show that there are activation functions for which the rate of approximation can be improved to $m^{-k/(2n-1)}$. On the other hand, we also show that there are activation functions for which an improvement of the approximation rate (up to logarithmic factors) is not possible.

**Theorem 4.2.** *There exists a function $\phi \in C^\infty(\mathbb{C}; \mathbb{C})$ which is non-polyharmonic with the following additional property: For every $k \in \mathbb{N}$ and $n \in \mathbb{N}$ there exists a constant $c = c(n,k) > 0$ such that*

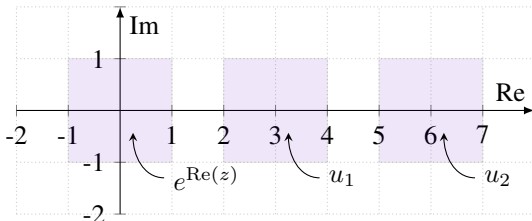

Figure 3: Illustration of the construction of the activation function in Theorem 4.2.

*for any $m \in \mathbb{N}$ and $f \in C^k(\Omega_n; \mathbb{C})$ there is a first layer $\Phi \in \mathcal{F}_{n,m}^\phi$ and a vector $\sigma \in \mathbb{C}^m$ such that*

$$\left\| f - \sigma^T \Phi \right\|_{L^\infty(\Omega_n; \mathbb{C})} \leq c \cdot m^{-k/(2n-1)} \cdot \|f\|_{C^k(\Omega_n; \mathbb{C})}.$$

*Sketch of proof.* The function $\phi$ is constructed in the following way: Take a countable dense subset $\{u_\ell\}_{\ell \in \mathbb{N}}$ of $C(\Omega_1; \mathbb{C})$, for instance the set of all polynomials in $z$ and $\overline{z}$ with coefficients in $\mathbb{Q} + i\mathbb{Q}$. Define $\phi$ in a way such that

$$\phi(z + 3\ell) = u_\ell(z)$$

for every $z \in \Omega_1$ and $\ell \in \mathbb{N}$. Furthermore, to ensure that $\phi$ is non-polyharmonic, let $\phi(z) = e^{\mathrm{Re}(z)}$ for every $z \in \Omega_1$. The smoothness of $\phi$ can be accomplished by multiplying with a smooth *bump function*; see Lemma F.4 for details. The construction of $\phi$ is illustrated in Figure 3.

Let then $f \in C^k(\Omega_n; \mathbb{C})$ be arbitrary. General results from the theory of *ridge functions* [22] show that there exist $b_1, ..., b_m \in \mathbb{C}^n$ and $g_1, ..., g_m \in C(\Omega_1; \mathbb{C})$ such that

$$\left\| f(z) - \sum_{j=1}^m g_j \left( b_j^T \cdot z \right) \right\|_{L^\infty(\Omega_n; \mathbb{C})} \leq c \cdot m^{-k/(2n-1)} \cdot \|f\|_{C^k(\Omega_n; \mathbb{C})};$$

see Proposition F.3 and Appendix F.1 for a detailed proof of this fact. Approximating the functions $g_j$ by suitable functions $u_{\ell_j}$ and expressing those functions via $\phi(\bullet + 3\ell_j)$ yields the claim. □

The preceding theorem showed that there exists an activation function for which the rate in Theorem 3.2 can be strictly improved, if one allows a discontinuous weight selection. In contrast, the following theorem shows for a certain (quite natural) activation function that the rate $m^{-k/(2n)}$ *cannot* be improved (up to logarithmic factors), even if one allows a discontinuous weight assignment.

**Theorem 4.3.** *Let $n, k \in \mathbb{N}$ and*

$$\phi: \quad \mathbb{C} \to \mathbb{C}, \quad \phi(z) := \frac{1}{1 + e^{-\mathrm{Re}(z)}}.$$

*Then $\phi$ is smooth but non-polyharmonic. Furthermore, there exists a constant $c = c(n, k) > 0$ with the following property: For any $m \in \mathbb{N}_{\geq 2}$ there exists a function $f \in C^k(\Omega_n; \mathbb{C})$ with $\|f\|_{C^k(\Omega_n; \mathbb{C})} \leq 1$, such that for every $\Phi \in \mathcal{F}_{n,m}^\phi$ and $\sigma \in \mathbb{C}^m$ we have*

$$\left\| f - \sigma^T \Phi \right\|_{L^\infty(\Omega_n; \mathbb{C})} \geq c \cdot (m \cdot \ln(m))^{-k/(2n)}.$$

*Sketch of proof.* The idea of the proof is based on that of [46, Theorem 4] but instead of the bound for the VC dimension of ReLU networks used in [46], we will employ a bound for the VC dimension stated in [4, Theorem 8.11] using the real sigmoid function. For a detailed introduction to the concept of the VC dimension and related topics, see for instance [39, Chapter 6].

A technical reduction from the complex to the real case (see Appendix F.3) shows that it suffices to show the following: If $\varepsilon \in (0, \frac{1}{2})$ and $m \in \mathbb{N}$ are such that for every $f \in C^k([-1,1]^n; \mathbb{R})$ with $\|f\|_{C^k([-1,1]^n; \mathbb{R})} \leq 1$ there exists a real-valued shallow network $\mathcal{N}$ with $\gamma(x) = \frac{1}{1+e^{-x}}$ as activation function satisfying $\|f - \mathcal{N}\|_{L^\infty([-1,1]^n; \mathbb{R})} \leq \varepsilon$, then necessarily

$$m \geq c \cdot \frac{\varepsilon^{-n/k}}{\ln(1/\varepsilon)},$$

where the constant $c$ only depends on $n$ and $k$.

To show that the latter claim holds, we assume that $\varepsilon$ and $m$ have the property stated above. Take $N \in \mathbb{N}$ such that $N^{-k} \asymp \varepsilon$ and consider the grid

$$\mathcal{G} := \frac{1}{N}\{-N, ..., N\}^n \subseteq [-1, 1]^n.$$

For every $\alpha \in \mathcal{G}$ we pick a number $z_\alpha \in \{0, 1\}$ arbitrarily and construct a map $f \in C^\infty([-1, 1]^n; \mathbb{R})$ satisfying $f(\alpha) = z_\alpha$ for very $\alpha \in \mathcal{G}$. Scaling of $f$ to $\tilde{f}$ ensures $\|\tilde{f}\|_{C^k([-1,1]^n;\mathbb{R})} \leq 1$, but then $\tilde{f}(\alpha) = c_0 \cdot z_\alpha \cdot N^{-k}$ where $c_0 = c_0(n, k) > 0$. By assumption, we can infer the existence of a shallow real-valued neural network $\mathcal{N}$ with $\gamma$ as activation function and $m$ hidden neurons satisfying $\|\tilde{f} - \mathcal{N}\|_{L^\infty([-1,1]^n;\mathbb{R})} \leq \varepsilon$. But this shows

$$\mathcal{N}(\alpha) \begin{cases} > \tilde{c}N^{-k}, & \text{if } z_\alpha = 1, \\ < \tilde{c}N^{-k}, & \text{if } z_\alpha = 0 \end{cases} \quad \text{for all } \alpha \in \mathcal{G}$$

with a constant $\tilde{c} = \tilde{c}(n, k) > 0$. Since the $z_\alpha$ are arbitrary, it follows that the set

$$H := \left\{ \mathbb{1}(\mathcal{N} > \tilde{c}N^{-k})\big|_\mathcal{G} : \mathcal{N} \text{ shallow NN with activation } \gamma \text{ and } m \text{ hidden neurons} \right\}$$

*shatters* the whole grid $\mathcal{G}$. This yields $\text{VC}(H) \geq |\mathcal{G}| = (2N + 1)^n$. On the other hand, the bound provided by [4, Theorem 8.11] for linear threshold networks yields $\text{VC}(H) \lesssim m \cdot \ln(N)$. Combining the two bounds and using $N^{-k} \asymp \varepsilon$ yields the claim. $\qquad\square$

## 5 Tractability of the considered problem in terms of the input dimension

In this section we discuss the *tractability* (in terms of the input dimension $n$) of the considered problem, i.e., the dependence of the approximation error on $n$. We show a novel result stating that, assuming a continuous weight selection, the problem of approximating $C^k$-functions is *intractable*, i.e., that the number of neurons that is required to achieve a non-trivial approximation error is necessarily exponential in $n$. In the literature this is referred to as the *curse of dimensionality*. The proof of the theorem combines ideas from [19] and [35] and is contained in Appendix E.2.

**Theorem 5.1.** *Let $s \in \mathbb{N}$. With $\|\cdot\|_{C^k([-1,1]^s;\mathbb{R})}$ defined similarly to (2.1), we write*

$$\|f\|_{C^\infty([-1,1]^s;\mathbb{R})} := \sup_{k \in \mathbb{N}} \|f\|_{C^k([-1,1]^s;\mathbb{R})} \in [0, \infty]$$

*for any function $f \in C^\infty([-1, 1]^s; \mathbb{R})$ and denote by $C^{\infty,*,s}$ the set of all $f \in C^\infty([-1, 1]^s; \mathbb{R})$ for which this expression is finite. Let $\overline{a} : C^{\infty,*,s} \to \mathbb{R}^{2^s-1}$ be continuous with respect to some norm on $C^{\infty,*,s}$ and moreover, let $M : \mathbb{R}^{2^s-1} \to C([-1, 1]^s; \mathbb{R})$ be an arbitrary map. Then it holds*

$$\sup_{\substack{f \in C^{\infty,*,s} \\ \|f\|_{C^\infty([-1,1]^s;\mathbb{R})} \leq 1}} \|f - M(\overline{a}(f))\|_{L^\infty([-1,1]^s;\mathbb{R})} \geq 1.$$

Note that Theorem 5.1 is formulated for real-valued functions but can be transferred to the complex-valued setting (see Corollary E.6). We decided to include the real-valued statement because it is expected to be of greater interest in the community than the complex-valued analog. Moreover, we stress that Theorem 5.1 is not limited to the class of shallow neural networks but refers to any function class that is parametrizable using finitely many parameters (in particular, e.g., the class of neural networks with possibly more than one hidden layer).

We now examine in what way the constant $c$ appearing in Theorem 3.2 suffers from the curse of dimensionality. To this end, it is convenient to rewrite the result from Theorem 3.2 as

$$\sup_{\|f\|_{C^k(\Omega_n;\mathbb{C})} \leq 1} \inf_{\Phi \in \mathcal{F}^\phi_{n,m}, \sigma \in \mathbb{C}^m} \|f - \sigma^T \Phi\|_{L^\infty(\Omega_n;\mathbb{C})} \leq (\tilde{c} \cdot m)^{-k/(2n)}$$

where the constant $\tilde{c} = \tilde{c}(n, k) > 0$ is independent of $m$. Writing the result in that way, one sees immediately that, if one seeks to have a worst-case approximation error of less than $\varepsilon > 0$, it is

sufficient to take $m = \left\lceil \frac{1}{\tilde{c}} \cdot \varepsilon^{-(2n)/k} \right\rceil$ neurons in the hidden layer of the network. Corollary E.7 shows that it *necessarily* holds $\tilde{c} \leq 16 \cdot 2^{-n}$ and therefore, the constant $\tilde{c}$ unavoidably suffers from the curse of dimensionality. An analysis of the constant (where we refer to Appendices C.1 to C.3 for the details) shows that in our case we can establish the bound $\tilde{c}(n, k) \geq \exp(-C \cdot n^2) \cdot k^{-4n}$ with an *absolute* constant $C > 0$. We remark that, since the constant suffers from the curse of dimensionality in any case, we have not put much effort into optimizing the constant; there is therefore probably ample room for improvement.

## 6  Limitations

To conduct a comprehensive evaluation of machine learning algorithms, one must analyze the questions of approximation, generalization, and optimization through training algorithms. The present paper, however, only focuses on the aspect of approximation. Analyzing if the proven approximation rate can be attained with learning algorithms such as (stochastic) gradient descent falls outside the scope of this paper. Furthermore, the examination of approximation rates under possibly discontinuous weight assignment is not yet fully resolved by our results. It is an open question which rate is optimally achievable in that case, depending on the choice of the activation function, and specifically in distinguishing between shallow and deep networks. We want to mention the two following points which indicate that this is a quite subtle question:

1. For deep NNs (with more than two hidden layers) with general smooth activation function, it is *not possible* to derive any non-trivial lower bounds in the setting of unrestricted weight assignment, since there exists an activation function with the property that NNs of *constant size* using this activation function can approximate any continuous function to arbitrary precision (see [31, Theorem 4]). Note that [31] considers real-valued NNs, but the results can be transferred to CVNNs with a suitable choice of the activation function.

2. In the real-valued case, fully general lower bounds for the approximation capabilities of shallow NNs have been derived by using results from [22] regarding the approximation properties of so-called ridge functions, i.e., functions of the form $\sum_{j=1}^{m} \phi_j(\langle a_j, x \rangle)$ with $a_j \in \mathbb{R}^d$ and each $\phi_j : \mathbb{R} \to \mathbb{R}$. It is an interesting problem to generalize these results to higher-dimensional ridge functions of the form $\sum_{j=1}^{m} \phi_j(A_j x)$, where each $\phi_j : \mathbb{R}^s \to \mathbb{R}$ and $A_j \in \mathbb{R}^{s \times d}$. This would imply lower bounds for shallow CVNNs. However, such a generalization seems to be highly non-trivial and is outside the scope of the present paper.

## 7  Conclusion

This paper analyzes error bounds for the approximation of complex-valued $C^k$-functions by means of complex-valued neural networks with smooth and non-polyharmonic activation functions. It is demonstrated that complex-valued neural networks with these activation functions achieve the *identical* approximation rate as real-valued networks that employ smooth and non-polynomial activation functions. This is an important theoretical finding, since CVNNs are on the one hand more restrictive than real-valued neural networks (since the mappings between layers should be $\mathbb{C}$-linear and not just $\mathbb{R}$-linear), but on the other hand more versatile, since the activation function is a mapping from $\mathbb{C}$ to $\mathbb{C}$ (i.e., from $\mathbb{R}^2 \to \mathbb{R}^2$) rather than from $\mathbb{R}$ to $\mathbb{R}$ as in the real case. Additionally, it is established that the proven approximation rate is optimal if one assumes a continuous weight selection. In summary, if one focuses on the approximation rate for $C^k$-functions, CVNNs have the same excellent approximation properties as real-valued networks.

The behavior of the approximation rate for unrestricted weight selection is more subtle. It is shown that a rate of $m^{-k/(2n-1)}$ can be achieved for certain activation functions (Theorem 4.2) but in general, one cannot improve on the rate that is attainable for continuous weight selection (Theorem 4.3).

While the proven approximation *rate* is optimal under the assumption of continuous weight selection, the involved constants suffer from the curse of dimensionality. Section 5, however, shows that this is inevitable in the given setting.

Such theoretical approximation results contribute to the mathematical understanding of Deep Learning. The remarkable approximation-theoretical properties of neural networks can be seen as one reason why neural networks provide outstanding results in many applications.

**Acknowledgments.** PG and FV acknowledge support by the German Science Foundation (DFG) in the context of the Emmy Noether junior research group VO 2594/1-1. FV acknowledges support by the Hightech Agenda Bavaria.

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

# A    Notation, Wirtinger derivatives and special activation functions

## A.1    Notation and Wirtinger derivatives

Throughout the paper, multi-indices (i.e., elements of $\mathbb{N}_0^n$) are denoted using boldface. For $\mathbf{m} \in \mathbb{N}_0^n$ we have the usual notation $|\mathbf{m}| = \sum_{j=1}^n \mathbf{m}_j$. For a number $m \in \mathbb{N}_0$ and another multi-index $\mathbf{p} \in \mathbb{N}_0^n$ we write $\mathbf{m} \leq m$ iff $\mathbf{m}_j \leq m$ for every $j \in \{1, ..., n\}$ and $\mathbf{m} \leq \mathbf{p}$ iff $\mathbf{m}_j \leq \mathbf{p}_j$ for every $j \in \{1, ..., n\}$. Furthermore we write

$$\binom{\mathbf{p}}{\mathbf{r}} := \prod_{j=1}^n \binom{\mathbf{p}_j}{\mathbf{r}_j}$$

for two multi-indices $\mathbf{p}, \mathbf{r} \in \mathbb{N}_0^n$ with $\mathbf{r} \leq \mathbf{p}$. For a complex vector $z \in \mathbb{C}^n$ we write

$$z^{\mathbf{m}} := \prod_{j=1}^n z_j^{\mathbf{m}_j} \quad \text{and} \quad \overline{z}^{\mathbf{m}} := \prod_{j=1}^n \overline{z_j}^{\mathbf{m}_j}.$$

For a point $x \in \mathbb{R}^n$ and $r > 0$ we define

$$B_r(x) := \{y \in \mathbb{R}^n : \|x - y\| < r\}$$

with $\|\cdot\|$ denoting the usual Euclidean distance. This definition is analogously transferred to $\mathbb{C}^n$.

$\mathbb{C}^n$ is canonically isomorphic to $\mathbb{R}^{2n}$ by virtue of the isomorphism

$$\varphi_n : \quad \mathbb{R}^{2n} \to \mathbb{C}^n, \quad (x_1, ..., x_n, y_1, ..., y_n) \mapsto (x_1 + iy_1, ..., x_n + iy_n). \tag{A.1}$$

The Wirtinger derivatives defined in Section 2 have the following properties that we are going to use, which can be found for example in [26, E.1a]. Here, we assume that $U \subseteq \mathbb{C}$ is open and $f \in C^1(U; \mathbb{C})$.

1.  $\partial_{\mathrm{wirt}}$ and $\overline{\partial}_{\mathrm{wirt}}$ are both $\mathbb{C}$-linear operators on the set $C^1(U; \mathbb{C})$.

2.  $f$ is complex-differentiable in $z \in U$ iff $\overline{\partial}_{\mathrm{wirt}} f(z) = 0$ and in this case the equality

    $$\partial_{\mathrm{wirt}} f(z) = f'(z)$$

    holds true, with $f'(z)$ denoting the complex derivative of $f$ at $z$.

3.  We have the conjugation rules

    $$\overline{\partial_{\mathrm{wirt}} f} = \overline{\partial}_{\mathrm{wirt}} \overline{f} \quad \text{and} \quad \overline{\overline{\partial}_{\mathrm{wirt}} f} = \partial_{\mathrm{wirt}} \overline{f}.$$

4.  If $g \in C^1(U; \mathbb{C})$, the following product rules for Wirtinger derivatives hold for every $z \in U$:

    $$\partial_{\mathrm{wirt}}(f \cdot g)(z) = \partial_{\mathrm{wirt}} f(z) \cdot g(z) + f(z) \cdot \partial_{\mathrm{wirt}} g(z),$$
    $$\overline{\partial}_{\mathrm{wirt}}(f \cdot g)(z) = \overline{\partial}_{\mathrm{wirt}} f(z) \cdot g(z) + f(z) \cdot \overline{\partial}_{\mathrm{wirt}} g(z).$$

    This product rule is not explicitly stated in [26] but follows easily from the product rule for $\frac{\partial}{\partial x}$ and $\frac{\partial}{\partial y}$.

5.  If $V \subseteq \mathbb{C}$ is an open set and $g \in C^1(V; \mathbb{C})$ with $g(V) \subseteq U$, then the following chain rules for Wirtinger derivatives hold true:

    $$\partial_{\mathrm{wirt}}(f \circ g) = [(\partial_{\mathrm{wirt}} f) \circ g] \cdot \partial_{\mathrm{wirt}} g + [(\overline{\partial}_{\mathrm{wirt}} f) \circ g] \cdot \partial_{\mathrm{wirt}} \overline{g},$$
    $$\overline{\partial}_{\mathrm{wirt}}(f \circ g) = [(\partial_{\mathrm{wirt}} f) \circ g] \cdot \overline{\partial}_{\mathrm{wirt}} g + [(\overline{\partial}_{\mathrm{wirt}} f) \circ g] \cdot \overline{\partial}_{\mathrm{wirt}} \overline{g}.$$

6.  If $f \in C^2(U; \mathbb{C})$ then we have

    $$\Delta f(z) = 4 \left( \partial_{\mathrm{wirt}} \overline{\partial}_{\mathrm{wirt}} f \right)(z)$$

    for every $z \in U$, with $\Delta$ denoting the usual Laplace-Operator $\Delta = \partial^{(2,0)} + \partial^{(0,2)}$ (cf. [7, equation (1.7)]).

For an open set $U \subseteq \mathbb{C}^n$, a function $f \in C^k(U; \mathbb{C})$ and a multi-index $\boldsymbol{\ell} \in \mathbb{N}_0^n$ with $|\boldsymbol{\ell}| \leq k$ we write $\partial_{\mathrm{wirt}}^{\boldsymbol{\ell}} f$ and $\overline{\partial}_{\mathrm{wirt}}^{\boldsymbol{\ell}} f$ for the iterated Wirtinger derivatives according to the multi-index $\boldsymbol{\ell}$.

**Proposition A.1.** *Let $U \subseteq \mathbb{C}^n$ be an open set and $f \in C^k(U; \mathbb{C})$. Then, identifying $f$ with the function $f \circ \varphi_n$ with $\varphi_n$ as in* (A.1)*, for any multi-indices $\boldsymbol{m}, \boldsymbol{\ell} \in \mathbb{N}_0^n$ with $|\boldsymbol{m} + \boldsymbol{\ell}| \leq k$ we have the representation*

$$\partial_{\mathrm{wirt}}^{\boldsymbol{m}} \overline{\partial}_{\mathrm{wirt}}^{\boldsymbol{\ell}} f(a) = \sum_{\substack{\boldsymbol{p} = (\boldsymbol{p}', \boldsymbol{p}'') \in \mathbb{N}_0^{2n} \\ \boldsymbol{p}' + \boldsymbol{p}'' = \boldsymbol{m} + \boldsymbol{\ell}}} b_{\boldsymbol{p}} \left( \partial^{\boldsymbol{p}} f \right)(a) \quad \forall a \in U$$

*with complex numbers $b_{\boldsymbol{p}} \in \mathbb{C}$ only depending on $\boldsymbol{p}, \boldsymbol{m}$ and $\boldsymbol{\ell}$ and writing $\boldsymbol{p} = (\boldsymbol{p}', \boldsymbol{p}'')$ for the concatenation of the multi-indices $\boldsymbol{p}'$ and $\boldsymbol{p}'' \in \mathbb{N}_0^n$. In particular, the coefficients do not depend on $f$.*

*Proof.* The proof is by induction over $\mathbf{m}$ and $\boldsymbol{\ell}$. We first assume $\mathbf{m} = 0$ and show the claim for all $\boldsymbol{\ell} \in \mathbb{N}_0^n$ with $|\boldsymbol{\ell}| \leq k$. In the case $\boldsymbol{\ell} = 0$ there is nothing to show, so we assume the claim to be true for fixed $\boldsymbol{\ell} \in \mathbb{N}_0^n$ with $|\boldsymbol{\ell}| < k$ and take $j \in \{1, ..., n\}$ arbitrarily. Then we get

$$\overline{\partial}_{\mathrm{wirt}}^{\boldsymbol{\ell} + e_j} f(a) = \overline{\partial}_{\mathrm{wirt}}^{e_j} \overline{\partial}_{\mathrm{wirt}}^{\boldsymbol{\ell}} f(a) \overset{\mathrm{IH}}{=} \sum_{\substack{\mathbf{p} = (\mathbf{p}', \mathbf{p}'') \in \mathbb{N}_0^{2n} \\ \mathbf{p}' + \mathbf{p}'' = \boldsymbol{\ell}}} b_{\mathbf{p}} \overline{\partial}_{\mathrm{wirt}}^{e_j} \left( \partial^{\mathbf{p}} f \right)(a)$$

$$= \sum_{\substack{\mathbf{p} = (\mathbf{p}', \mathbf{p}'') \in \mathbb{N}_0^{2n} \\ \mathbf{p}' + \mathbf{p}'' = \boldsymbol{\ell}}} \frac{b_{\mathbf{p}}}{2} \left( \partial^{(\mathbf{p}' + e_j, \mathbf{p}'')} f \right)(a) + \frac{i b_{\mathbf{p}}}{2} \left( \partial^{(\mathbf{p}', \mathbf{p}'' + e_j)} f \right)(a)$$

$$=: \sum_{\substack{\mathbf{p} = (\mathbf{p}', \mathbf{p}'') \in \mathbb{N}_0^{2n} \\ \mathbf{p}' + \mathbf{p}'' = \boldsymbol{\ell} + e_j}} b_{\mathbf{p}} \left( \partial^{\mathbf{p}} f \right)(a),$$

as was to be shown.

Since we have shown the case $\mathbf{m} = 0$ we may assume the claim to be true for fixed $\mathbf{m}, \boldsymbol{\ell} \in \mathbb{N}_0^n$ with $|\mathbf{m} + \boldsymbol{\ell}| < k$. Then we deduce

$$\partial_{\mathrm{wirt}}^{\mathbf{m} + e_j} \overline{\partial}_{\mathrm{wirt}}^{\boldsymbol{\ell}} f(a) = \partial_{\mathrm{wirt}}^{e_j} \partial_{\mathrm{wirt}}^{\mathbf{m}} \overline{\partial}_{\mathrm{wirt}}^{\boldsymbol{\ell}} f(a) \overset{\mathrm{IH}}{=} \sum_{\substack{\mathbf{p} = (\mathbf{p}', \mathbf{p}'') \in \mathbb{N}_0^{2n} \\ \mathbf{p}' + \mathbf{p}'' = \mathbf{m} + \boldsymbol{\ell}}} b_{\mathbf{p}} \partial_{\mathrm{wirt}}^{e_j} \left( \partial^{\mathbf{p}} f \right)(a)$$

$$= \sum_{\substack{\mathbf{p} = (\mathbf{p}', \mathbf{p}'') \in \mathbb{N}_0^{2n} \\ \mathbf{p}' + \mathbf{p}'' = \mathbf{m} + \boldsymbol{\ell}}} \frac{b_{\mathbf{p}}}{2} \left( \partial^{(\mathbf{p}' + e_j, \mathbf{p}'')} f \right)(a) - \frac{i b_{\mathbf{p}}}{2} \left( \partial^{(\mathbf{p}', \mathbf{p}'' + e_j)} f \right)(a)$$

$$=: \sum_{\substack{\mathbf{p} = (\mathbf{p}', \mathbf{p}'') \in \mathbb{N}_0^{2n} \\ \mathbf{p}' + \mathbf{p}'' = \mathbf{m} + \boldsymbol{\ell} + e_j}} b_{\mathbf{p}} \left( \partial^{\mathbf{p}} f \right)(a).$$

Hence the claim follows using the principle of mathematical induction. $\qquad\square$

Proposition A.1 is technical but crucial: In the course of the paper we will need to approximate Wirtinger derivatives of certain functions. In fact, however, we will approximate real derivatives and take advantage of the fact that Wirtinger derivatives are linear combinations of these.

## A.2 Concrete examples of activation functions

In this section we analyze concrete activation functions that are commonly used in practical applications of complex-valued neural networks. We are going to show that those activation functions are smooth and non-polyharmonic on some non-empty open subset of $\mathbb{C}$. Our first result analyzes a family of "real activation functions interpreted as complex activation functions".

**Proposition A.2.** *Let $\rho \in C^\infty(\mathbb{R}; \mathbb{R})$ be non-polynomial and let $\psi : \mathbb{C} \to \mathbb{C}$ be defined as*

$$\psi(z) := \rho(\mathrm{Re}(z)) \quad resp. \quad \psi(z) := \rho(\mathrm{Im}(z)).$$

*Then $\psi$ is smooth and non-polyharmonic.*

*Proof.* Since $\psi$ depends only on the real-, resp. imaginary part of the input, we see directly from the definition of the Wirtinger derivatives that

$$\partial_{\text{wirt}}\psi(z) = \overline{\partial}_{\text{wirt}}\psi(z) = \frac{1}{2}\rho'(\text{Re}(z)) \quad \text{resp.} \quad \partial_{\text{wirt}}\psi(z) = -\overline{\partial}_{\text{wirt}}\psi(z) = -\frac{i}{2}\rho'(\text{Im}(z)).$$

Hence we see for arbitrary $m, \ell \in \mathbb{N}_0$ that

$$\left|\partial_{\text{wirt}}^m \overline{\partial}_{\text{wirt}}^\ell \psi(z)\right| = \frac{1}{2^{m+\ell}}\left|\rho^{(m+\ell)}(\text{Re}(z))\right| \quad \text{resp.} \quad \left|\partial_{\text{wirt}}^m \overline{\partial}_{\text{wirt}}^\ell \psi(z)\right| = \frac{1}{2^{m+\ell}}\left|\rho^{(m+\ell)}(\text{Im}(z))\right|.$$

Since $\rho$ is non-polynomial we can choose a real number $x$, such that $\rho^{(n)}(x) \neq 0$ for all $n \in \mathbb{N}_0$ (cf. for instance [20, p. 53]). Thus, $\partial_{\text{wirt}}^m \overline{\partial}_{\text{wirt}}^\ell \psi(z) \neq 0$ for all $m, \ell \in \mathbb{N}_0$, whenever $z \in \mathbb{C}$ with $\text{Re}(z) = x$, or $\text{Im}(z) = y$, respectively. $\qquad\square$

Next, we consider the modReLU which was defined in Section 2.

**Theorem A.3.** *Let $b \in (-\infty, 0)$. Writing $\sigma = \sigma_{\text{modReLU},b}$, one has for every $z \in \mathbb{C}$ with $|z| + b > 0$ the identity*

$$\left(\partial_{\text{wirt}}^m \overline{\partial}_{\text{wirt}}^\ell \sigma\right)(z) = \begin{cases} z + b\frac{z}{|z|}, & m = \ell = 0, \\ 1 + \frac{b}{2}\cdot\frac{1}{|z|}, & m = 1, \ell = 0, \\ b \cdot q_{m,\ell} \cdot \frac{z^{\ell-m+1}}{|z|^{2\ell+1}}, & m \leq \ell + 1, \ell \geq 1, \\ b \cdot q_{m,\ell} \cdot \frac{\overline{z}^{m-\ell-1}}{|z|^{2m-1}}, & m \geq \ell + 1, m \geq 2 \end{cases}$$

*for every $m \in \mathbb{N}_0$ and $\ell \in \mathbb{N}_0$. Here, the numbers $q_{m,\ell}$ are non-zero and rational. Furthermore, note that all cases for choices of $m$ and $\ell$ are covered, by observing that we can either have the case $m \geq \ell + 1$ (where either $m \geq 2$ or $m = 1, \ell = 0$) or $m \leq \ell + 1$, where the latter case is again split into $\ell = 0$ and $\ell \geq 1$.*

*Proof.* We first calculate certain Wirtinger derivatives for $z \neq 0$. First note

$$\begin{aligned} \overline{\partial}_{\text{wirt}}\left(\frac{1}{|z|^m}\right) &= \frac{1}{2}\left(\partial^{(1,0)}\left(\frac{1}{|z|^m}\right) + i \cdot \partial^{(0,1)}\left(\frac{1}{|z|^m}\right)\right) \\ &= \frac{1}{2}\left(\left(-\frac{m}{2}\right)\frac{2\,\text{Re}(z) + i \cdot 2\,\text{Im}(z)}{|z|^{m+2}}\right) \\ &= -\frac{m}{2}\cdot\frac{z}{|z|^{m+2}} \end{aligned}$$

and similarly

$$\partial_{\text{wirt}}\left(\frac{1}{|z|^m}\right) = -\frac{m}{2}\cdot\frac{\overline{z}}{|z|^{m+2}}$$

for any $m \in \mathbb{N}$. Using the product rule for Wirtinger derivatives (see Appendix A.1), we see

$$\overline{\partial}_{\text{wirt}}\left(\frac{z^\ell}{|z|^m}\right) = \underbrace{\overline{\partial}_{\text{wirt}}\left(z^\ell\right)}_{=0}\cdot\frac{1}{|z|^m} + z^\ell\cdot\overline{\partial}_{\text{wirt}}\left(\frac{1}{|z|^m}\right) = -\frac{m}{2}\cdot\frac{z^{\ell+1}}{|z|^{m+2}} \tag{A.2}$$

for any $m \in \mathbb{N}$ and $\ell \in \mathbb{N}_0$ and furthermore

$$\begin{aligned} \partial_{\text{wirt}}\left(\frac{z^\ell}{|z|^m}\right) &= \partial_{\text{wirt}}\left(z^\ell\right)\cdot\frac{1}{|z|^m} + z^\ell\cdot\partial_{\text{wirt}}\left(\frac{1}{|z|^m}\right) \\ &= \ell\cdot z^{\ell-1}\cdot\frac{1}{|z|^m} - z^\ell\cdot\frac{m}{2}\cdot\frac{\overline{z}}{|z|^{m+2}} \\ &= \left(\ell - \frac{m}{2}\right)\cdot\frac{z^{\ell-1}}{|z|^m} \end{aligned} \tag{A.3}$$

for $m, \ell \in \mathbb{N}$, and finally

$$\partial_{\text{wirt}}\left(\frac{\overline{z}^\ell}{|z|^m}\right) = \underbrace{\partial_{\text{wirt}}\left(\overline{z}^\ell\right)}_{=0}\cdot\frac{1}{|z|^m} + \overline{z}^\ell\cdot\partial_{\text{wirt}}\left(\frac{1}{|z|^m}\right) = -\frac{m}{2}\cdot\frac{\overline{z}^{\ell+1}}{|z|^{m+2}} \tag{A.4}$$

for $m \in \mathbb{N}$ and $\ell \in \mathbb{N}_0$.

Having proven those three identities, we can start with the actual computation. We first fix $m = 0$ and prove the claimed identity by induction over $\ell$. The case $\ell = 0$ follows directly from the definition of the modReLU and for $\ell = 1$, we note

$$\overline{\partial}_{\text{wirt}}\sigma(z) = \underbrace{\overline{\partial}_{\text{wirt}}(z)}_{=0} + b \cdot \overline{\partial}_{\text{wirt}}\left(\frac{z}{|z|}\right) \stackrel{(A.2)}{=} b \cdot \left(-\frac{1}{2}\right)\frac{z^2}{|z|^3},$$

which is the claimed form. Then, using induction, we compute

$$\overline{\partial}_{\text{wirt}}^{\ell+1}\sigma(z) = \overline{\partial}_{\text{wirt}}\left(b \cdot q_{0,\ell} \cdot \frac{z^{\ell+1}}{|z|^{2\ell+1}}\right) \stackrel{(A.2)}{=} b \cdot \underbrace{q_{0,\ell} \cdot \left(-\frac{2\ell+1}{2}\right)}_{=:q_{0,\ell+1}} \cdot \frac{z^{\ell+2}}{|z|^{2\ell+3}},$$

so that the case $m = 0$ is complete.

Now we deal with the case $m \le \ell + 1$. The case $\ell = 0$ and $m = 1$ is proven by computing

$$\partial_{\text{wirt}}\sigma(z) = \partial_{\text{wirt}}(z) + b \cdot \partial_{\text{wirt}}\left(\frac{z}{|z|}\right) \stackrel{(A.3)}{=} 1 + b \cdot \frac{1}{2} \cdot \frac{1}{|z|}$$

so we can assume $\ell > 0$. Since we already dealt with the case $m = 0$, we can inductively assume the claim to be true for a fixed $m \le \ell$. Then we compute

$$\left(\partial_{\text{wirt}}^{m+1}\overline{\partial}_{\text{wirt}}^{\ell}\sigma\right)(z) = \partial_{\text{wirt}}\left(b \cdot q_{m,\ell} \cdot \frac{z^{\ell-m+1}}{|z|^{2\ell+1}}\right) \stackrel{(A.3)}{=} b \cdot \underbrace{q_{m,\ell} \cdot \left(-m + \frac{1}{2}\right)}_{=:q_{m+1,\ell}} \cdot \frac{z^{\ell-m}}{|z|^{2\ell+1}},$$

which is of the desired form. Note that (A.3) is indeed applicable because $\ell - m + 1 \ge 1$.

Finally, we consider the case where $m \ge \ell + 1$ and $m \ge 2$. The case $m = \ell + 1$ has already been shown. Using induction, we see

$$\left(\partial_{\text{wirt}}^{m+1}\overline{\partial}_{\text{wirt}}^{\ell}\sigma\right)(z) = \partial_{\text{wirt}}\left(\delta_{(m,\ell)=(1,0)} + b \cdot q_{m,\ell} \cdot \frac{\overline{z}^{m-\ell-1}}{|z|^{2m-1}}\right)$$

$$\stackrel{(A.4)}{=} b \cdot \underbrace{q_{m,\ell} \cdot \left(-m + \frac{1}{2}\right)}_{=:q_{m+1,\ell}} \cdot \frac{\overline{z}^{m-\ell}}{|z|^{2m+1}},$$

so the proof is complete. $\qquad\square$

From Theorem A.3 we can now deduce that the modReLU is smooth and non-polyharmonic on some non-empty open subset of $\mathbb{C}$.

**Corollary A.4.** *Let $b \in (-\infty, 0)$ and $z \in \mathbb{C}$ with $|z| > -b$. Then we have*

$$\partial_{\text{wirt}}^{m}\overline{\partial}_{\text{wirt}}^{\ell}\sigma_{\text{modReLU},b}(z) \ne 0$$

*for all $m, \ell \in \mathbb{N}_0$. In particular, $\sigma_{\text{modReLU},b}$ is smooth and non-polyharmonic on the set*

$$\{z \in \mathbb{C} : |z| > -b\}.$$

*Furthermore, $\sigma_{\text{modReLU},b}$ is continuous on the entire complex plane.*

*Proof.* The first part follows from Theorem A.3 by noting that if $|z| > -b$ we have in particular $|z| > -b/2$ and $|z| > 0$. For the second part, we first note that $\sigma_{\text{modReLU},b}$ is trivially continuous in $z \in \mathbb{C}$ if $|z| \ne -b$. Hence, we assume $|z| = -b$. Take any sequence $(z_j)_{j \in \mathbb{N}}$ with $z_j \to z$ as $j \to \infty$ and note

$$|\sigma_{\text{modReLU},b}(z_j) - \sigma_{\text{modReLU},b}(z)| = |\sigma_{\text{modReLU},b}(z_j)| \to \begin{cases} 0 \to 0, & \text{if } |z_j| < b, \\ |z_j| + b \to |z| + b = 0, & \text{if } |z_j| \ge b. \end{cases}$$

This shows the continuity of $\sigma_{\text{modReLU},b}$. $\qquad\square$

Next we analyze the complex cardioid, which was defined in Section 2.

**Theorem A.5.** *For any $z \in \mathbb{C}$ with $z \neq 0$ and any $m, \ell \in \mathbb{N}_0$ we have*

$$\partial_{\text{wirt}}^m \overline{\partial}_{\text{wirt}}^\ell \operatorname{card}(z) = \begin{cases} \frac{1}{2}z + \frac{1}{4}\frac{z^2}{|z|} + \frac{|z|}{4}, & m = \ell = 0, \\ a_{m,\ell} \frac{z^{\ell-m}}{|z|^{2\ell-1}} + b_{m,\ell} \frac{z^{\ell+2-m}}{|z|^{2\ell+1}}, & m \leq \ell \neq 0, \\ \frac{1}{2} + \frac{1}{8} \cdot \frac{\overline{z}}{|z|} + \frac{3}{8} \cdot \frac{z}{|z|}, & m = \ell + 1 = 1, \\ a_{m,\ell} \frac{\overline{z}}{|z|^{2\ell+1}} + b_{m,\ell} \frac{z}{|z|^{2\ell+1}}, & m = \ell + 1 > 1, \\ a_{m,\ell} \frac{\overline{z}^{m-\ell}}{|z|^{2m-1}} + b_{m,\ell} \frac{\overline{z}^{m-\ell-2}}{|z|^{2m-3}}, & m \geq \ell + 2. \end{cases}$$

*Here, the numbers $a_{m,\ell}$ and $b_{m,\ell}$ are non-zero and rational. Furthermore, note that all cases for possible choices of $m$ and $\ell$ are covered: The case $m \leq \ell$ is split into $\ell = 0$ and $\ell \neq 0$. The case $m = \ell + 1$ is split into $m = 1$ and $m > 1$. Then, the case $m \geq \ell + 2$ remains.*

*Proof.* For the following we always assume $z \in \mathbb{C}$ with $z \neq 0$. Then we can apply the identity $\cos(\sphericalangle z) = \frac{\operatorname{Re}(z)}{|z|}$, so we can rewrite

$$\operatorname{card}(z) = \frac{1}{2}\left(1 + \frac{\operatorname{Re}(z)}{|z|}\right)z = \frac{1}{2}z + \frac{1}{4}\frac{(z + \overline{z})z}{|z|} = \frac{1}{2}z + \frac{1}{4}\frac{z^2}{|z|} + \frac{|z|}{4}. \tag{A.5}$$

This establishes the case $m = \ell = 0$. Next, we compute

$$\overline{\partial}_{\text{wirt}}(|z|) = \frac{1}{2}\left(\frac{1}{2}\frac{2\operatorname{Re}(z)}{|z|} + \frac{i}{2}\frac{2\operatorname{Im}(z)}{|z|}\right) = \frac{1}{2}\frac{z}{|z|} \tag{A.6}$$

and similarly

$$\partial_{\text{wirt}}(|z|) = \frac{1}{2}\frac{\overline{z}}{|z|}. \tag{A.7}$$

We deduce

$$\overline{\partial}_{\text{wirt}} \operatorname{card}(z) \overset{\text{(A.2),(A.6)}}{=} \underbrace{\frac{1}{4} \cdot \left(-\frac{1}{2}\right)}_{=:b_{0,1}} \cdot \frac{z^3}{|z|^3} + \underbrace{\frac{1}{8}}_{=:a_{0,1}} \cdot \frac{z}{|z|}.$$

Using induction, we derive

$$\overline{\partial}_{\text{wirt}}^{\ell+1} \operatorname{card}(z) = a_{0,\ell} \overline{\partial}_{\text{wirt}}\left(\frac{z^\ell}{|z|^{2\ell-1}}\right) + b_{0,\ell} \overline{\partial}_{\text{wirt}}\left(\frac{z^{\ell+2}}{|z|^{2\ell+1}}\right)$$

$$\overset{\text{(A.2)}}{=} \underbrace{a_{0,\ell} \cdot \left(-\frac{2\ell-1}{2}\right)}_{=:a_{0,\ell+1}} \cdot \frac{z^{\ell+1}}{|z|^{2\ell+1}} + \underbrace{b_{0,\ell} \cdot \left(-\frac{2\ell+1}{2}\right)}_{=:b_{0,\ell+1}} \cdot \frac{z^{\ell+3}}{|z|^{2\ell+3}},$$

so the claim has been shown if $m = 0$. If we now fix any $\ell \in \mathbb{N}$ and assume that the claim holds true for some $m \in \mathbb{N}_0$ with $m < \ell$, we get

$$\partial_{\text{wirt}}^{m+1} \overline{\partial}_{\text{wirt}}^\ell \operatorname{card}(z) = a_{m,\ell} \partial_{\text{wirt}}\left(\frac{z^{\ell-m}}{|z|^{2\ell-1}}\right) + b_{m,\ell} \partial_{\text{wirt}}\left(\frac{z^{\ell+2-m}}{|z|^{2\ell+1}}\right)$$

$$\overset{\text{(A.3)}}{=} \underbrace{a_{m,\ell} \cdot \left(\frac{1}{2} - m\right)}_{=:a_{m+1,\ell}} \cdot \frac{z^{\ell-m-1}}{|z|^{2\ell-1}} + \underbrace{b_{m,\ell} \cdot \left(\frac{3}{2} - m\right)}_{=:b_{m+1,\ell}} \cdot \frac{z^{\ell+2-m-1}}{|z|^{2\ell+1}},$$

so the claim holds true if $m \leq \ell$.

The case $m = \ell + 1$ is split into the case $m = 1$ and $m > 1$. If $m = 1$, then $\ell = 0$ and we compute

$$\partial_{\text{wirt}} \operatorname{card}(z) \overset{\text{(A.3),(A.5),(A.7)}}{=} \frac{1}{2} + \frac{1}{4}\left(2 - \frac{1}{2}\right) \cdot \frac{z}{|z|} + \frac{1}{8} \cdot \frac{\overline{z}}{|z|}.$$

If $m > 1$, we get

$$\partial_{\text{wirt}}^{\ell+1}\overline{\partial}_{\text{wirt}}^{\ell}\operatorname{card}(z) = a_{\ell,\ell}\partial_{\text{wirt}}\left(\frac{1}{|z|^{2\ell-1}}\right) + b_{\ell,\ell}\cdot\partial_{\text{wirt}}\left(\frac{z^2}{|z|^{2\ell+1}}\right)$$

$$\overset{(A.3),(A.4)}{=} \underbrace{a_{\ell,\ell}\cdot\left(-\frac{2\ell-1}{2}\right)}_{=:a_{\ell+1,\ell}}\cdot\frac{\overline{z}}{|z|^{2\ell+1}} + \underbrace{b_{\ell,\ell}\cdot\left(2-\frac{2\ell+1}{2}\right)}_{=:b_{\ell+1,\ell}}\cdot\frac{z}{|z|^{2\ell+1}}.$$

Next, we deal with the case $m = \ell + 2$: Here, we see

$$\partial_{\text{wirt}}^{\ell+2}\overline{\partial}_{\text{wirt}}^{\ell}\operatorname{card}(z) = \partial_{\text{wirt}}\left(\frac{1}{2}\delta_{\ell=0} + a_{\ell+1,\ell}\frac{\overline{z}}{|z|^{2\ell+1}} + b_{\ell+1,\ell}\frac{z}{|z|^{2\ell+1}}\right)$$

$$\overset{(A.3),(A.4)}{=} \underbrace{a_{\ell+1,\ell}\cdot\left(-\frac{2\ell+1}{2}\right)}_{=:a_{\ell+2,\ell}}\cdot\frac{\overline{z}^2}{|z|^{2\ell+3}} + \underbrace{b_{\ell+1,\ell}\cdot\left(1-\frac{2\ell+1}{2}\right)}_{=:b_{\ell+2,\ell}}\cdot\frac{1}{|z|^{2\ell+1}}.$$

If we assume the claim to be true for a fixed $m \geq \ell + 2$, we get

$$\partial_{\text{wirt}}^{m+1}\overline{\partial}_{\text{wirt}}^{\ell}\operatorname{card}(z) = a_{m,\ell}\cdot\partial_{\text{wirt}}\left(\frac{\overline{z}^{m-\ell}}{|z|^{2m-1}}\right) + b_{m,\ell}\cdot\partial_{\text{wirt}}\left(\frac{\overline{z}^{m-\ell-2}}{|z|^{2m-3}}\right)$$

$$\overset{(A.4)}{=} \underbrace{a_{m,\ell}\cdot\left(-\frac{2m-1}{2}\right)}_{=:a_{m+1,\ell}}\cdot\frac{\overline{z}^{m+1-\ell}}{|z|^{2m+1}} + \underbrace{b_{m,\ell}\cdot\left(-\frac{2m-3}{2}\right)}_{=:b_{m+1,\ell}}\cdot\frac{\overline{z}^{m-\ell-1}}{|z|^{2m-1}}.$$

Hence, using induction, we have proven the claimed identity. $\qquad\square$

The statement regarding the non-polyharmonicity of the complex cardioid is formulated in the following corollary.

**Corollary A.6.** *For every $z \in \mathbb{C}$ with $z \notin \mathbb{R} \cup i\mathbb{R}$ and every $m, \ell \in \mathbb{N}_0$ we have*

$$\partial_{\text{wirt}}^{m}\overline{\partial}_{\text{wirt}}^{\ell}\operatorname{card}(z) \neq 0.$$

*In particular,* card *is smooth and non-polyharmonic on $\mathbb{C}\setminus(\mathbb{R}\cup i\mathbb{R})$. Futhermore,* card *is continuous on the entire complex plane.*

*Proof.* We start with the first part of the corollary. For the following, let $z \in \mathbb{C}$ with $z \notin \mathbb{R} \cup i\mathbb{R}$, i.e., $\operatorname{Re}(z) \neq 0$ and $\operatorname{Im}(z) \neq 0$. Using the definition of card, we see

$$\operatorname{card}(z) = 0 \quad\Longleftrightarrow\quad z = 0 \quad\text{or}\quad \cos(\sphericalangle z) = -1 \quad\Longleftrightarrow\quad z \in \mathbb{R}_{\leq 0},$$

and thus, $\operatorname{card}(z) \neq 0$ since $\operatorname{Im}(z) \neq 0$. In the case $m \leq \ell \neq 0$ we use Theorem A.5 to get the following chain of implications:

$$\partial_{\text{wirt}}^{m}\overline{\partial}_{\text{wirt}}^{\ell}\operatorname{card}(z) = 0 \Leftrightarrow a_{m,\ell} + b_{m,\ell}\frac{z^2}{|z|^2} = 0 \Leftrightarrow z^2 = -|z|^2\cdot\frac{a_{m,\ell}}{b_{m,\ell}} \Rightarrow z^2 \in \mathbb{R} \Leftrightarrow z \in \mathbb{R}\cup i\mathbb{R}$$

holds, and thus $\partial_{\text{wirt}}^{m}\overline{\partial}_{\text{wirt}}^{\ell}\operatorname{card}(z) \neq 0$ for $z \notin \mathbb{R} \cup i\mathbb{R}$.

For the case $m = \ell + 1 = 1$, note

$$\partial_{\text{wirt}}\operatorname{card}(z) = 0 \Leftrightarrow \frac{1}{8}\cdot\overline{z} + \frac{3}{8}\cdot z = -\frac{|z|}{2} \Rightarrow \frac{3}{8}\operatorname{Im}(z) - \frac{1}{8}\operatorname{Im}(z) = 0 \Leftrightarrow \operatorname{Im}(z) = 0,$$

and thus $\partial_{\text{wirt}}\operatorname{card}(z) \neq 0$ since $z \notin \mathbb{R}$.

For $m = \ell + 1 > 1$, we see by considering the real- and imaginary parts that

$$\partial_{\text{wirt}}^{\ell+1}\overline{\partial}_{\text{wirt}}^{\ell}\operatorname{card}(z) = 0 \Leftrightarrow a_{m,\ell}\overline{z} + b_{m,\ell}z = 0$$

$$\overset{\operatorname{Re}(z),\operatorname{Im}(z)\neq 0}{\Leftrightarrow} a_{m,\ell} + b_{m,\ell} = 0 \text{ and } -a_{m,\ell} + b_{m,\ell} = 0$$

$$\Leftrightarrow a_{m,\ell} = b_{m,\ell} = 0,$$

and hence $\partial_{\mathrm{wirt}}^{\ell+1} \overline{\partial}_{\mathrm{wirt}}^{\ell} \operatorname{card}(z) \neq 0$, since $a_{m,\ell} \neq 0 \neq b_{m,\ell}$ by Theorem A.5.

Therefore, it remains to consider the case $m \geq \ell + 2$. But here we easily see

$$\partial_{\mathrm{wirt}}^{m} \overline{\partial}_{\mathrm{wirt}}^{\ell} \operatorname{card}(z) = 0 \quad \Leftrightarrow \quad a_{m,\ell} \frac{\overline{z}^2}{|z|^2} + b_{m,\ell} = 0 \quad \Rightarrow \quad \overline{z}^2 \in \mathbb{R} \quad \Leftrightarrow \quad z \in \mathbb{R} \cup i\mathbb{R},$$

and hence $\partial_{\mathrm{wirt}}^{m} \overline{\partial}_{\mathrm{wirt}}^{\ell} \operatorname{card}(z) \neq 0$. Since all cases have been considered, the claim follows.

Regarding the second part of the corollary, first note that card is trivially continuous on $\mathbb{C} \setminus \{0\}$. Further note that

$$|\operatorname{card}(z)| = \left| \frac{1}{2} \left( 1 + \frac{\operatorname{Re}(z)}{|z|} \right) z \right| \leq |z| \to 0$$

as $z \to 0$, showing the continuity of card on the entire complex plane $\mathbb{C}$. $\qquad\square$

# B    Postponed proofs concerning the approximation of polynomials

## B.1    Divided Differences

Divided differences are well-known in numerical mathematics as they are for example used to calculate the coefficients of an interpolation polynomial in its Newton representation. In our case, we are interested in divided differences since they can be used to obtain a generalization of the classical mean-value theorem for differentiable functions: Given an interval $I \subseteq \mathbb{R}$ and $n+1$ pairwise distinct data points $x_0, ..., x_n \in I$ as well as an $n$-times differentiable real-valued function $f : I \to \mathbb{R}$, there exists $\xi \in (\min \{x_0, ..., x_n\}, \max \{x_0, ..., x_n\})$ such that

$$f[x_0, ..., x_n] = \frac{f^{(n)}(\xi)}{n!},$$

where the left-hand side is a divided difference of $f$ (defined below). The classical mean-value theorem is obtained in the case $n = 1$. Our goal in this section is to generalize this result to a multivariate setting by considering divided differences in multiple variables. Such a generalization is probably well-known, but since we could not locate a convenient reference and to make the paper more self-contained, we provide a proof.

Let us first define divided differences formally. Given $n+1$ data points $(x_0, y_0), ..., (x_n, y_n) \in \mathbb{R} \times \mathbb{R}$ with pairwise distinct $x_k$, we define the associated divided differences recursively via

$$[y_k] := y_k, \ k \in \{0, ..., n\},$$

$$[y_k, ..., y_{k+j}] := \frac{[y_{k+1}, ..., y_{k+j}] - [y_k, ..., y_{k+j-1}]}{x_{k+j} - x_k}, \ j \in \{1, ..., n\}, \ k \in \{0, ..., n - j\}.$$

If the data points are defined by a function $f$ (i.e. $y_k = f(x_k)$ for all $k \in \{0, ..., n\}$), we write

$$[x_k, ..., x_{k+j}] \, f := [y_k, ..., y_{k+j}].$$

We first consider an alternative representation of divided differences, the so-called *Hermite-Genocchi-Formula*. To state it, we introduce the notation $\Sigma^k$ to denote a certain $k$-dimensional simplex.

**Definition B.1.** Let $s \in \mathbb{N}$. Then we define

$$\Sigma^s := \left\{ x \in \mathbb{R}^s : \ x_\ell \geq 0 \text{ for all } \ell \text{ and } \sum_{\ell=1}^{s} x_\ell \leq 1 \right\}.$$

We further set $\Sigma^0 := \{0\} \subseteq \mathbb{R}$. Denoting by $\lambda^s$ the $s$-dimensional Lebesgue-measure (and by $\lambda^0$ the counting measure), the identity $\lambda^s(\Sigma^s) = \frac{1}{s!}$ holds true; see for instance [42] and note that the case $s = 0$ can be seen directly.

In the following, we consider integrals over the sets $\Sigma^s$. These integrals are always formed with respect to the measure $\lambda^s$. However, we only write, e.g., $\int_{\Sigma^s} f(x)dx$, with the implicit understanding that the integral is formed with respect to the measure $\lambda^s$. Using this convention, we can now consider the alternative representation of divided differences.

**Lemma B.2** (Hermite-Genocchi-Formula). *Let $k \in \mathbb{N}_0$. For real numbers $a, b \in \mathbb{R}$ with $a < b$, a function $f \in C^k([a, b]; \mathbb{R})$ (where $C^0([a, b]; \mathbb{R})$ denotes the set of continuous functions) and pairwise distinct $x_0, ..., x_k \in [a, b]$, the divided difference of $f$ at the data points $x_0, ..., x_k$ satisfies the identity*

$$[x_0, ..., x_k] f = \int_{\Sigma^k} f^{(k)} \left( x_0 + \sum_{\ell=1}^{k} s_\ell (x_\ell - x_0) \right) ds. \tag{B.1}$$

*Proof.* The case $k = 0$ follows directly from $[x_0] f = f(x_0)$. For the case $k > 0$, see [6, Theorem 3.3]. $\qquad\square$

We will make use of the above formula by combining it with the following generalization of the mean-value theorem for integrals.

**Lemma B.3.** *Let $D$ be a connected and compact topological space. Let $\mathcal{A}$ be some $\sigma$-algebra over $D$ and let $\mu : \mathcal{A} \to [0, \infty)$ be a finite measure on $D$ with $\mu(D) > 0$. Let $f : D \to \mathbb{R}$ be a continuous function with respect to the standard topology on $\mathbb{R}$ and the topology on $D$. Moreover, let $f$ be measurable with respect to $\mathcal{A}$ and the Borel $\sigma$-algebra on $\mathbb{R}$. Then there exists $\xi \in D$ such that*

$$f(\xi) = \frac{1}{\mu(D)} \cdot \int_D f(x) d\mu(x).$$

*Proof.* Since $D$ is compact and connected and $f$ continuous, it follows that $f(D) \subset \mathbb{R}$ is compact and connected, and hence $f(D)$ is a compact interval in $\mathbb{R}$. Therefore, there exist $x_{\min} \in D$ and $x_{\max} \in D$ satisfying

$$f(x_{\min}) \leq f(x) \leq f(x_{\max})$$

for all $x \in D$. Thus, one gets

$$f(x_{\min}) = \frac{1}{\mu(D)} \int_D f(x_{\min}) d\mu(x) \leq \frac{1}{\mu(D)} \int_D f(x) d\mu(x)$$

$$\leq \frac{1}{\mu(D)} \int_D f(x_{\max}) d\mu(x) = f(x_{\max}).$$

The claim now follows from the fact that $f(D)$ is an interval. $\qquad\square$

We also get a convenient representation of divided differences for the case of equidistant data points.

**Lemma B.4.** *Let $f : \mathbb{R} \to \mathbb{R}$, $x_0 \in \mathbb{R}$, $k \in \mathbb{N}_0$ and $h > 0$. We consider the case of equidistant data points, meaning $x_j := x_0 + jh$ for all $j = 1, ..., k$. In this case, we have the formula*

$$[x_0, ..., x_k] f = \frac{1}{k! h^k} \cdot \sum_{r=0}^{k} (-1)^{k-r} \binom{k}{r} f(x_r). \tag{B.2}$$

*Proof.* We prove the result via induction over the number $j$ of considered data points, meaning the following: For all $j \in \{0, ..., k\}$ we have

$$[x_\ell, ..., x_{\ell+j}] f = \frac{1}{j! h^j} \cdot \sum_{r=0}^{j} (-1)^{j-r} \binom{j}{r} f(x_{\ell+r})$$

for all $\ell \in \{0, ..., k\}$ satisfying $\ell + j \leq k$. The case $j = 0$ is trivial. Therefore, we assume the claim to be true for a fixed $j \in \{0, ..., k-1\}$, and let $\ell \in \{0, ..., k\}$ be arbitrary with $\ell + j + 1 \leq k$. We then get

$$[x_\ell, ..., x_{\ell+j+1}] f = \frac{[x_{\ell+1}, ..., x_{\ell+j+1}] f - [x_\ell, ..., x_{\ell+j}] f}{x_{\ell+j+1} - x_\ell}$$

$$\overset{\text{I.H.}}{=} \frac{1}{j! h^j} \cdot \frac{\sum_{r=0}^{j} (-1)^{j-r} \binom{j}{r} (f(x_{\ell+r+1}) - f(x_{\ell+r}))}{(j+1)h}$$

$$= \frac{1}{(j+1)! h^{j+1}} \sum_{r=0}^{j} (-1)^{j-r} \binom{j}{r} (f(x_{\ell+r+1}) - f(x_{\ell+r})).$$

Using an index shift, we deduce

$$\sum_{r=0}^{j}(-1)^{j-r}\binom{j}{r}f\left(x_{\ell+r+1}\right)-\sum_{r=0}^{j}(-1)^{j-r}\binom{j}{r}f\left(x_{\ell+r}\right)$$

$$=\sum_{r=1}^{j+1}(-1)^{j+1-r}\binom{j}{r-1}f\left(x_{\ell+r}\right)+\sum_{r=0}^{j}(-1)^{j+1-r}\binom{j}{r}f\left(x_{\ell+r}\right)$$

$$=(-1)^{j+1}f\left(x_{\ell}\right)+\sum_{r=1}^{j}\left((-1)^{j+1-r}f\left(x_{\ell+r}\right)\left[\binom{j}{r-1}+\binom{j}{r}\right]\right)+f\left(x_{\ell+j+1}\right)$$

$$=\sum_{r=0}^{j+1}(-1)^{j+1-r}\binom{j+1}{r}f\left(x_{\ell+r}\right),$$

which yields the claim. $\qquad\square$

The final result for divided differences, which is the result that is actually used in the proof of Theorem 3.1 in Appendix B.3, reads as follows:

**Theorem B.5.** *Let* $f:\mathbb{R}^s\to\mathbb{R}$ *and* $k\in\mathbb{N}_0, r>0$, *such that* $f|_{(-r,r)^s}\in C^k\left((-r,r)^s;\mathbb{R}\right)$. *For* $\boldsymbol{p}\in\mathbb{N}_0^s$ *with* $|\boldsymbol{p}|\le k$ *and* $h>0$ *let*

$$f_{\boldsymbol{p},h}:=(2h)^{-|\boldsymbol{p}|}\sum_{0\le\boldsymbol{r}\le\boldsymbol{p}}(-1)^{|\boldsymbol{p}|-|\boldsymbol{r}|}\binom{\boldsymbol{p}}{\boldsymbol{r}}f\left(h(2\boldsymbol{r}-\boldsymbol{p})\right).$$

*Let* $m:=\max\limits_{j}\boldsymbol{p}_j$. *Then, for* $0<h<\frac{r}{\max\{1,m\}}$ *there exists* $\xi\in h[-m,m]^s$ *satisfying*

$$f_{\boldsymbol{p},h}=\partial^{\boldsymbol{p}}f(\xi).$$

*Proof.* We may assume $m>0$, since $m=0$ implies $\mathbf{p}=0$ and thus $f_{\mathbf{p},h}=f(0)$, so that the claim holds for $\xi=0$.

We prove via induction over $s\in\mathbb{N}$ that the formula

$$f_{\mathbf{p},h}=\mathbf{p}!\int_{\Sigma^{\mathbf{p}_s}}\int_{\Sigma^{\mathbf{p}_{s-1}}}\cdots\int_{\Sigma^{\mathbf{p}_1}}\partial^{\mathbf{p}}f\left(-h\mathbf{p}_1+2h\sum_{\ell=1}^{\mathbf{p}_1}\ell\sigma_{\ell}^{(1)},...,-h\mathbf{p}_s+2h\sum_{\ell=1}^{\mathbf{p}_s}\ell\sigma_{\ell}^{(s)}\right)d\sigma^{(1)}\cdots d\sigma^{(s)}$$

(B.3)

holds for all $\mathbf{p}\in\mathbb{N}_0^s$ with $1\le|\mathbf{p}|\le k$ and all $0<h<\frac{r}{m}$. The case $s=1$ is exactly the Hermite-Genocchi-Formula (B.1), combined with (B.2) applied to the data points

$$-hp,-hp+2h,...,hp-2h,hp.$$

By induction, assume that the claim holds for some $s\in\mathbb{N}$. For a fixed point $y\in(-r,r)$, let

$$f_y:\quad(-r,r)^s\to\mathbb{R},\quad x\mapsto f(x,y).$$

For $\mathbf{p}\in\mathbb{N}_0^{s+1}$ with $|\mathbf{p}|\le k$ and $\mathbf{p}':=(p_1,...,p_s)$, we define

$$\Gamma:\quad(-r,r)\to\mathbb{R},\quad y\mapsto(f_y)_{\mathbf{p}',h}=(2h)^{-|\mathbf{p}'|}\sum_{0\le\mathbf{r}'\le\mathbf{p}'}(-1)^{|\mathbf{p}'|-|\mathbf{r}'|}\binom{\mathbf{p}'}{\mathbf{r}'}f\left(h(2\mathbf{r}'-\mathbf{p}'),y\right).$$

Using the induction hypothesis, we get

$$\Gamma(y)$$

$$=\mathbf{p}'!\int_{\Sigma^{\mathbf{p}_s}}\int_{\Sigma^{\mathbf{p}_{s-1}}}\cdots\int_{\Sigma^{\mathbf{p}_1}}\partial^{(\mathbf{p}',0)}f\left(-h\mathbf{p}_1+2h\sum_{\ell=1}^{\mathbf{p}_1}\ell\sigma_{\ell}^{(1)},...,-h\mathbf{p}_s+2h\sum_{\ell=1}^{\mathbf{p}_s}\ell\sigma_{\ell}^{(s)},y\right)d\sigma^{(1)}\cdots d\sigma^{(s)}$$

for all $y \in (-r, r)$. Furthermore, we calculate

$$\mathbf{p}_{s+1}! \cdot [-h \cdot \mathbf{p}_{s+1}, -h \cdot \mathbf{p}_{s+1} + 2h, ..., h \cdot \mathbf{p}_{s+1}]\Gamma$$

$$\overset{(B.2)}{=} (2h)^{-\mathbf{p}_{s+1}} \sum_{r'=0}^{\mathbf{p}_{s+1}} (-1)^{\mathbf{p}_{s+1}-r'} \binom{\mathbf{p}_{s+1}}{r'} \Gamma\left(h\left(2r' - \mathbf{p}_{s+1}\right)\right)$$

$$= (2h)^{-\mathbf{p}_{s+1}} \left[ \sum_{r'=0}^{\mathbf{p}_{s+1}} (-1)^{\mathbf{p}_{s+1}-r'} \binom{\mathbf{p}_{s+1}}{r'} (2h)^{-|\mathbf{p}'|} \right.$$

$$\left. \times \sum_{0 \le \mathbf{r}' \le \mathbf{p}'} (-1)^{|\mathbf{p}'|-|\mathbf{r}'|} \binom{\mathbf{p}'}{\mathbf{r}'} f\left(h(2\mathbf{r}' - \mathbf{p}'), h(2r' - \mathbf{p}_{s+1})\right) \right]$$

$$= (2h)^{-|\mathbf{p}|} \sum_{0 \le \mathbf{r} \le \mathbf{p}} (-1)^{|\mathbf{p}|-|\mathbf{r}|} \binom{\mathbf{p}}{\mathbf{r}} f\left(h(2\mathbf{r} - \mathbf{p})\right)$$

$$= f_{\mathbf{p},h}.$$

On the other hand, we get

$$[-h \cdot \mathbf{p}_{s+1}, -h \cdot \mathbf{p}_{s+1} + 2h, ..., h \cdot \mathbf{p}_{s+1}]\Gamma$$

$$\overset{(B.1)}{=} \int_{\Sigma^{\mathbf{p}_{s+1}}} \Gamma^{(\mathbf{p}_{s+1})} \left(-h\mathbf{p}_{s+1} + 2h \sum_{\ell=1}^{\mathbf{p}_{s+1}} \ell \sigma_\ell^{(s+1)}\right) d\sigma^{(s+1)}$$

$$= \mathbf{p}'! \int_{\Sigma^{\mathbf{p}_{s+1}}} \cdots \int_{\Sigma^{\mathbf{p}_1}} \partial^{\mathbf{p}} f \left(-h\mathbf{p}_1 + 2h \sum_{\ell=1}^{\mathbf{p}_1} \ell \sigma_\ell^{(1)}, ..., -h\mathbf{p}_{s+1} + 2h \sum_{\ell=1}^{\mathbf{p}_{s+1}} \ell \sigma_\ell^{(s+1)}\right) d\sigma^{(1)} \cdots d\sigma^{(s+1)}.$$

Interchanging the order of integration and derivative is possible, since we integrate on compact sets and only consider continuously differentiable functions (see, e.g., [11, Lemma 16.2]).

We have thus proven (B.3) using the principle of induction. The claim of the theorem then follows directly using the mean-value theorem for integrals (Lemma B.3) applied to the topological space

$$D := \Sigma^{\mathbf{p}_1} \times \cdots \times \Sigma^{\mathbf{p}_{s+1}}$$

equipped with the product topology where each factor is endowed with the standard topology on $\Sigma^{\mathbf{p}_\ell}$ (where the standard topology on $\Sigma^0$ is the discrete topology), the measure

$$\mu := \lambda^{\mathbf{p}_1} \otimes \cdots \otimes \lambda^{\mathbf{p}_{s+1}}$$

defined on the product of the Borel $\sigma$-algebras on $\Sigma^{\mathbf{p}_\ell}$ (and the $\sigma$-algebra of $\{0\}$ is its power set) and to the function

$$(\sigma^{(1)}, ..., \sigma^{(s+1)}) \mapsto \partial^{\mathbf{p}} f \left(-h\mathbf{p}_1 + 2h \sum_{\ell=1}^{\mathbf{p}_1} \ell \sigma_\ell^{(1)}, ..., -h\mathbf{p}_{s+1} + 2h \sum_{\ell=1}^{\mathbf{p}_{s+1}} \ell \sigma_\ell^{(s+1)}\right).$$

Moreover, note that all the simplices $\Sigma^{\mathbf{p}_\ell}$ are compact and connected (in fact convex) with

$$\lambda^{\mathbf{p}_1}\left(\Sigma^{\mathbf{p}_1}\right) \cdots \lambda^{\mathbf{p}_{s+1}}\left(\Sigma^{\mathbf{p}_{s+1}}\right) = \frac{1}{\mathbf{p}!},$$

see Definition B.1. Therefore, Lemma B.3 yields the existence of a certain $(\xi^{(1)}, ..., \xi^{(s+1)}) \in D$ with

$$f_{\mathbf{p},h} = \partial^{\mathbf{p}} f \left(-h\mathbf{p}_1 + 2h \sum_{\ell=1}^{\mathbf{p}_1} \ell \xi_\ell^{(1)}, ..., -h\mathbf{p}_{s+1} + 2h \sum_{\ell=1}^{\mathbf{p}_{s+1}} \ell \xi_\ell^{(s+1)}\right).$$

Hence, the claim follows letting

$$\xi := \left(-h\mathbf{p}_1 + 2h \sum_{\ell=1}^{\mathbf{p}_1} \ell \xi_\ell^{(1)}, ..., -h\mathbf{p}_{s+1} + 2h \sum_{\ell=1}^{\mathbf{p}_{s+1}} \ell \xi_\ell^{(s+1)}\right) \in h[-m, m]^s. \qquad \square$$

## B.2 Proof of Proposition 2.1

*Proof of Proposition 2.1.* Let $M \in \mathbb{N}_0$. Since $\phi$ is not polyharmonic we can pick $z \in U$ with $\Delta^M \phi(z) \neq 0$. By continuity we can choose $\delta > 0$ with $B_\delta(z) \subseteq U$ and $\Delta^M \phi(w) \neq 0$ for all $w \in B_\delta(z)$. For all $m, \ell \in \mathbb{N}_0$ let

$$A_{m,\ell} := \left\{ w \in B_\delta(z) : \partial_{\mathrm{wirt}}^m \overline{\partial}_{\mathrm{wirt}}^\ell \phi(w) = 0 \right\}$$

and assume towards a contradiction that

$$\bigcup_{m,\ell \leq M} A_{m,\ell} = B_\delta(z).$$

By [3, Corollary 3.35], $B_\delta(z)$ with its usual topology is completely metrizable. By continuity of $\partial_{\mathrm{wirt}}^m \overline{\partial}_{\mathrm{wirt}}^\ell \phi$ the sets $A_{m,\ell}$ are closed in $B_\delta(z)$. Hence, using the Baire category theorem [3, Theorems 3.46 and 3.47], there are $m, \ell \in \mathbb{N}_0$ with $m, \ell \leq M$, $z' \in A_{m,\ell}$ and $\varepsilon > 0$ such that

$$B_\varepsilon(z') \subseteq A_{m,\ell} \subseteq B_\delta(z).$$

But thanks to $\Delta^M \phi = 4^M \partial_{\mathrm{wirt}}^M \overline{\partial}_{\mathrm{wirt}}^M \phi = 4^M \partial_{\mathrm{wirt}}^{M-\ell} \overline{\partial}_{\mathrm{wirt}}^{M-m} \partial_{\mathrm{wirt}}^\ell \overline{\partial}_{\mathrm{wirt}}^m \phi$ (see property 6 in Appendix A.1), this directly implies $\Delta^M \phi \equiv 0$ on $B_\varepsilon(z')$ in contradiction to the choice of $B_\delta(z)$. $\square$

## B.3 Proof of Theorem 3.1

The following section is dedicated to proving Theorem 3.1. We are going to show that it is possible to approximate complex polynomials in $z$ and $\overline{z}$ arbitrarily well on $\Omega_n$ using shallow complex-valued neural networks. To do so, we follow the proof sketch given after the statement of Theorem 3.1, starting with the following lemma.

**Lemma B.6.** *Let $\phi : \mathbb{C} \to \mathbb{C}$ and $\delta > 0$, $b \in \mathbb{C}$, $k \in \mathbb{N}_0$, such that $\phi|_{B_\delta(b)} \in C^k(B_\delta(b); \mathbb{C})$. For fixed $z \in \Omega_n$, where we recall that $\Omega_n = [-1,1]^n + i[-1,1]^n$, we consider the map*

$$\phi_z : \quad B_{\frac{\delta}{\sqrt{2n}}}(0) \to \mathbb{C}, \quad w \mapsto \phi\left(w^T z + b\right),$$

*which is in $C^k$ since for $w \in B_{\frac{\delta}{\sqrt{2n}}}(0) \subseteq \mathbb{C}^n$ we have*

$$\left| w^T z \right| \leq \|w\|_2 \cdot \|z\|_2 < \frac{\delta}{\sqrt{2n}} \cdot \sqrt{2n} = \delta.$$

*For all multi-indices $\boldsymbol{m}, \boldsymbol{\ell} \in \mathbb{N}_0^n$ with $|\boldsymbol{m} + \boldsymbol{\ell}| \leq k$ we have*

$$\partial_{\mathrm{wirt}}^{\boldsymbol{m}} \overline{\partial}_{\mathrm{wirt}}^{\boldsymbol{\ell}} \phi_z(w) = z^{\boldsymbol{m}} \overline{z}^{\boldsymbol{\ell}} \cdot \left( \partial_{\mathrm{wirt}}^{|\boldsymbol{m}|} \overline{\partial}_{\mathrm{wirt}}^{|\boldsymbol{\ell}|} \phi \right) \left( w^T z + b \right)$$

*for all $w \in B_{\frac{\delta}{\sqrt{2n}}}(0)$.*

*Proof.* First we prove the statement

$$\overline{\partial}_{\mathrm{wirt}}^{\boldsymbol{\ell}} \phi_z(w) = \overline{z}^{\boldsymbol{\ell}} \cdot (\overline{\partial}_{\mathrm{wirt}}^{|\boldsymbol{\ell}|} \phi)\left( w^T z + b \right) \quad \text{for all } w \in B_{\frac{\delta}{\sqrt{2n}}}(0) \tag{B.4}$$

by induction over $0 \leq |\boldsymbol{\ell}| \leq k$. The case $\boldsymbol{\ell} = 0$ is trivial. Assuming that (B.4) holds for fixed $\boldsymbol{\ell} \in \mathbb{N}_0^n$ with $|\boldsymbol{\ell}| < k$, we want to show

$$\overline{\partial}_{\mathrm{wirt}}^{\boldsymbol{\ell}+e_j} \phi_z(w) = \overline{z}^{\boldsymbol{\ell}+e_j} \cdot \left( \overline{\partial}_{\mathrm{wirt}}^{|\boldsymbol{\ell}|+1} \phi \right) \left( w^T z + b \right) \tag{B.5}$$

for all $w \in B_{\frac{\delta}{\sqrt{2n}}}(0)$, where $j \in \{1, ..., n\}$ is chosen arbitrarily. To this end, first note

$$\overline{\partial}_{\mathrm{wirt}}^{\boldsymbol{\ell}+e_j} \phi_z(w) = \overline{\partial}_{\mathrm{wirt}}^{e_j} \overline{\partial}_{\mathrm{wirt}}^{\boldsymbol{\ell}} \phi_z(w) \overset{\text{induction}}{=} \overline{\partial}_{\mathrm{wirt}}^{e_j} \left[ w \mapsto \overline{z}^{\boldsymbol{\ell}} \cdot \left( \overline{\partial}_{\mathrm{wirt}}^{|\boldsymbol{\ell}|} \phi \right) \left( w^T z + b \right) \right]$$

$$= \overline{z}^{\boldsymbol{\ell}} \overline{\partial}_{\mathrm{wirt}}^{e_j} \left[ w \mapsto \left( \overline{\partial}_{\mathrm{wirt}}^{|\boldsymbol{\ell}|} \phi \right) \left( w^T z + b \right) \right].$$

Applying the chain rule for Wirtinger derivatives and using that

$$\overline{\partial}_{\text{wirt}}^{e_j} \left[ w \mapsto w^T z + b \right] = 0$$

since $w \mapsto w^T z + b$ is holomorphic in every variable, we see

$$\begin{aligned}
\overline{\partial}_{\text{wirt}}^{e_j} \left[ w \mapsto \left( \overline{\partial}_{\text{wirt}}^{|\boldsymbol{\ell}|} \phi \right) (w^T z + b) \right] &= \left( \partial_{\text{wirt}} \overline{\partial}_{\text{wirt}}^{|\boldsymbol{\ell}|} \phi \right) (w^T z + b) \cdot \overline{\partial}_{\text{wirt}}^{e_j} \left[ w \mapsto w^T z + b \right] \\
&\quad + \left( \overline{\partial}_{\text{wirt}}^{|\boldsymbol{\ell}|+1} \phi \right) (w^T z + b) \cdot \overline{\partial}_{\text{wirt}}^{e_j} \left[ w \mapsto \overline{w^T z + b} \right] \\
&= \left( \overline{\partial}_{\text{wirt}}^{|\boldsymbol{\ell}|+1} \phi \right) (w^T z + b) \cdot \overline{\partial_{\text{wirt}}^{e_j} \left[ w \mapsto w^T z + b \right]} \\
&= \overline{z}^{e_j} \cdot \left( \overline{\partial}_{\text{wirt}}^{|\boldsymbol{\ell}|+1} \phi \right) (w^T z + b) \, ,
\end{aligned}$$

using the fact that $w_j \mapsto w^T z + b$ is holomorphic and hence

$$\overline{\partial}_{\text{wirt}}^{e_j} \left[ w \mapsto w^T z + b \right] = 0 \quad \text{and} \quad \partial_{\text{wirt}}^{e_j} \left[ w \mapsto w^T z + b \right] = z_j.$$

Thus, we have proven (B.5) and induction yields (B.4).

It remains to show the full claim. We use induction over $|\mathbf{m}|$ and note that the case $\mathbf{m} = 0$ has just been shown. We assume that the claim holds true for fixed $\mathbf{m} \in \mathbb{N}_0^n$ with $|\mathbf{m} + \boldsymbol{\ell}| < k$ and choose $j \in \{1, ..., n\}$. Thus, we get

$$\begin{aligned}
\partial_{\text{wirt}}^{\mathbf{m}+e_j} \overline{\partial}_{\text{wirt}}^{\boldsymbol{\ell}} \phi_z(w) = \partial_{\text{wirt}}^{e_j} \partial_{\text{wirt}}^{\mathbf{m}} \overline{\partial}_{\text{wirt}}^{\boldsymbol{\ell}} \phi_z(w) &\overset{\text{IH}}{=} \partial_{\text{wirt}}^{e_j} \left( w \mapsto z^{\mathbf{m}} \overline{z}^{\boldsymbol{\ell}} \cdot \left( \partial_{\text{wirt}}^{|\mathbf{m}|} \overline{\partial}_{\text{wirt}}^{|\boldsymbol{\ell}|} \phi \right) (w^T z + b) \right) \\
&= z^{\mathbf{m}} \overline{z}^{\boldsymbol{\ell}} \cdot \partial_{\text{wirt}}^{e_j} \left[ w \mapsto \left( \partial_{\text{wirt}}^{|\mathbf{m}|} \overline{\partial}_{\text{wirt}}^{|\boldsymbol{\ell}|} \phi \right) (w^T z + b) \right].
\end{aligned}$$

Using the chain rule again, we calculate

$$\begin{aligned}
&\partial_{\text{wirt}}^{e_j} \left[ w \mapsto \left( \partial_{\text{wirt}}^{|\mathbf{m}|} \overline{\partial}_{\text{wirt}}^{|\boldsymbol{\ell}|} \phi \right) (w^T z + b) \right] \\
&= \left( \partial_{\text{wirt}}^{|\mathbf{m}|+1} \overline{\partial}_{\text{wirt}}^{|\boldsymbol{\ell}|} \phi \right) (w^T z + b) \cdot \partial_{\text{wirt}}^{e_j} \left[ w \mapsto w^T z + b \right] \\
&\quad + \left( \partial_{\text{wirt}}^{|\mathbf{m}|} \overline{\partial}_{\text{wirt}}^{|\boldsymbol{\ell}|+1} \phi \right) (w^T z + b) \cdot \partial_{\text{wirt}}^{e_j} \left[ w \mapsto \overline{w^T z + b} \right] \\
&= z^{e_j} \cdot \left( \partial_{\text{wirt}}^{|\mathbf{m}|+1} \overline{\partial}_{\text{wirt}}^{|\boldsymbol{\ell}|} \phi \right) (w^T z + b) + \left( \partial_{\text{wirt}}^{|\mathbf{m}|} \overline{\partial}_{\text{wirt}}^{|\boldsymbol{\ell}|+1} \phi \right) (w^T z + b) \cdot \overline{\partial_{\text{wirt}}^{e_j} \left[ w \mapsto w^T z + b \right]} \\
&= z^{e_j} \cdot \left( \partial_{\text{wirt}}^{|\mathbf{m}|+1} \overline{\partial}_{\text{wirt}}^{|\boldsymbol{\ell}|} \phi \right) (w^T z + b) \, .
\end{aligned}$$

By induction, this proves the claim. $\qquad\square$

As the last preparation for the proof of Theorem 3.1, we need the following lemma.

**Lemma B.7.** *Let $\phi : \mathbb{C} \to \mathbb{C}$ and $\delta > 0$, $b \in \mathbb{C}$, $k \in \mathbb{N}_0$, such that $\phi|_{B_\delta(b)} \in C^k (B_\delta(b); \mathbb{C})$. Let $m, n \in \mathbb{N}$ and $\varepsilon > 0$. For $\boldsymbol{p} \in \mathbb{N}_0^{2n}$, $h > 0$ and $z \in \Omega_n$ we write*

$$\begin{aligned}
\Phi_{\boldsymbol{p},h}(z) &:= (2h)^{-|\boldsymbol{p}|} \sum_{0 \le \boldsymbol{r} \le \boldsymbol{p}} (-1)^{|\boldsymbol{p}|-|\boldsymbol{r}|} \binom{\boldsymbol{p}}{\boldsymbol{r}} (\phi_z \circ \varphi_n) (h(2\boldsymbol{r} - \boldsymbol{p})) \\
&= (2h)^{-|\boldsymbol{p}|} \sum_{0 \le \boldsymbol{r} \le \boldsymbol{p}} (-1)^{|\boldsymbol{p}|-|\boldsymbol{r}|} \binom{\boldsymbol{p}}{\boldsymbol{r}} \phi \left( [\varphi_n (h(2\boldsymbol{r} - \boldsymbol{p}))]^T \cdot z + b \right),
\end{aligned}$$

*where $\phi_z$ is as introduced in Lemma B.6 and $\varphi_n$ is as in (A.1). Furthermore, let*

$$\phi_{\boldsymbol{p}} : \quad \Omega_n \times B_{\frac{\delta}{\sqrt{2n}}}(0) \to \mathbb{C}, \quad (z, w) \mapsto \partial^{\boldsymbol{p}} \phi_z(w).$$

*Then there exists $h^* > 0$ such that*

$$\| \Phi_{\boldsymbol{p},h} - \phi_{\boldsymbol{p}}(\cdot, 0) \|_{L^\infty(\Omega_n; \mathbb{C})} \le \varepsilon$$

*for all $\boldsymbol{p} \in \mathbb{N}_0^{2n}$ with $|\boldsymbol{p}| \le k$ and $\boldsymbol{p} \le m$ and $h \in (0, h^*)$.*

*Proof.* Fix $\mathbf{p} \in \mathbb{N}_0^{2n}$ with $|\mathbf{p}| \le k$ and $\mathbf{p} \le m$. The map

$$B_{\sqrt{2n}+1}(0) \times B_{\delta/(\sqrt{2n}+1)}(0) \to \mathbb{C}, \quad (z,w) \mapsto \phi\left(w^T z + b\right)$$

is in $C^k$ since

$$\left|w^T z\right| \le \|w\| \cdot \|z\| < \frac{\delta}{\sqrt{2n}+1} \cdot (\sqrt{2n}+1) = \delta.$$

Therefore, the map

$$B_{\sqrt{2n}+1}(0) \times B_{\delta/(\sqrt{2n}+1)}(0) \to \mathbb{C}, \quad (z,w) \mapsto \partial^{\mathbf{P}}\phi_z(w)$$

is continuous and in particular uniformly continuous on the compact set

$$\Omega_n \times \overline{B_{\delta/(3n)}}(0) \subseteq B_{\sqrt{2n}+1}(0) \times B_{\delta/(\sqrt{2n}+1)}(0).$$

Here, we employed $\sqrt{2n} + 1 < 3n$ for every $n \in \mathbb{N}$. Hence, there exists $h_{\mathbf{p}} \in (0, \frac{\delta}{3n \cdot \sqrt{2n} \cdot m})$, such that

$$|\phi_{\mathbf{p}}(z, \xi) - \phi_{\mathbf{p}}(z, 0)| \le \frac{\varepsilon}{\sqrt{2}}$$

for all $\xi \in \varphi_n\left(h \cdot [-m, m]^{2n}\right)$, $h \in (0, h_{\mathbf{p}})$ and $z \in \Omega_n$. Now fix such an $h \in (0, h_{\mathbf{p}})$ and $z \in \Omega_n$. Applying Theorem B.5 to both components of $\left(\varphi_1^{-1} \circ \Phi_{\mathbf{p},h}\right)(z)$ and $\varphi_1^{-1} \circ \phi_z \circ \varphi_n|_{\left(-\frac{\delta}{3n}, \frac{\delta}{3n}\right)^{2n}}$ separately yields the existence of two real vectors $\xi_1, \xi_2 \in h \cdot [-m, m]^{2n}$, such that

$$\left(\varphi_1^{-1} \circ \Phi_{\mathbf{p},h}(z)\right)_1 = \left[\partial^{\mathbf{P}}\left(\varphi_1^{-1} \circ \phi_z \circ \varphi_n\right)(\xi_1)\right]_1$$

$$\text{and} \quad \left(\varphi_1^{-1} \circ \Phi_{\mathbf{p},h}(z)\right)_2 = \left[\partial^{\mathbf{P}}\left(\varphi_1^{-1} \circ \phi_z \circ \varphi_n\right)(\xi_2)\right]_2.$$

Rewriting this yields

$$\mathrm{Re}\left(\Phi_{\mathbf{p},h}(z)\right) = \mathrm{Re}\left(\phi_{\mathbf{p}}(z, \varphi_n(\xi_1))\right) \quad \text{and} \quad \mathrm{Im}\left(\Phi_{\mathbf{p},h}(z)\right) = \mathrm{Im}\left(\phi_{\mathbf{p}}(z, \varphi_n(\xi_2))\right).$$

Using this property, we deduce

$$|\mathrm{Re}\left(\Phi_{\mathbf{p},h}(z) - \phi_{\mathbf{p}}(z, 0)\right)| = |\mathrm{Re}\left(\phi_{\mathbf{p}}(z, \varphi_n(\xi_1)) - \phi_{\mathbf{p}}(z, 0)\right)| \le |\phi_{\mathbf{p}}(z, \varphi_n(\xi_1)) - \phi_{\mathbf{p}}(z, 0)| \le \frac{\varepsilon}{\sqrt{2}}$$

and analogously for the imaginary part. Since $z \in \Omega_n$ and $h \in (0, h_{\mathbf{p}})$ have been chosen arbitrarily we get the claim by choosing

$$h^* := \min\left\{h_{\mathbf{p}} : \ \mathbf{p} \in \mathbb{N}_0^{2n} \text{ with } |\mathbf{p}| \le k \text{ and } \mathbf{p} \le m\right\}. \qquad \square$$

Using the previous two lemmas and the results from Appendix B.1 and Appendix B.2, we can now prove Theorem 3.1.

*Proof of Theorem 3.1.* Let $b \in U$ satisfy

$$\partial_{\mathrm{wirt}}^{\ell_1} \overline{\partial}_{\mathrm{wirt}}^{\ell_2} \phi(b) \neq 0 \quad \text{for all } \ell_1, \ell_2 \in \mathbb{N}_0 \text{ with } \ell_1, \ell_2 \le mn.$$

Such a point $b$ exists according to Proposition 2.1. Let $p \in \mathcal{P}'$ and fix $\mathbf{m}, \boldsymbol{\ell} \in \mathbb{N}_0^n$ with $\mathbf{m}, \boldsymbol{\ell} \le m$. For each $z \in \Omega_n$, using Lemma B.6, we then have

$$z^{\mathbf{m}} \overline{z}^{\boldsymbol{\ell}} = \left[\left(\partial_{\mathrm{wirt}}^{|\mathbf{m}|} \overline{\partial}_{\mathrm{wirt}}^{|\boldsymbol{\ell}|} \phi\right)(b)\right]^{-1} \partial_{\mathrm{wirt}}^{\mathbf{m}} \overline{\partial}_{\mathrm{wirt}}^{\boldsymbol{\ell}} \phi_z(0)$$

$$\overset{\text{Prop. A.1}}{=} \left[\left(\partial_{\mathrm{wirt}}^{|\mathbf{m}|} \overline{\partial}_{\mathrm{wirt}}^{|\boldsymbol{\ell}|} \phi\right)(b)\right]^{-1} \cdot \sum_{\substack{\mathbf{p}=(\mathbf{p}',\mathbf{p}'')\in\mathbb{N}_0^{2n} \\ \mathbf{p}'+\mathbf{p}''=\mathbf{m}+\boldsymbol{\ell}}} b_{\mathbf{p}',\mathbf{p}''} \partial^{(\mathbf{p}',\mathbf{p}'')} \phi_z(0) \qquad (B.6)$$

with suitably chosen complex coefficients $b_{\mathbf{p}',\mathbf{p}''} \in \mathbb{C}$ depending only on $\mathbf{p}'$, $\mathbf{p}''$, $\mathbf{m}$ and $\boldsymbol{\ell}$. Here we used that $|\mathbf{m}|, |\boldsymbol{\ell}| \le mn$. Since $\mathcal{P}' \subseteq \mathcal{P}_m^n$ is bounded and $p \in \mathcal{P}'$, we can write

$$p(z) = \sum_{\substack{\mathbf{m},\boldsymbol{\ell}\in\mathbb{N}_0^n \\ \mathbf{m},\boldsymbol{\ell}\le m}} a_{\mathbf{m},\boldsymbol{\ell}} z^{\mathbf{m}} \overline{z}^{\boldsymbol{\ell}}$$

with $|a_{\mathbf{m},\boldsymbol{\ell}}| \leq c$ for some constant $c = c(\mathcal{P}') > 0$. In combination with (B.6), this easily implies that we can rewrite $p$ as

$$p(z) = \sum_{\substack{\mathbf{p} \in \mathbb{N}_0^{2n} \\ \mathbf{p} \leq 2m}} c_{\mathbf{p}}(p) \partial^{\mathbf{p}} \phi_z(0) \tag{B.7}$$

with coefficients $c_{\mathbf{p}}(p) \in \mathbb{C}$ satisfying $|c_{\mathbf{p}}(p)| \leq c'$ for some constant $c' = c'(\phi, b, \mathcal{P}', m, n)$. By Lemma B.7, we choose $h^* > 0$, such that

$$|\Phi_{\mathbf{p},h^*}(z) - \partial^{\mathbf{p}} \phi_z(0)| \leq \frac{\varepsilon}{\sum_{\mathbf{q} \in \mathbb{N}_0^{2n}, \mathbf{q} \leq 2m} c'}$$

for all $z \in \Omega_n$ and $\mathbf{p} \in \mathbb{N}_0^{2n}$ with $\mathbf{p} \leq 2m$. Furthermore, we can rewrite each function $\Phi_{\mathbf{p},h^*}$ as

$$\Phi_{\mathbf{p},h^*}(z) = \sum_{\substack{\boldsymbol{\alpha} \in \mathbb{Z}^{2n} \\ |\boldsymbol{\alpha}_j| \leq 2m \; \forall j}} \lambda_{\boldsymbol{\alpha},\mathbf{p}} \phi([\varphi_n(h^* \boldsymbol{\alpha})]^T \cdot z + b)$$

with suitable coefficients $\lambda_{\boldsymbol{\alpha},\mathbf{p}} \in \mathbb{C}$. Since the cardinality of the set

$$\{\boldsymbol{\alpha} \in \mathbb{Z}^{2n} : |\boldsymbol{\alpha}_j| \leq 2m \; \forall j\}$$

is $(4m+1)^{2n}$, this can be converted to

$$\Phi_{\mathbf{p},h^*}(z) = \sum_{j=1}^{N} \lambda_{j,\mathbf{p}} \phi\left(\rho_j^T \cdot z + b\right).$$

For $p$ as in (B.7), we then define

$$\theta(z) := \sum_{\substack{\mathbf{p} \in \mathbb{N}_0^{2n} \\ \mathbf{p} \leq 2m}} c_{\mathbf{p}}(p) \cdot \Phi_{\mathbf{p},h^*}(z) = \sum_{j=1}^{N} \left[ \left( \sum_{\substack{\mathbf{p} \in \mathbb{N}_0^{2n} \\ \mathbf{p} \leq 2m}} c_{\mathbf{p}}(p) \lambda_{j,\mathbf{p}} \right) \phi(\rho_j^T \cdot z + b) \right]$$

and note

$$|\theta(z) - p(z)| \leq \sum_{\mathbf{p} \leq 2m} |c_{\mathbf{p}}(p)| \cdot |\Phi_{\mathbf{p},h^*}(z) - \partial^{\mathbf{p}} \phi_z(0)| \leq \varepsilon.$$

Since the coefficients $\rho_j$ have been chosen independently of the polynomial $p$, we can rewrite $\theta$ in the desired form. $\qquad\square$

## C  Postponed proofs for the approximation of $C^k$-functions

### C.1  Prerequisites from Fourier Analysis

This section is dedicated to reviewing some notations and results from Fourier Analysis. In the end, a quantitative result for the approximation of $C^k\left([-1,1]^s; \mathbb{R}\right)$-functions using linear combinations of multivariate Chebyshev polynomials is derived; see Theorem C.15.

We start by recalling several notations and concepts from Fourier Analysis.

**Definition C.1.** Let $s \in \mathbb{N}$ and $k \in \mathbb{N}_0$. We define

$$C_{2\pi}^k\left(\mathbb{R}^s; \mathbb{C}\right) := \left\{ f \in C^k\left(\mathbb{R}^s; \mathbb{C}\right) : \forall \mathbf{p} \in \mathbb{Z}^s \; \forall x \in \mathbb{R}^s : \; f(x + 2\pi\mathbf{p}) = f(x) \right\}.$$

and $C_{2\pi}\left(\mathbb{R}^s; \mathbb{C}\right) := C_{2\pi}^0\left(\mathbb{R}^s; \mathbb{C}\right)$. For a function $f \in C_{2\pi}^k\left(\mathbb{R}^s; \mathbb{C}\right)$ we write

$$\|f\|_{C^k([-\pi,\pi]^s;\mathbb{C})} := \max_{\substack{\mathbf{k} \in \mathbb{N}_0^s \\ |\mathbf{k}| \leq k}} \left\|\partial^{\mathbf{k}} f\right\|_{L^\infty([-\pi,\pi]^s;\mathbb{C})} \quad \text{and}$$

$$\|f\|_{L^p([-\pi,\pi]^s;\mathbb{C})} := \left( \frac{1}{(2\pi)^s} \cdot \int_{[-\pi,\pi]^s} |f(x)|^p \, dx \right)^{1/p} \quad \text{for } p \in [1,\infty).$$

Moreover, we set $\|f\|_{L^\infty([-\pi,\pi]^s;\mathbb{R})} := \|f\|_{C^0([-\pi,\pi]^s;\mathbb{C})}$.

**Definition C.2.** For any $s \in \mathbb{N}$ and $\mathbf{k} \in \mathbb{Z}^s$, we write

$$e_{\mathbf{k}}: \quad \mathbb{R}^s \to \mathbb{C}, \quad e_{\mathbf{k}}(x) = e^{i\langle \mathbf{k}, x \rangle}$$

where $\langle \cdot, \cdot \rangle$ denotes the usual inner product of two vectors. For any $f \in C_{2\pi}(\mathbb{R}^s; \mathbb{C})$ we define the $\mathbf{k}$-th Fourier coefficient of $f$ to be

$$\hat{f}(\mathbf{k}) := \frac{1}{(2\pi)^s} \int_{[-\pi,\pi]^s} f(x) \overline{e_{\mathbf{k}}(x)} dx.$$

**Definition C.3.** For two functions $f, g \in C_{2\pi}(\mathbb{R}^s; \mathbb{C})$, we define their convolution as

$$f * g: \quad \mathbb{R}^s \to \mathbb{C}, \quad (f * g)(x) := \frac{1}{(2\pi)^s} \int_{[-\pi,\pi]^s} f(t)g(x-t)dt.$$

In the following we define several so-called kernels.

**Definition C.4.** Let $m \in \mathbb{N}_0$ be arbitrary.

1. The $m$-th one-dimensional *Dirichlet-kernel* is defined as

$$D_m := \sum_{h=-m}^{m} e_h.$$

2. The $m$-th one-dimensional *Fejèr-kernel* is defined as

$$F_m := \frac{1}{m} \sum_{h=0}^{m-1} D_h.$$

3. The $m$-th one-dimensional *de-la-Vallée-Poussin-kernel* is defined as

$$V_m := (1 + e_m + e_{-m}) \cdot F_m.$$

4. Let $s \in \mathbb{N}$. We extend the above definitions to dimension $s$ by letting

$$D_m^s(x_1, ..., x_s) := \prod_{p=1}^{s} D_m(x_p),$$

$$F_m^s(x_1, ..., x_s) := \prod_{p=1}^{s} F_m(x_p),$$

$$V_m^s(x_1, ..., x_s) := \prod_{p=1}^{s} V_m(x_p).$$

We need the following property of the multivariate extension of the de-la-Vallée-Poussin-kernel.

**Proposition C.5.** *Let $m, s \in \mathbb{N}$. Then one has* $\|V_m^s\|_{L^1([-\pi,\pi]^s; \mathbb{C})} \leq 3^s$.

*Proof.* From [34, Exercise 1.3 and Lemma 1.4] it follows $\|F_m\|_{L^1([-\pi,\pi]; \mathbb{C})} = 1$ and hence using the triangle inequality $\|V_m\|_{L^1([-\pi,\pi]; \mathbb{C})} \leq 3$. The claim then follows using Tonelli's theorem. $\qquad\square$

The following definition introduces the term of trigonometric polynomial.

**Definition C.6.** For any $s \in \mathbb{N}$ and $m \in \mathbb{N}_0$ we call a function of the form

$$\mathbb{R}^s \to \mathbb{C}, \quad x \mapsto \sum_{\substack{\mathbf{k} \in \mathbb{Z}_0^s \\ -m \leq \mathbf{k} \leq m}} a_{\mathbf{k}} e^{i\langle \mathbf{k}, x \rangle}$$

with coefficients $a_{\mathbf{k}} \in \mathbb{C}$ a *trigonometric polynomial of coordinatewise degree at most $m$* and denote the space of all those functions with $\mathbb{H}_m^s$. Here, we consider the sum over all $\mathbf{k} \in \mathbb{Z}^s$ with $-m \leq \mathbf{k}_j \leq m$ for all $j \in \{1, ..., s\}$. We then write

$$E_m^s(f) := \min_{T \in \mathbb{H}_m^s} \|f - T\|_{L^\infty(\mathbb{R}^s; \mathbb{C})} \tag{C.1}$$

for any function $f \in C_{2\pi}(\mathbb{R}^s; \mathbb{C})$.

The following proposition shows that convolving with the Fejèr kernel produces a trigonometric polynomial of degree at most $2m - 1$, while reproducing trigonometric polynomials of degree $m$. Furthermore, the norm of the convolution operator is bounded uniformly in $m$. These properties will be useful for our proof of Theorem C.15.

**Proposition C.7.** *Let $s, m \in \mathbb{N}$ and $k \in \mathbb{N}_0$. The map*

$$v_m: \quad C_{2\pi}(\mathbb{R}^s; \mathbb{C}) \to \mathbb{H}_{2m-1}^s, \quad f \mapsto f * V_m^s$$

*is well-defined and satisfies*

$$v_m(T) = T \quad \text{for all} \quad T \in \mathbb{H}_m^s. \tag{C.2}$$

*Furthermore, there exists a constant $c = c(s) > 0$ (independent of $m$), such that*

$$\|v_m(f)\|_{C^k([-\pi,\pi]^s;\mathbb{C})} \leq c \cdot \|f\|_{C^k([-\pi,\pi]^s;\mathbb{C})} \ \forall f \in C_{2\pi}^k(\mathbb{R}^s;\mathbb{C}),$$

$$\|v_m(f)\|_{L^\infty([\pi,\pi]^s;\mathbb{C})} \leq c \cdot \|f\|_{L^\infty([-\pi,\pi]^s;\mathbb{C})} \ \forall f \in C_{2\pi}(\mathbb{R}^s;\mathbb{C}). \tag{C.3}$$

*In fact, it holds $c(s) \leq \exp(C \cdot s)$ with an absolute constant $C > 0$.*

*Proof.* A direct computation shows that $f * e_{\mathbf{k}} = \hat{f}(\mathbf{k}) \cdot e_{\mathbf{k}}$. This implies that $v_m$ is well-defined since $V_m^s$ is a trigonometric polynomial of coordinatewise degree at most $2m - 1$.

The operator is bounded on $C_{2\pi}^k(\mathbb{R}^s; \mathbb{C})$ and $C_{2\pi}(\mathbb{R}^s; \mathbb{C})$ with norm at most $c = 3^s$, as follows from Young's inequality [34, Lemma 1.1 (ii)], Proposition C.5, and the fact that one has for all $\mathbf{k} \in \mathbb{N}_0^s$ with $|\mathbf{k}| \leq k$ the identity

$$\partial^{\mathbf{k}} (f * V_m^s) = (\partial^{\mathbf{k}} f) * V_m^s \quad \text{for } f \in C_{2\pi}^k(\mathbb{R}^s; \mathbb{C}).$$

It remains to show that $v_m$ is the identity on $\mathbb{H}_m^s$. We first prove that

$$e_k * V_m = e_k \tag{C.4}$$

holds for all $k \in \mathbb{Z}$ with $|k| \leq m$. First note that

$$e_k * V_m = e_k * F_m + e_k * (e_m \cdot F_m) + e_k * (e_{-m} \cdot F_m).$$

We then compute

$$e_k * F_m = \frac{1}{m} \sum_{\ell=0}^{m-1} D_\ell * e_k = \frac{1}{m} \sum_{\ell=0}^{m-1} \sum_{h=-\ell}^{\ell} \underbrace{e_h * e_k}_{=\delta_{k,h} \cdot e_k} = \frac{1}{m} \sum_{\ell=|k|}^{m-1} e_k = \frac{m - |k|}{m} \cdot e_k.$$

Similarly, we get

$$e_k * (e_m \cdot F_m) = \frac{1}{m} \sum_{\ell=0}^{m-1} (e_m D_\ell) * e_k = \frac{1}{m} \sum_{\ell=0}^{m-1} \sum_{h=-\ell}^{\ell} \underbrace{e_{h+m} * e_k}_{=\delta_{k,h+m} \cdot e_k}$$

$$= \frac{1}{m} \sum_{\substack{0 \leq \ell \leq m-1 \\ \ell \geq m-k}} e_k = \delta_{k \geq 1} \cdot \frac{k}{m} \cdot e_k$$

and

$$e_k * (e_{-m} \cdot F_m) = \frac{1}{m} \sum_{\ell=0}^{m-1} (e_{-m} D_\ell) * e_k = \frac{1}{m} \sum_{\ell=0}^{m-1} \sum_{h=-\ell}^{\ell} \underbrace{e_{h-m} * e_k}_{=\delta_{k,h-m} \cdot e_k}$$

$$= \frac{1}{m} \sum_{\substack{0 \leq \ell \leq m-1 \\ \ell \geq k+m}} e_k = \delta_{k \leq -1} \cdot \frac{-k}{m} \cdot e_k.$$

Adding up those three identities yields (C.4).

To finally prove (C.2), it clearly suffices to show that

$$e_{\mathbf{k}} * V_m^s = e_{\mathbf{k}}$$

for all $\mathbf{k} \in \mathbb{Z}^s$ with $-m \leq \mathbf{k} \leq m$. But for such $\mathbf{k}$, using $e_{\mathbf{k}}(x) = \prod_{j=1}^{s} e_{\mathbf{k}_j}(x_j)$, one obtains

$$(e_{\mathbf{k}} * V_m^s)(x) = \frac{1}{(2\pi)^s} \int_{[-\pi,\pi]^s} \prod_{j=1}^{s} e_{\mathbf{k}_j}(t_j) \cdot V_m(x_j - t_j)\, dt$$

$$\stackrel{\text{Fubini}}{=} \prod_{j=1}^{s} (e_{\mathbf{k}_j} * V_m)(x_j) \stackrel{\text{(C.4)}}{=} \prod_{j=1}^{s} e_{\mathbf{k}_j}(x_j) = e_{\mathbf{k}}(x)$$

for any $x \in \mathbb{R}^s$, as was to be shown. $\qquad\square$

The following result follows from a theorem in [29].

**Proposition C.8.** *Let $s, k \in \mathbb{N}$. Then there exists a constant $c = c(s,k) > 0$, such that, for $E_m^s$ as defined in* (C.1),

$$E_m^s(f) \leq \frac{c}{m^k} \cdot \|f\|_{C^k([-\pi,\pi]^s;\mathbb{R})}$$

*for all $m \in \mathbb{N}$ and $f \in C_{2\pi}^k(\mathbb{R}^s;\mathbb{R})$.*
*In fact, it holds $c(s,k) \leq \exp(C \cdot ks) \cdot k^k$ with an absolute constant $C > 0$.*

*Proof.* We apply [29, Theorem 6.6] with $n_i = m$ and $p_i = k$, which yields the existence of a constant $c_1 = c_1(s,k) > 0$, such that

$$E_m^s(f) \leq c_1 \cdot \sum_{\ell=1}^{s} \frac{1}{m^k} \cdot \omega_\ell\left(f, \frac{1}{m}\right)$$

for all $m \in \mathbb{N}$ and $f \in C_{2\pi}^k(\mathbb{R}^s;\mathbb{R})$, where $\omega_\ell(f, \bullet)$ denotes the modulus of continuity of $\frac{\partial^k f}{\partial x_\ell^k}$ with respect to $x_\ell$, where we have the trivial bound

$$\omega_\ell\left(f, \frac{1}{m}\right) \leq 2 \cdot \|f\|_{C^k([-\pi,\pi]^s;\mathbb{R})}.$$

Hence, we get

$$E_m^s(f) \leq c_1 \cdot s \cdot 2 \cdot \|f\|_{C^k([-\pi,\pi]^s;\mathbb{R})} \frac{1}{m^k},$$

so the claim follows by choosing $c := 2s \cdot c_1$.

We refer to Appendix C.2 (see Theorem C.18) for a proof of the claimed bound on the constant $c(s,k)$. $\qquad\square$

The above proposition bounds the best possible error of approximating $f$ by trigonometric polynomials of coordinatewise degree at most $m$, but this is in general non-constructive. Our next result shows that a similar bound holds for approximating $f$ by $v_m(f)$.

**Theorem C.9.** *Let $s \in \mathbb{N}$. Then there exists a constant $c = c(s) > 0$, such that the operator $v_m$ from Proposition C.7 satisfies*

$$\|f - v_m(f)\|_{L^\infty(\mathbb{R}^s)} \leq c \cdot E_m^s(f)$$

*for any $m \in \mathbb{N}$ and $f \in C_{2\pi}(\mathbb{R}^s;\mathbb{C})$.*
*In fact, it holds $c(s) \leq \exp(C \cdot s)$ with an absolute constant $C > 0$.*

*Proof.* For any $T \in \mathbb{H}_m^s$ one has

$$\|f - v_m(f)\|_{L^\infty(\mathbb{R}^s)} \stackrel{\text{(C.2)}}{\leq} \|f - T\|_{L^\infty(\mathbb{R}^s)} + \|v_m(T) - v_m(f)\|_{L^\infty(\mathbb{R}^s)} \stackrel{\text{(C.3)}}{\leq} (c+1)\|f - T\|_{L^\infty(\mathbb{R}^s)}.$$

Taking the infimum over all $T \in \mathbb{H}_m^s$ yields the claim. $\qquad\square$

By combining Proposition C.8 and Theorem C.9, we immediately get the following bound.

**Corollary C.10.** *Let $s, k \in \mathbb{N}_0$. Then there exists a constant $c = c(s, k) > 0$, such that*

$$\|f - v_m(f)\|_{L^\infty(\mathbb{R}^s)} \le \frac{c}{m^k} \cdot \|f\|_{C^k([-\pi,\pi]^s;\mathbb{R})}$$

*for every $m \in \mathbb{N}$ and $f \in C^k_{2\pi}(\mathbb{R}^s; \mathbb{R})$.*
*In fact, we have $c(s, k) \le \exp(C \cdot ks) \cdot k^k$ with an absolute constant $C > 0$.*

Up to now, we have studied the approximation of periodic functions by trigonometric polynomials, but our actual goal is to approximate non-periodic functions by algebraic polynomials. The next lemma establishes a connection between the two settings.

**Lemma C.11.** *Let $k \in \mathbb{N}_0$ and $s \in \mathbb{N}$. For any function $f \in C^k([-1, 1]^s; \mathbb{C})$, we define the corresponding periodic function via*

$$f^* : \quad \mathbb{R}^s \to \mathbb{C}, \quad f^*(x_1, ..., x_s) = f(\cos(x_1), ..., \cos(x_s))$$

*and note $f^* \in C^k_{2\pi}(\mathbb{R}^s; \mathbb{C})$. The map*

$$C^k([-1, 1]^s; \mathbb{C}) \to C^k_{2\pi}(\mathbb{R}^s; \mathbb{C}), \quad f \mapsto f^*$$

*is a continuous linear operator with respect to the $C^k$-norms on $C^k([-1, 1]^s; \mathbb{C})$ and $C^k_{2\pi}(\mathbb{R}^s; \mathbb{C})$. The operator norm can be bounded from above by $k^k$.*

*Proof.* The map is well-defined since $\cos$ is a smooth function and $2\pi$-periodic. The linearity of the operator is obvious, so it remains to show its continuity.

The goal is to apply the closed graph theorem [21, Theorem 5.12]. By definition of $f^*$, and since $\cos : [-\pi, \pi] \to [-1, 1]$ is surjective, we have the equality $\|f\|_{L^\infty([-1,1]^s;\mathbb{C})} = \|f^*\|_{L^\infty([-\pi,\pi]^s;\mathbb{C})}$. Let then $(f_n)_{n\in\mathbb{N}}$ be a sequence of functions $f_n \in C^k([-1, 1]^s; \mathbb{C})$ and $g^* \in C^k_{2\pi}(\mathbb{R}^s; \mathbb{C})$ such that $f_n \to f$ in $C^k([-1, 1]^s; \mathbb{C})$ and $f_n^* \to g^*$ in $C^k_{2\pi}(\mathbb{R}^s; \mathbb{C})$. We then have

$$
\begin{aligned}
\|f^* - g^*\|_{L^\infty([-\pi,\pi]^s)} &\le \|f^* - f_n^*\|_{L^\infty([-\pi,\pi]^s)} + \|f_n^* - g^*\|_{L^\infty([-\pi,\pi]^s)} \\
&= \|f - f_n\|_{L^\infty([-1,1]^s;\mathbb{C})} + \|f_n^* - g^*\|_{L^\infty([-\pi,\pi]^s)} \\
&\le \|f - f_n\|_{C^k([-1,1]^s;\mathbb{C})} + \|f_n^* - g^*\|_{C^k([-\pi,\pi]^s;\mathbb{C})} \to 0 \ (n \to \infty).
\end{aligned}
$$

It follows $f^* = g^*$ and the closed graph theorem yields the desired continuity.

We refer to Appendix C.2 (see Theorem C.19 and Remark C.20) for a proof of the claimed bound on the operator norm. $\qquad \square$

For a function $f \in C^k([-1, 1]^s; \mathbb{C})$ we want to express $v_m(f^*)$ in a convenient way, involving a product of cosines. To this end, we make use of the following identity, which is a generalization of the well-known product-to-sum formula for $\cos$.

**Lemma C.12.** *Let $s \in \mathbb{N}$. Then it holds for any $x \in \mathbb{R}^s$ that*

$$\prod_{j=1}^s \cos(x_j) = \frac{1}{2^s} \sum_{\sigma \in \{-1,1\}^s} \cos(\langle \sigma, x \rangle).$$

*Proof.* This is an inductive generalization of the product-to-sum formula

$$2\cos(x)\cos(y) = \cos(x - y) + \cos(x + y) \tag{C.5}$$

for $x, y \in \mathbb{R}$, which can be found for instance in [1, Eq. 4.3.32]. The case $s = 1$ holds since $\cos$ is an even function. Assume that the claim holds for a fixed $s \in \mathbb{N}$ and take $x \in \mathbb{R}^{s+1}$. Writing

$x' = (x_1, ..., x_s)$, we derive

$$\prod_{j=1}^{s+1} \cos(x_j) = \left( \frac{1}{2^s} \sum_{\sigma \in \{-1,1\}^s} \cos(\langle \sigma, x' \rangle) \right) \cdot \cos(x_{s+1})$$

$$= \frac{1}{2^s} \sum_{\sigma \in \{-1,1\}^s} \cos(\langle \sigma, x' \rangle) \cos(x_{s+1})$$

$$\stackrel{(C.5)}{=} \frac{1}{2^{s+1}} \sum_{\sigma \in \{-1,1\}^s} [\cos(\langle \sigma, x' \rangle + x_{s+1}) + \cos(\langle \sigma, x' \rangle - x_{s+1})]$$

$$= \frac{1}{2^{s+1}} \sum_{\sigma \in \{-1,1\}^{s+1}} \cos(\langle \sigma, x \rangle),$$

as was to be shown. $\qquad\qquad\qquad\square$

The following proposition states that $v_m(f^*)$ can be expressed as a linear combination of products of cosines. This representation is useful since these cosines can be interpolated by Chebyshev polynomials which in the end leads to the desired approximation result.

**Proposition C.13.** *Let $s \in \mathbb{N}$ and $k \in \mathbb{N}_0$. For any $f \in C^k([-1,1]^s; \mathbb{C})$ and $m \in \mathbb{N}$ the de-la-Vallée-Poussin operator given as $f \mapsto v_m(f^*)$ with $v_m$ as in Proposition C.7 and $f \mapsto f^*$ as in Lemma C.11 has a representation*

$$v_m(f^*)(x_1, ..., x_s) = \sum_{\substack{\mathbf{k} \in \mathbb{N}_0^s \\ \mathbf{k} \leq 2m-1}} \mathcal{V}_{\mathbf{k}}^m(f) \prod_{j=1}^{s} \cos(\mathbf{k}_j x_j)$$

*for continuous linear functionals*

$$\mathcal{V}_{\mathbf{k}}^m : \ C^k([-1,1]^s; \mathbb{C}) \to \mathbb{C}, \quad f \mapsto 2^{\|\mathbf{k}\|_0} \cdot a_{\mathbf{k}}^m \cdot \widehat{f^*}(\mathbf{k}),$$

*where $\|\mathbf{k}\|_0 = \#\{j \in \{1, ..., s\} : \mathbf{k}_j \neq 0\}$ and $a_{\mathbf{k}}^m = \widehat{V_m^s}(\mathbf{k})$. Furthermore, if $f \in C^k([-1,1]^s; \mathbb{R})$, then $\mathcal{V}_{\mathbf{k}}^m(f) \in \mathbb{R}$ for every $\mathbf{k} \in \mathbb{N}_0^s$ with $\mathbf{k} \leq 2m-1$.*

*Proof.* First of all, it is easy to see that $v_m(f^*)$ is even in each variable, which follows directly from the fact that $f^*$ and $V_m^s$ are both even in each variable. Furthermore, if we write

$$V_m^s = \sum_{\substack{\mathbf{k} \in \mathbb{Z}^s \\ -(2m-1) \leq \mathbf{k} \leq 2m-1}} a_{\mathbf{k}}^m e_{\mathbf{k}}$$

with appropriately chosen coefficients $a_{\mathbf{k}}^m \in \mathbb{R}$, we easily see

$$v_m(f^*) = \sum_{\substack{\mathbf{k} \in \mathbb{Z}^s \\ -(2m-1) \leq \mathbf{k} \leq 2m-1}} a_{\mathbf{k}}^m \widehat{f^*}(\mathbf{k}) e_{\mathbf{k}}.$$

Using Euler's identity and the fact that $v_m(f^*)$ is an even function, we get the representation

$$v_m(f^*)(x) = \sum_{\substack{\mathbf{k} \in \mathbb{Z}^s \\ -(2m-1) \leq \mathbf{k} \leq 2m-1}} a_{\mathbf{k}}^m \widehat{f^*}(\mathbf{k}) \cos(\langle \mathbf{k}, x \rangle)$$

for all $x \in \mathbb{R}^s$. Using $\odot$ to denote the componentwise product of two vectors of the same size, i.e., $x \odot y = (x_i \cdot y_i)_i$, and using the identity $\langle \mathbf{k}, \sigma \odot x \rangle = \langle \sigma, \mathbf{k} \odot x \rangle$, we see since $v_m(f^*)$ is even in

every variable that

$$v_m(f^*)(x) = \frac{1}{2^s} \cdot \sum_{\sigma \in \{-1,1\}^s} v_m(f^*)(\sigma \odot x)$$

$$= \frac{1}{2^s} \cdot \sum_{\substack{\sigma \in \{-1,1\}^s}} \sum_{\substack{\mathbf{k} \in \mathbb{Z}^s \\ -(2m-1) \leq \mathbf{k} \leq 2m-1}} a_{\mathbf{k}}^m \widehat{f^*}(\mathbf{k}) \cos(\langle \mathbf{k}, \sigma \odot x \rangle)$$

$$= \sum_{\substack{\mathbf{k} \in \mathbb{Z}^s \\ -(2m-1) \leq \mathbf{k} \leq 2m-1}} \left( a_{\mathbf{k}}^m \widehat{f^*}(\mathbf{k}) \frac{1}{2^s} \sum_{\sigma \in \{-1,1\}^s} \cos(\langle \sigma, \mathbf{k} \odot x \rangle) \right)$$

$$\overset{\text{Lemma C.12}}{=} \sum_{\substack{\mathbf{k} \in \mathbb{Z}^s \\ -(2m-1) \leq \mathbf{k} \leq 2m-1}} a_{\mathbf{k}}^m \widehat{f^*}(\mathbf{k}) \prod_{j=1}^s \cos(\mathbf{k}_j x_j)$$

$$= \sum_{\substack{\mathbf{k} \in \mathbb{N}_0^s \\ \mathbf{k} \leq 2m-1}} 2^{\|\mathbf{k}\|_0} a_{\mathbf{k}}^m \widehat{f^*}(\mathbf{k}) \prod_{j=1}^s \cos(\mathbf{k}_j x_j)$$

with

$$\|\mathbf{k}\|_0 := \#\{j \in \{1, ..., s\} : \mathbf{k}_j \neq 0\}.$$

In the last step we again used that cos is an even function and that

$$\widehat{f^*}(\mathbf{k}) = \widehat{f^*}(\sigma \odot \mathbf{k})$$

for all $\sigma \in \{-1,1\}^s$, which also follows easily since $f^*$ and $V_m^s$ are even in every component. Letting

$$\mathcal{V}_{\mathbf{k}}^m(f) := 2^{\|\mathbf{k}\|_0} a_{\mathbf{k}}^m \widehat{f^*}(\mathbf{k}),$$

we have the desired form. The fact that $\mathcal{V}_{\mathbf{k}}^m$ is a continuous linear functional on $C_{2\pi}^k([-1,1]^s;\mathbb{C})$ follows directly since $f \mapsto \widehat{f^*}(\mathbf{k})$ is a continuous linear functional for every $\mathbf{k}$. If $f$ is real-valued, so is $\widehat{f^*}(\mathbf{k})$ for every $\mathbf{k} \in \mathbb{N}_0^s$ with $\mathbf{k} \leq 2m-1$, since $f^*$ is real-valued and even in every component. $\qquad\square$

Our main approximation result involves linear combinations of Chebyshev polynomials where the coefficients in this linear combination are given as $\mathcal{V}_{\mathbf{k}}^m(f)$. It is therefore important to be able to bound the sum of the absolute values $|\mathcal{V}_{\mathbf{k}}^m(f)|$.

**Lemma C.14.** *Let $s \in \mathbb{N}$. There exists a constant $c = c(s) > 0$, such that the inequality*

$$\sum_{\substack{\boldsymbol{k} \in \mathbb{N}_0^s \\ \boldsymbol{k} \leq 2m-1}} |\mathcal{V}_{\boldsymbol{k}}^m(f)| \leq c \cdot m^{s/2} \cdot \|f\|_{L^\infty([-1,1]^s;\mathbb{C})}$$

*holds for all $m \in \mathbb{N}$ and $f \in C([-1,1]^s;\mathbb{C})$, where $\mathcal{V}_{\boldsymbol{k}}^m$ is as in Proposition C.13.*
*In fact, we have $c(s) \leq \exp(C \cdot s)$ with an absolute constant $C > 0$.*

*Proof.* Let $f \in C([-1,1]^s;\mathbb{C})$ and $m \in \mathbb{N}$. For any multi-index $\boldsymbol{\ell} \in \mathbb{N}_0^s$, it follows from Proposition C.13 that

$$\widehat{v_m(f^*)}(\boldsymbol{\ell}) = \sum_{\substack{\mathbf{k} \in \mathbb{N}_0^s \\ \mathbf{k} \leq 2m-1}} \mathcal{V}_{\mathbf{k}}^m(f) \widehat{g_{\mathbf{k}}}(\boldsymbol{\ell}),$$

with

$$g_{\mathbf{k}} : \quad \mathbb{R}^s \to \mathbb{R}, \quad (x_1, ..., x_s) \mapsto \prod_{j=1}^s \cos(\mathbf{k}_j x_j).$$

Now, a calculation using Fubini's theorem and using

$$g_{\mathbf{k}} = \prod_{j=1}^s \frac{1}{2}(e_{\mathbf{k}_j} + e_{-\mathbf{k}_j}) = \prod_{\substack{1 \leq j \leq s \\ \mathbf{k}_j \neq 0}} \frac{1}{2}(e_{\mathbf{k}_j} + e_{-\mathbf{k}_j})$$

for any number $k \in \mathbb{N}_0$ shows

$$\widehat{g_{\mathbf{k}}}(\boldsymbol{\ell}) = \begin{cases} \frac{1}{2^{\|\mathbf{k}\|_0}}, & \mathbf{k} = \boldsymbol{\ell}, \\ 0, & \text{otherwise} \end{cases} \quad \text{for } \mathbf{k}, \boldsymbol{\ell} \in \mathbb{N}_0^s.$$

Therefore, we have the bound $|\mathcal{V}_{\boldsymbol{\ell}}^m(f)| \leq 2^s \cdot \left| \widehat{v_m(f^*)}(\boldsymbol{\ell}) \right|$ for $\boldsymbol{\ell} \in \mathbb{N}_0^s$ with $|\boldsymbol{\ell}| \leq 2m - 1$. Using the Cauchy-Schwarz and the Parseval inequality, we therefore see

$$\sum_{\substack{\mathbf{k} \in \mathbb{N}_0^s \\ \mathbf{k} \leq 2m-1}} |\mathcal{V}_{\mathbf{k}}^m(f)| \leq 2^s \cdot \sum_{\substack{\mathbf{k} \in \mathbb{N}_0^s \\ \mathbf{k} \leq 2m-1}} \left| \widehat{v_m(f^*)}(\mathbf{k}) \right| \overset{\text{CS}}{\leq} 2^s \cdot (2m)^{s/2} \cdot \left( \sum_{\substack{\mathbf{k} \in \mathbb{N}_0^s \\ \mathbf{k} \leq 2m-1}} \left| \widehat{v_m(f^*)}(\mathbf{k}) \right|^2 \right)^{1/2}$$

$$\overset{\text{Parseval}}{\leq} 2^s \cdot 2^{s/2} \cdot m^{s/2} \cdot \|v_m(f^*)\|_{L^2([-\pi,\pi]^s;\mathbb{C})}$$

$$\leq \underbrace{2^s \cdot 2^{s/2}}_{=:c_1(s)} \cdot m^{s/2} \cdot \|v_m(f^*)\|_{L^\infty([-\pi,\pi]^s;\mathbb{C})} \, .$$

Using Proposition C.7, we get a constant $c_2(s) \leq \exp(C_0 \cdot s)$ such that

$$\sum_{\substack{\mathbf{k} \in \mathbb{N}_0^s \\ \mathbf{k} \leq 2m-1}} |\mathcal{V}_{\mathbf{k}}^m(f)| \leq c_1(s) \cdot c_2(s) \cdot m^{s/2} \cdot \|f^*\|_{L^\infty([-\pi,\pi]^s;\mathbb{C})} = c(s) \cdot m^{s/2} \cdot \|f\|_{L^\infty([-1,1]^s;\mathbb{C})} \, ,$$

as claimed. $\qquad \square$

For any natural number $\ell \in \mathbb{N}_0$, we denote by $T_\ell$ the $\ell$-th *Chebyshev polynomial*, satisfying

$$T_\ell(\cos(x)) = \cos(\ell x), \quad x \in \mathbb{R}.$$

One can show that $T_\ell$ is in fact a polynomial of degree $\ell$. For a multi-index $\mathbf{k} \in \mathbb{N}_0^s$, we define

$$T_{\mathbf{k}}(x) := \prod_{j=1}^s T_{\mathbf{k}_j}(x_j), \quad x \in \mathbb{R}^s.$$

We then get the following approximation result about approximating (non-periodic) $C^k$-functions by linear combinations of Chebyshev polynomials.

**Theorem C.15.** *Let $k, s, m \in \mathbb{N}$. Then there exists a constant $c = c(s, k) > 0$ with the following property: For any $f \in C^k([-1,1]^s; \mathbb{R})$ the polynomial $P$ defined as*

$$P(x) := \sum_{\substack{\mathbf{k} \in \mathbb{N}_0^s \\ \mathbf{k} \leq 2m-1}} \mathcal{V}_{\mathbf{k}}^m(f) \cdot T_{\mathbf{k}}(x),$$

*with $\mathcal{V}_{\mathbf{k}}^m$ as in Proposition C.13, satisfies*

$$\|f - P\|_{L^\infty([-1,1]^s;\mathbb{R})} \leq \frac{c}{m^k} \cdot \|f\|_{C^k([-1,1]^s;\mathbb{R})} \, .$$

*Here, the maps*

$$C([-1,1]^s; \mathbb{R}) \to \mathbb{R}, \quad f \mapsto \mathcal{V}_{\mathbf{k}}^m(f)$$

*are continuous and linear functionals with respect to the $L^\infty$-norm. Furthermore, there exists a constant $\tilde{c} = \tilde{c}(s) > 0$, such that the inequality*

$$\sum_{\substack{\mathbf{k} \in \mathbb{N}_0^s \\ \mathbf{k} \leq 2m-1}} |\mathcal{V}_{\mathbf{k}}^m(f)| \leq \tilde{c} \cdot m^{s/2} \cdot \|f\|_{L^\infty([-1,1]^s;\mathbb{R})}$$

*holds for all $f \in C([-1,1]^s; \mathbb{R})$.*
*Moreover, we have $c(s,k) \leq \exp(C \cdot ks) \cdot k^{2k}$ and $\tilde{c}(s) \leq \exp(C \cdot s)$ with an absolute constant $C > 0$.*

*Proof.* We choose the constant $c_0 = c_0(s, k)$ according to Corollary C.10. Let $f \in C^k([-1,1]^s; \mathbb{R})$ be arbitrary. Then we define the corresponding function $f^* \in C_{2\pi}^k(\mathbb{R}^s; \mathbb{R})$ as above. Let $P$ be defined as in the statement of the theorem. Then it follows from the definition of the Chebyshev polynomials $T_{\mathbf{k}}$, the definition of $P$, and the formula for $v_m(f^*)$ from Proposition C.13 that

$$P^*(x) = v_m(f^*)(x)$$

is satisfied, where $P^*$ is the corresponding function to $P$ defined similarly to $f^*$. Overall, we get the bound

$$\|f - P\|_{L^\infty([-1,1]^s; \mathbb{R})} = \|f^* - P^*\|_{L^\infty([-\pi,\pi]^s; \mathbb{R})} \overset{\text{Cor. C.10}}{\leq} \frac{c_0}{m^k} \cdot \|f^*\|_{C^k([-\pi,\pi]^s; \mathbb{R})}.$$

The first claim then follows using the continuity of the map $f \mapsto f^*$ as proven in Lemma C.11. The second part of the theorem has already been proven in Lemma C.14. $\qquad\square$

### C.2 Details on bounding the constant $c$ in Theorem 3.2

In this appendix we provide details on the bound of the constant $c$ that appears in the formulation of Theorem 3.2. Specifically, we perform a careful investigation of several results from [29] to get an upper bound for the constant appearing in Proposition C.8. Moreover, we analyze the operator norm of the operator

$$C^k([-1,1]^s; \mathbb{C}) \to C_{2\pi}^k(\mathbb{R}^s; \mathbb{C}), \quad f \mapsto f^* \quad \text{with} \quad f^*(x) := f(\cos(x_1), ..., \cos(x_s))$$

appearing in Lemma C.11 and show that it is bounded from above by $k^k$.

We start with the analysis of some bounds in [29, Chapter 4.3]. Here, a generalization of *Jackson's kernel* is defined for any $m, r \in \mathbb{N}$ as

$$L_{m,r}(t) := \lambda_{m,r}^{-1} \cdot \left( \frac{\sin(mt/2)}{\sin(t/2)} \right)^{2r}, \quad t \in \mathbb{R},$$

where $\lambda_{m,r}$ is chosen such that

$$\int_{[-\pi,\pi]} L_{m,r}(t) \, dt = 1.$$

The first two important bounds are provided in the following proposition.

**Proposition C.16.** *Let $m, r \in \mathbb{N}$. Then it holds*

$$\lambda_{m,r}^{-1} \leq \exp(C \cdot r) \cdot m^{1-2r} \quad \text{and} \quad \int_{[0,\pi]} t^k L_{m,r}(t) \, dt \leq \exp(C \cdot r) \cdot m^{-k}$$

*for any $k \leq 2r - 2$, with an absolute constant $C > 0$.*

*Proof.* Since $L_{m,r} \geq 0$ and since $\sin(t/2) \leq t/2$ for $t \in [0, \pi]$, we get

$$\lambda_{m,r} \geq \int_{[0,\pi]} \left( \frac{\sin(mt/2)}{t/2} \right)^{2r} dt = 2^{2r} \cdot \int_{[0,\pi]} \left( \frac{\sin(mt/2)}{t} \right)^{2r} dt$$

$$= 2^{2r} \cdot \int_{[0,\pi m/2]} \left( \frac{\sin(u)}{(2u)/m} \right)^{2r} du \cdot \frac{2}{m} \geq m^{2r-1} \cdot \int_{[0,\pi m/2]} \left( \frac{\sin(u)}{u} \right)^{2r} du$$

$$\geq m^{2r-1} \cdot \int_{[0,\pi/2]} \left( \frac{\sin(u)}{u} \right)^{2r} du \geq m^{2r-1} \cdot \int_{[0,\pi/2]} \left( \frac{2u}{\pi \cdot u} \right)^{2r} du \geq \left( \frac{2}{\pi} \right)^{2r} \cdot m^{2r-1}.$$

Here, we employed the inequality $\sin(u) \geq \frac{2}{\pi} u$ for $u \in [0, \pi/2]$ in the penultimate step.[1] This shows the first part of the claim.

---

[1] To see that this inequality holds, note that $\sin''(u) = -\sin(u) \leq 0$ for $u \in [0, \pi/2]$, so that sin is concave on that interval, and hence $\sin(u) = \sin((1 - \frac{2u}{\pi}) \cdot 0 + \frac{2u}{\pi} \cdot \frac{\pi}{2}) \geq \frac{2u}{\pi} \sin(\frac{\pi}{2}) = \frac{2u}{\pi}$.

For the second part, we again use the estimate $\sin(u) \geq \frac{2}{\pi}u$ for $u \in [0, \pi/2]$ to compute

$$\int_{[0,\pi]} t^k L_{m,r}(t)\, dt = \lambda_{m,r}^{-1} \cdot \int_{[0,\pi]} t^k \left(\frac{\sin(mt/2)}{\sin(t/2)}\right)^{2r} dt \leq \lambda_{m,r}^{-1} \cdot \int_{[0,\pi]} t^k \left(\frac{\sin(mt/2)}{t/\pi}\right)^{2r} dt$$

$$= \lambda_{m,r}^{-1} \cdot \pi^{2r} \cdot \int_{[0,\pi]} t^{k-2r} \sin(mt/2)^{2r}\, dt$$

$$= \lambda_{m,r}^{-1} \cdot \pi^{2r} \cdot \int_{[0,\pi m/2]} \left(\frac{2u}{m}\right)^{k-2r} \sin(u)^{2r}\, du \cdot \frac{2}{m}$$

$$\leq \lambda_{m,r}^{-1} \cdot \pi^{2r} \cdot m^{2r-1-k} \int_{[0,\pi m/2]} u^{k-2r} \sin(u)^{2r}\, du$$

$$\leq \exp(C_1 \cdot r) \cdot \int_{[0,\infty)} u^{k-2r} \cdot \sin(u)^{2r}\, du \cdot m^{-k}$$

with an absolute constant $C_1 > 0$. Here, we employed the first part of this proposition. It remains to bound the integral. This is done via

$$\int_{[0,\infty)} u^{k-2r} \cdot \sin(u)^{2r}\, du = \int_{[0,1]} u^{k-2r} \cdot \sin(u)^{2r}\, du + \int_{[1,\infty)} u^{k-2r} \cdot \sin(u)^{2r}\, du$$

$$\leq \int_{[0,1]} u^k \cdot \underbrace{\left(\frac{\sin(u)}{u}\right)^{2r}}_{\leq 1} du + \int_{[1,\infty)} u^{-2}\, du \leq C_2$$

with an absolute constant $C_2 > 0$. This proves the claim. $\qquad\square$

The proof in [29] proceeds by defining

$$K_{m,r}(t) := L_{m',r}(t), \quad m' = \left\lfloor \frac{m}{r} \right\rfloor + 1.$$

Proposition C.16 shows for $k \leq 2r - 2$ that

$$\int_{[0,\pi]} t^k K_{m,r}(t)\, dt \leq \exp(C \cdot r) \cdot (m')^{-k}.$$

Since $m' \geq \frac{m}{r}$ we infer

$$\int_{[0,\pi]} t^k K_{m,r}(t)\, dt \leq \exp(C \cdot r) \cdot \left(\frac{r}{m}\right)^k \leq \exp(C \cdot r) \cdot r^k \cdot m^{-k} \qquad\text{(C.6)}$$

with an absolute constant $C > 0$.

We can now quantify the constant appearing in [29, Theorem 4.3].

**Theorem C.17** (cf. [29, Theorem 4.3]). *Let $k, m \in \mathbb{N}$ and $f \in C_{2\pi}^k(\mathbb{R}; \mathbb{R})$. Let*

$$\omega(f^{(k)}, 1/m) := \max_{x \in \mathbb{R}, |t| \leq 1/m} |f^{(k)}(x+t) - f^{(k)}(x)|.$$

*Then it holds*

$$E_m^1(f) \leq (\exp(C \cdot k) \cdot k^k) \cdot m^{-k} \cdot \omega(f^{(k)}, 1/m).$$

*Here, we recall that $E_m^1(f)$ denotes the best possible approximation error when approximating $f$ using trigonometric polynomials of degree $m$; see Equation* (C.1).

*Proof.* We follow the proof of [29, Theorem 4.3]. Take $r = k + 1$ and define

$$I_m(x) := -\int_{[-\pi,\pi]} K_{m,r}(t) \sum_{\ell=1}^{k+1} (-1)^\ell \binom{k+1}{\ell} f(x + \ell t)\, dt.$$

Then it is shown in the proof of [29, Theorem 4.3] that $I_m$ is a trigonometric polynomial of degree at most $m$ and that

$$|f(x) - I_m(x)| \leq 2 \cdot \omega_{k+1}(f, 1/m) \cdot \int_{[0,\pi]} (mt+1)^{k+1} K_{m,r}(t)\, dt.$$

Here, $\omega_{k+1}(f, 1/m)$ denotes the *modulus of smoothness* of $f$ as defined on [29, p. 47]. The integral can be bounded via

$$\int_{[0,\pi]} (mt+1)^{k+1} K_{m,r}(t)\, dt$$

$$= \int_{[0,1/m]} \underbrace{(mt+1)}_{\leq 2}{}^{k+1} K_{m,r}(t)\, dt + \int_{[1/m,\pi]} \underbrace{(mt+1)}_{\leq 2mt}{}^{k+1} K_{m,r}(t)\, dt$$

$$\leq 2^{k+1} \cdot \underbrace{\int_{[-\pi,\pi]} K_{m,r}(t)\, dt}_{=1} + 2^{k+1} m^{k+1} \cdot \int_{[0,\pi]} t^{k+1} K_{m,r}(t)\, dt$$

$$\overset{(C.6)}{\leq} 2^{k+1} + 2^{k+1} m^{k+1} \exp(C_1 \cdot r) \cdot (k+1)^{k+1} \cdot m^{-(k+1)} \overset{r \leq 2k}{\leq} \exp(C \cdot k) \cdot k^k$$

with absolute constants $C, C_1 > 0$. Since $\omega_{k+1}(f, 1/m) \leq m^{-k} \cdot \omega(f^{(k)}, 1/m)$ follows from [29, Equation 3.6(5)], the claim is shown. $\qquad\square$

Therefore, we can bound the constant appearing in [29, Theorem 4.3] by $\exp(C \cdot k) \cdot k^k$. It remains to deal with the approximation of *multivariate* periodic functions by *multivariate* trigonometric polynomials which is contained in [29, Theorem 6.6].

**Theorem C.18** (cf. [29, Theorem 6.6]). *Let* $s, k \in \mathbb{N}$ *and* $f \in C_{2\pi}^k(\mathbb{R}^s; \mathbb{R})$. *Let* $\omega_j$ *denote the modulus of continuity of* $\frac{\partial^k f}{\partial x_j^k}$ *for* $j = 1, ..., s$. *Then, with* $E_m^s$ *as introduced in Equation* (C.1), *it holds*

$$E_m^s(f) \leq \exp(C \cdot ks) \cdot k^k \cdot m^{-k} \sum_{j=1}^{s} \omega_j(1/m),$$

*with an absolute constant* $C > 0$.

*Proof.* We follow the proof of [29, Theorem 6.6] with $p_j = k$ and $n_j = m$ for every index $j = 1, ..., s$. For $j = 1, ..., s+1$ define the set $\mathcal{T}_j$ consisting of all functions $g \in C_{2\pi}(\mathbb{R}^s; \mathbb{R})$ that are a trigonometric polynomial in $x_\ell$ of degree at most $m$ for $\ell < j$; in $x_\ell$ for $\ell \geq j$ they should have continuous partial derivatives $\frac{\partial^p g}{\partial x_\ell^p}$ for $0 \leq p \leq k$; the modulus of continuity of $\frac{\partial^k g}{\partial x_\ell^k}$ should not exceed $2^{K_j} \omega_j$, where

$$K_j = (j-1)(k+1) \leq 2ks \quad \text{for } j > 1 \text{ and } K_1 = 1 \leq 2ks.$$

Then it is shown that if $j \in \{1, ..., s\}$ and $f_j \in \mathcal{T}_j$ there exists a function $f_{j+1} \in \mathcal{T}_{j+1}$ for which

$$\|f_j - f_{j+1}\|_{L^\infty(\mathbb{R}^s; \mathbb{R})} \leq \exp(C_1 \cdot k) \cdot k^k \cdot m^{-k} \cdot 2^{2ks} \cdot \omega_j(1/m) \leq \exp(C_2 \cdot ks) \cdot k^k \cdot m^{-k} \cdot \omega_j(1/m)$$

for absolute constants $C_1, C_2 > 0$. This is an application of Theorem C.17. Hence, defining $f_1 := f$, we see

$$\|f - f_{s+1}\|_{L^\infty(\mathbb{R}^s; \mathbb{R})} \leq \sum_{j=1}^{s} \|f_j - f_{j+1}\|_{L^\infty(\mathbb{R}^s; \mathbb{R})} \leq \exp(C_2 \cdot ks) \cdot k^k \cdot m^{-k} \cdot \sum_{j=1}^{s} \omega_j(1/m). \quad\square$$

Therefore, we have shown that the constant appearing in [29, Theorem 6.6] can be bounded from above by $\exp(C \cdot ks) \cdot k^k$.

In the rest of this section, we discuss the operator norm of the operator defined in Lemma C.11. In Lemma C.11 the closed graph theorem is used to show that the operator is bounded. However, the closed graph theorem does not provide any bound on the norm of the operator. Therefore, in order to quantify the operator norm, we need to apply a different technique, which is *Faa di Bruno's formula*. This formula is a generalization of the chain rule to higher order derivatives.

**Theorem C.19.** *Let* $s, k \in \mathbb{N}$. *We define the operator*

$$T: \quad C^k([-1,1]^s; \mathbb{C}) \to C_{2\pi}^k([-\pi, \pi]^s; \mathbb{C}), \quad (Tf)(x_1, ..., x_s) := f(\cos(x_1), ..., \cos(x_s)).$$

*Let* $\boldsymbol{\alpha} \in \mathbb{N}_0^s$ *with* $|\boldsymbol{\alpha}| \leq k$. *Then, for any* $f \in C^k([-1,1]^s; \mathbb{C})$, *we have*

$$\|\partial^{\boldsymbol{\alpha}}(Tf)\|_{L^\infty([\pi,\pi]^s; \mathbb{C})} \leq \prod_{j=1}^{s} \boldsymbol{\alpha}_j^{\boldsymbol{\alpha}_j} \cdot \|f\|_{C^{|\boldsymbol{\alpha}|}([-1,1]^s)}.$$

*Proof.* The proof is by induction over $s$. The case $s = 1$ is an application of Faa di Bruno's formula: We can write $Tf = f \circ g$ with $g(x) = \cos(x)$. We then take $\ell \in \mathbb{N}_0$ with $\ell \leq k$ and some $x \in [-\pi, \pi]$. The set partition version of Faa di Bruno's formula (see for instance [24, p. 219]) then yields

$$\left|(f \circ g)^{(\ell)}(x)\right| \leq \sum_{\pi \in \Pi_\ell} \left(\left|f^{(|\pi|)}(g(x))\right| \cdot \prod_{B \in \pi} \left|g^{(|B|)}(x)\right|\right).$$

Here, $\Pi_\ell$ denotes the set of all partitions of the set $\{1, ..., \ell\}$. Since all derivatives of $g$ are bounded by 1 in absolute value and $|\pi| \leq \ell$ for every partition $\pi \in \Pi_\ell$ we get

$$\|(f \circ g)^{(\ell)}\|_{L^\infty([-\pi,\pi];\mathbb{C})} \leq |\Pi_\ell| \cdot \|f\|_{C^\ell([-1,1];\mathbb{C})}.$$

The number $|\Pi_\ell|$ is the number of possible partitions of the set $\{1, ..., \ell\}$ and is the so-called $\ell$-th *Bell number*. It can be bounded from above by $\ell^\ell$ (see [13, Theorem 2.1]). This proves the case $s = 1$.

We now assume that the claim holds for an arbitrary but fixed $s \in \mathbb{N}$. Take $\boldsymbol{\alpha} \in \mathbb{N}_0^{s+1}$ with $|\boldsymbol{\alpha}| \leq k$. We decompose $\boldsymbol{\alpha} = (\boldsymbol{\alpha}', \boldsymbol{\alpha}_{s+1})$ with $\boldsymbol{\alpha}' \in \mathbb{N}_0^s$. For a fixed variable $y_{s+1} \in [-1, 1]$, we define

$$f_{y_{s+1}}(y_1, ..., y_s) := f(y_1, ..., y_s, y_{s+1}) \quad \text{for} \quad (y_1, ..., y_s) \in [-1, 1]^s.$$

We denote $g(x_1, ..., x_{s+1}) := (\cos(x_1), ..., \cos(x_{s+1}))$, $g_s(x_1, ..., x_s) := (\cos(x_1), ..., \cos(x_s))$ and $\theta(x_{s+1}) := \cos(x_{s+1})$. For every $(x_1, ..., x_{s+1}) \in [-\pi, \pi]^{s+1}$ it then holds

$$(f \circ g)(x_1, ..., x_{s+1}) = \left(f_{\theta(x_{s+1})} \circ g_s\right)(x_1, ..., x_s).$$

We now differentiate $f \circ g$ with respect to the multiindex $\boldsymbol{\alpha}$ and get

$$[\partial^{\boldsymbol{\alpha}}(f \circ g)](x_1, ..., x_{s+1}) = \frac{\partial^{\boldsymbol{\alpha}_{s+1}}}{\partial x_{s+1}^{\boldsymbol{\alpha}_{s+1}}}\left[\partial^{\boldsymbol{\alpha}'}\left(f_{\theta(x_{s+1})} \circ g_s\right)(x_1, ..., x_s)\right]$$

$$= (h_{x_1,...,x_s} \circ \theta)^{(\boldsymbol{\alpha}_{s+1})}(x_{s+1})$$

where we define

$$h_{x_1,...,x_s}(y_{s+1}) := \partial^{\boldsymbol{\alpha}'}\left(f_{y_{s+1}} \circ g_s\right)(x_1, ..., x_s) \quad \text{for} \quad (x_1, ..., x_s) \in [-\pi, \pi]^s \text{ and } y_{s+1} \in [-1, 1].$$

Using the case $s = 1$, we get

$$|[\partial^{\boldsymbol{\alpha}}(f \circ g)](x_1, ..., x_{s+1})| = \left|(h_{x_1,...,x_s} \circ \theta)^{(\boldsymbol{\alpha}_{s+1})}(x_{s+1})\right| \leq \boldsymbol{\alpha}_{s+1}^{\boldsymbol{\alpha}_{s+1}} \cdot \|h_{x_1,...,x_s}\|_{C^{\boldsymbol{\alpha}_{s+1}}([-1,1];\mathbb{C})}$$

for any fixed $(x_1, ..., x_s) \in [-\pi, \pi]^s$.

It remains to bound $\|h_{x_1,...,x_s}\|_{C^{\boldsymbol{\alpha}_{s+1}}([-1,1];\mathbb{C})}$. To this end, we fix $\ell \in \mathbb{N}_0$ with $\ell \leq \boldsymbol{\alpha}_{s+1}$. We further denote

$$F_{y_1,...,y_s}(y_{s+1}) := f(y_1, ..., y_s, y_{s+1}) \quad \text{for} \quad (y_1, ..., y_{s+1}) \in [-1, 1]^{s+1}.$$

For arbitrary $(x_1, ..., x_s) \in [-\pi, \pi]^s$ and $y_{s+1} \in [-1, 1]$ we then see

$$h_{x_1,...,x_s}^{(\ell)}(y_{s+1}) = \partial^{\boldsymbol{\alpha}'}\left[(x_1, ..., x_s) \mapsto F_{g_s(x_1,...,x_s)}^{(\ell)}(y_{s+1})\right] = \partial^{\boldsymbol{\alpha}'}\left[H_{y_{s+1}} \circ g_s\right](x_1, ..., x_s)$$

where

$$H_{y_{s+1}}(y_1, ..., y_s) := F_{y_1,...,y_s}^{(\ell)}(y_{s+1}) \quad \text{for } (y_1, ..., y_s) \in [-1, 1]^s.$$

Hence, we see by induction that

$$\left|h_{x_1,...,x_s}^{(\ell)}(y_{s+1})\right| = \left|\partial^{\boldsymbol{\alpha}'}\left[H_{y_{s+1}} \circ g_s\right](x_1, ..., x_s)\right| \overset{\text{IH}}{\leq} \prod_{j=1}^s \boldsymbol{\alpha}_j^{\boldsymbol{\alpha}_j} \cdot \|H_{y_{s+1}}\|_{C^{|\boldsymbol{\alpha}'|}([-1,1]^s;\mathbb{C})}$$

$$\leq \prod_{j=1}^s \boldsymbol{\alpha}_j^{\boldsymbol{\alpha}_j} \cdot \|f\|_{C^{|\boldsymbol{\alpha}|}([-1,1]^{s+1};\mathbb{C})}$$

as was to be shown. $\qquad\qquad\square$

*Remark* C.20. For a multiindex $\boldsymbol{\alpha} \in \mathbb{N}_0^s$ with $|\boldsymbol{\alpha}| \leq k$ we see

$$\prod_{j=1}^s \boldsymbol{\alpha}_j^{\boldsymbol{\alpha}_j} \leq k^{\sum_{j=1}^s \boldsymbol{\alpha}_j} \leq k^k.$$

Hence, the norm of the operator introduced in Lemma C.11 can be bounded from above by $k^k$.

## C.3 Proof of Theorem 3.2

For any natural number $\ell \in \mathbb{N}_0$, we denote by $T_\ell$ the $\ell$-th Chebyshev polynomial, satisfying

$$T_\ell\left(\cos(x)\right) = \cos(\ell x), \quad x \in \mathbb{R}.$$

For a multi-index $\mathbf{k} \in \mathbb{N}_0^s$ we define

$$T_{\mathbf{k}}(x) := \prod_{j=1}^{s} T_{\mathbf{k}_j}\left(x_j\right), \quad x \in [-1,1]^s.$$

The proof of Theorem 3.2 relies on the fact that $C^k$-functions can by approximated at a certain rate using linear combinations of the $T_{\mathbf{k}}$ (see Theorem C.15). We also refer to Figure 4 for an illustration of the overall proof strategy of Theorem 3.2.

*Proof of Theorem 3.2.* Choose $M \in \mathbb{N}$ as the largest integer for which $(16M - 7)^{2n} \leq m$, where we assume without loss of generality that $9^{2n} \leq m$, which can be done by choosing $\sigma_j = 0$ for all $j \in \{1, ..., m\}$ for $m < 9^{2n}$, at the cost of possibly enlarging $c$. First we note that by the choice of $M$ the inequality

$$m \leq (16M + 9)^{2n}$$

holds true. Since $16M + 9 \leq 25M$, we get $m \leq 25^{2n} \cdot M^{2n}$ or equivalently

$$\frac{m^{1/2n}}{25} \leq M. \tag{C.7}$$

According to Theorem C.15 we choose a constant $c_1 = c_1(n,k)$ with the property that for any function $f \in C^k\left([-1,1]^{2n}; \mathbb{R}\right)$ there exists a polynomial

$$P = \sum_{0 \leq \mathbf{k} \leq 2M-1} \mathcal{V}_{\mathbf{k}}^M(f) \cdot T_{\mathbf{k}}$$

of coordinatewise degree at most $2M - 1$ satisfying

$$\|f - P\|_{L^\infty([-1,1]^{2n};\mathbb{R})} \leq \frac{c_1}{M^k} \cdot \|f\|_{C^k([-1,1]^{2n};\mathbb{R})}.$$

Furthermore, according to Theorem C.15, we choose a constant $c_2 = c_2(n)$, such that the inequality

$$\sum_{0 \leq \mathbf{k} \leq 2M-1} \left|\mathcal{V}_{\mathbf{k}}^M(f)\right| \leq c_2 \cdot M^n \cdot \|f\|_{L^\infty([-1,1]^{2n};\mathbb{R})} \leq c_2 \cdot M^n \cdot \|f\|_{C^k([-1,1]^{2n};\mathbb{R})}$$

holds for all $f \in C^k\left([-1,1]^{2n}; \mathbb{R}\right)$. The final constant is then defined to be

$$c = c(n,k) := \sqrt{2} \cdot 25^k \cdot (c_1 + c_2).$$

Fix $\mathbf{k} \leq 2M - 1$. Since $T_{\mathbf{k}}$ is a polynomial of componentwise degree less or equal to $2M - 1$ with $\varphi_n$ as in (A.1), we have a representation

$$\left(T_{\mathbf{k}} \circ \varphi_n^{-1}\right)(z) = \sum_{\substack{\boldsymbol{\ell}^1, \boldsymbol{\ell}^2 \in \mathbb{N}_0^n \\ \boldsymbol{\ell}^1, \boldsymbol{\ell}^2 \leq 2M-1}} a_{\boldsymbol{\ell}^1, \boldsymbol{\ell}^2}^{\mathbf{k}} \prod_{t=1}^{n} \operatorname{Re}(z_t)^{\boldsymbol{\ell}_t^1} \operatorname{Im}(z_t)^{\boldsymbol{\ell}_t^2}$$

with suitably chosen coefficients $a_{\boldsymbol{\ell}^1,\boldsymbol{\ell}^2}^{\mathbf{k}} \in \mathbb{C}$. By using the identities $\operatorname{Re}(z_t) = \frac{1}{2}(z_t + \overline{z_t})$ and also $\operatorname{Im}(z_t) = \frac{1}{2i}(z_t - \overline{z_t})$ we can rewrite $T_{\mathbf{k}} \circ \varphi_n^{-1}$ into a complex polynomial in $z$ and $\overline{z}$, i.e.,

$$\left(T_{\mathbf{k}} \circ \varphi_n^{-1}\right)(z) = \sum_{\substack{\boldsymbol{\ell}^1, \boldsymbol{\ell}^2 \in \mathbb{N}_0^n \\ \boldsymbol{\ell}^1, \boldsymbol{\ell}^2 \leq 4M-2}} b_{\boldsymbol{\ell}^1, \boldsymbol{\ell}^2}^{\mathbf{k}} z^{\boldsymbol{\ell}^1} \overline{z}^{\boldsymbol{\ell}^2}$$

with complex coefficients $b_{\boldsymbol{\ell}^1,\boldsymbol{\ell}^2}^{\mathbf{k}} \in \mathbb{C}$. Using Theorem 3.1, we choose $\rho_1, ..., \rho_m \in \mathbb{C}^n$ and $b \in \mathbb{C}$, such that for any polynomial $P \in \left\{T_{\mathbf{k}} \circ \varphi_n^{-1} : \mathbf{k} \leq 2M - 1\right\} \subseteq \mathcal{P}_{4M-2}^n$ there are coefficients $\sigma_1(P), ..., \sigma_m(P) \in \mathbb{C}$, such that

$$\|g_P - P\|_{L^\infty(\Omega_n;\mathbb{C})} \leq M^{-k-n}, \tag{C.8}$$

where

$$g_P := \sum_{t=1}^{m} \sigma_t(P)\phi\left(\rho_t^T z + b\right).$$

Note that here we implicitly use the bound $(4 \cdot (4M - 2) + 1)^{2n} \leq m$. We are now going to show that the chosen constant and the chosen vectors $\rho_t$ have the desired property.

Let $f \in C^k(\Omega_n; \mathbb{C})$. By splitting $f$ into real and imaginary part, we write $f = f_1 + i \cdot f_2$ with $f_1, f_2 \in C^k(\Omega_n; \mathbb{R})$. For the following, fix $j \in \{1, 2\}$ and note that $f_j \circ \varphi_n \in C^k\left([-1, 1]^{2n}; \mathbb{R}\right)$. By choice of $c_1$, there exists a polynomial $P$ with the property

$$\|f_j \circ \varphi_n - P\|_{L^\infty([-1,1]^{2n};\mathbb{R})} \leq \frac{c_1}{M^k} \cdot \|f_j \circ \varphi_n\|_{C^k([-1,1]^{2n};\mathbb{R})}$$

or equivalently

$$\left\|f_j - P \circ \varphi_n^{-1}\right\|_{L^\infty(\Omega_n;\mathbb{R})} \leq \frac{c_1}{M^k} \cdot \|f_j\|_{C^k(\Omega_n;\mathbb{R})}, \tag{C.9}$$

where $P \circ \varphi_n^{-1}$ can be written in the form

$$\left(P \circ \varphi_n^{-1}\right)(z) = \sum_{0 \leq \mathbf{k} \leq 2M-1} \mathcal{V}_{\mathbf{k}}^M\left(f_j \circ \varphi_n\right) \cdot \left(T_{\mathbf{k}} \circ \varphi_n^{-1}\right)(z).$$

We choose the function $g_{T_{\mathbf{k}} \circ \varphi_n^{-1}}$ according to (C.8). Thus, writing

$$g_j := \sum_{0 \leq \mathbf{k} \leq 2M-1} \mathcal{V}_{\mathbf{k}}^M\left(f_j \circ \varphi_n\right) \cdot g_{T_{\mathbf{k}} \circ \varphi_n^{-1}},$$

we obtain

$$\left\|P \circ \varphi_n^{-1} - g_j\right\|_{L^\infty(\Omega_n;\mathbb{R})} \leq \sum_{0 \leq \mathbf{k} \leq 2M-1} \left|\mathcal{V}_{\mathbf{k}}^M\left(f_j \circ \varphi_n\right)\right| \cdot \underbrace{\left\|T_{\mathbf{k}} \circ \varphi_n^{-1} - g_{T_{\mathbf{k}} \circ \varphi_n^{-1}}\right\|_{L^\infty(\Omega_n;\mathbb{R})}}_{\leq M^{-k-n}}$$

$$\leq M^{-k-n} \cdot \sum_{0 \leq \mathbf{k} \leq 2M-1} \left|\mathcal{V}_{\mathbf{k}}^M\left(f_j \circ \varphi_n\right)\right|$$

$$\leq \frac{c_2}{M^k}\|f_j \circ \varphi_n\|_{C^k([-1,1]^{2n};\mathbb{R})} = \frac{c_2}{M^k}\|f_j\|_{C^k(\Omega_n;\mathbb{R})}. \tag{C.10}$$

Combining (C.9) and (C.10), we see

$$\|f_j - g_j\|_{L^\infty(\Omega_n;\mathbb{R})} \leq \frac{c_1 + c_2}{M^k} \cdot \|f_j\|_{C^k(\Omega_n;\mathbb{R})} \leq \frac{c_1 + c_2}{M^k} \cdot \|f\|_{C^k(\Omega_n;\mathbb{C})}.$$

In the end, define

$$g := g_1 + i \cdot g_2.$$

Since the vectors $\rho_t$ have been chosen fixed, it is clear that, after rearranging, $g$ has the desired form, i.e., $g = \sigma^T \Phi$ where $\Phi(z) = (\phi(\rho_t z + b))_{t=1}^m$. Furthermore, one obtains the bound

$$\|f - g\|_{L^\infty(\Omega_n;\mathbb{C})} \leq \sqrt{\|f_1 - g_1\|_{L^\infty(\Omega_n;\mathbb{R})}^2 + \|f_2 - g_2\|_{L^\infty(\Omega_n;\mathbb{R})}^2}$$

$$\leq \frac{c_1 + c_2}{M^k} \cdot \sqrt{\|f\|_{C^k(\Omega_n;\mathbb{C})}^2 + \|f\|_{C^k(\Omega_n;\mathbb{C})}^2}$$

$$\leq \frac{\sqrt{2} \cdot (c_1 + c_2)}{M^k} \cdot \|f\|_{C^k(\Omega_n;\mathbb{C})}.$$

Using (C.7), we see

$$\|f - g\|_{L^\infty(\Omega_n;\mathbb{C})} \leq \frac{\sqrt{2} \cdot 25^k \cdot (c_1 + c_2)}{m^{k/2n}} \cdot \|f\|_{C^k(\Omega_n;\mathbb{C})},$$

as desired.

The linearity and continuity of the maps $f \mapsto \sigma_j(f)$ (with respect to the $\|\cdot\|_{L^\infty}$-norm) follow easily from the fact that the map $f \mapsto \mathcal{V}_{\mathbf{k}}^M(f)$ is a continuous linear functional for every multiindex $0 \leq \mathbf{k} \leq 2M - 1$. $\qquad\square$

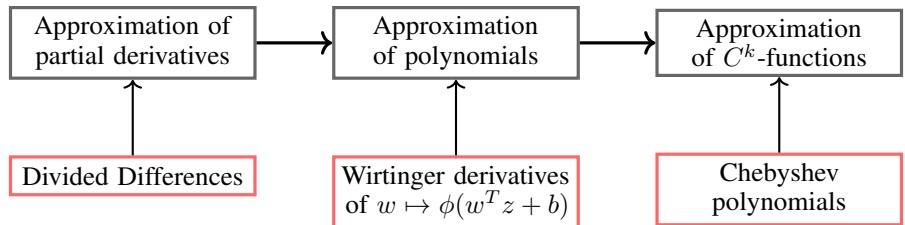

Figure 4: Schematic for the proof of the main result (Theorem 3.2). The first row shows the different steps of the proof and the second row indicates the main tools used.

# D    Approximation of holomorphically extendable functions

In this appendix we provide the proofs for the statements contained in Remark 3.3. The proofs mainly rely on results about sparse polynomial approximation [2].

**Definition D.1** (cf. [2, Assumption 2.3]). Let $s \in \mathbb{N}$ and $\boldsymbol{\nu} \in (1, \infty)^s$. For every $j \in \{1, ..., s\}$ let

$$\mathcal{E}_{\boldsymbol{\nu}_j} := \left\{ \frac{z + z^{-1}}{2} \; : \; z \in \mathbb{C}, 1 \leq |z| \leq \boldsymbol{\nu}_j \right\}.$$

We then define the *(filled-in) Bernstein polyellipse* of parameter $\boldsymbol{\nu}$ as

$$\mathcal{E}_{\boldsymbol{\nu}} := \mathcal{E}_{\boldsymbol{\nu}_1} \times \cdots \times \mathcal{E}_{\boldsymbol{\nu}_s} \subseteq \mathbb{C}^s$$

and observe that $[-1, 1]^s \subseteq \mathcal{E}_{\boldsymbol{\nu}}$ when we interpret $[-1, 1]^s$ as a subset of $\mathbb{C}^s$. We further define

$$\mathcal{V}_s(\boldsymbol{\nu}) := \left\{ f : [-1, 1]^s \to \mathbb{C} \; : \; \exists \text{ open } U \supseteq \mathcal{E}_{\boldsymbol{\nu}} \text{ and } \tilde{f} : U \to \mathbb{C} \text{ holomorphic with } \tilde{f}\Big|_{[-1,1]^s} = f \right\}.$$

Here, $[-1, 1]^s$ is again interpreted as a subset of $\mathbb{C}^s$. Moreover, note that such an extension $\tilde{f}$ is, if existent, unique, as follows from the identity theorem for holomorphic functions. Hence, the expression

$$\|f\|_{\mathcal{V}_s(\boldsymbol{\nu})} := \|\tilde{f}\|_{L^\infty(\mathcal{E}_{\boldsymbol{\nu}};\mathbb{C})}$$

is well-defined.

**Definition D.2** (cf. [2, pp. 25, 28 ff.]). Let $s \in \mathbb{N}$. We define a probability measure on $[-1, 1]^s$ via

$$d\mu_s := \prod_{j=1}^{s} \frac{1}{\pi \sqrt{1 - x_j^2}} \, dx.$$

We define the *normalized* Chebyshev polynomials for $\mathbf{k} \in \mathbb{N}_0^s$ as

$$\widetilde{T_{\mathbf{k}}}(x) := 2^{\|\mathbf{k}\|_0/2} \prod_{j=1}^{s} \cos(\mathbf{k}_j \arccos(x_j)),$$

where $\|\mathbf{k}\|_0 := \#\{1 \leq j \leq s : \mathbf{k}_j \neq 0\}$. Note that this definition differs slightly from the notion used in Appendices C.1 and C.3.

The following lemma is crucial for deriving the approximation rate in Theorem D.5. The proofs can be found in [2].

**Lemma D.3** (cf. [2, Remark 2.15 and Theorem 3.2]). *Let $s \in \mathbb{N}$, $\boldsymbol{k} \in \mathbb{N}_0^s$, $\boldsymbol{\nu} \in (1, \infty)^s$ and $f \in \mathcal{V}_s(\boldsymbol{\nu})$. Then it holds*

1. *$\|\widetilde{T_{\boldsymbol{k}}}\|_{L^\infty([-1,1]^s;\mathbb{R})} = 2^{\|\boldsymbol{k}\|_0/2}$, where $\|\boldsymbol{k}\|_0 = \#\{1 \leq j \leq s : k_j \neq 0\}$;*

2. *$|\langle \widetilde{T_{\boldsymbol{k}}}, f \rangle_{\mu_s}| \leq \boldsymbol{\nu}^{-\boldsymbol{k}} \cdot 2^{\|\boldsymbol{k}\|_0/2} \cdot \|f\|_{\mathcal{V}_s(\boldsymbol{\nu})}$.*

It is a well-known fact that the Chebyshev polynomials $\widetilde{T_{\mathbf{k}}}$ form an *orthonormal basis* of $L^2_{\mu_s}([-1, 1]^s; \mathbb{C})$. The following proposition states that functions from $\mathcal{V}_s(\boldsymbol{\nu})$ can even be approximated uniformly by linear combinations of the $\widetilde{T_{\mathbf{k}}}$ at a certain rate. The proof follows essentially by applying Lemma D.3.

**Proposition D.4.** *Let $s, m \in \mathbb{N}$ and $\boldsymbol{\nu} \in (1, \infty)^s$. Let $\nu := \min\limits_{j=1,\dots,s} \boldsymbol{\nu}_j$. Then there exists a constant $c = c(s, \boldsymbol{\nu}) > 0$ with the following property: For every $f \in \mathcal{V}_s(\boldsymbol{\nu})$, defining*

$$P_m := \sum_{\substack{\boldsymbol{k} \in \mathbb{N}_0^s \\ \boldsymbol{k} \leq m}} \langle f, \widetilde{T_{\boldsymbol{k}}} \rangle_{\mu_s} \cdot \widetilde{T_{\boldsymbol{k}}},$$

*it holds*

$$\|f - P_m\|_{L^\infty([-1,1]^s;\mathbb{C})} \leq c \cdot \nu^{-m} \cdot \|f\|_{\mathcal{V}_s(\boldsymbol{\nu})}.$$

*Proof.* Let $f \in \mathcal{V}_s(\boldsymbol{\nu})$. Since the $\widetilde{T_{\mathbf{k}}}$ form an orthonormal basis of $L_{\mu_s}^2([-1,1]^s;\mathbb{C})$ and since $\mu_s$ is a probability measure, so that $f \in C([-1,1]^s;\mathbb{C}) \subseteq L_{\mu_2}^2([-1,1]^s;\mathbb{C})$, it follows that

$$f = \sum_{\mathbf{k} \in \mathbb{N}_0^s} \langle f, \widetilde{T_{\mathbf{k}}} \rangle_{\mu_s} \cdot \widetilde{T_{\mathbf{k}}}$$

with unconditional convergence in $L_{\mu_s}^2$. For $x \in [-1,1]^s$, note that

$$\sum_{\mathbf{k} \in \mathbb{N}_0^s} |\langle f, \widetilde{T_{\mathbf{k}}} \rangle_{\mu_s}| \cdot |\widetilde{T_{\mathbf{k}}}(x)| \leq \sum_{\mathbf{k} \in \mathbb{N}_0^s} |\langle f, \widetilde{T_{\mathbf{k}}} \rangle_{\mu_s}| \cdot \|\widetilde{T_{\mathbf{k}}}\|_{L^\infty([-1,1]^s;\mathbb{C})} \leq 2^s \|f\|_{\mathcal{V}_s(\boldsymbol{\nu})} \cdot \sum_{\mathbf{k} \in \mathbb{N}_0^s} \boldsymbol{\nu}^{-\mathbf{k}}.$$

Here, we employed Lemma D.3 at the last inequality. From $\boldsymbol{\nu} > 1$ it follows that

$$\sum_{\mathbf{k} \in \mathbb{N}_0^s} \langle f, \widetilde{T_{\mathbf{k}}} \rangle_{\mu_s} \cdot \widetilde{T_{\mathbf{k}}}$$

converges *pointwise* (even uniformly) and, since all the involved functions are continuous, this pointwise limit then has to coincide with the $L_{\mu_s}^2$-limit $f$. Hence, it holds

$$\|f - P_m\|_{L^\infty([-1,1]^s;\mathbb{C})} \leq \sum_{\substack{\mathbf{k} \in \mathbb{N}_0^s \\ \mathbf{k} \not\leq m}} |\langle f, \widetilde{T_{\mathbf{k}}} \rangle_{\mu_s}| \cdot \|\widetilde{T_{\mathbf{k}}}\|_{L^\infty([-1,1]^s;\mathbb{C})}$$

$$\leq 2^s \|f\|_{\mathcal{V}_s(\boldsymbol{\nu})} \cdot \sum_{\substack{\mathbf{k} \in \mathbb{N}_0^s \\ |\mathbf{k}| \not\leq m}} \boldsymbol{\nu}^{-\mathbf{k}},$$

where we again used Lemma D.3. To complete the proof we compute

$$\sum_{\substack{\mathbf{k} \in \mathbb{N}_0^s \\ |\mathbf{k}| \not\leq m}} \boldsymbol{\nu}^{-\mathbf{k}} \leq \sum_{j=1}^s \sum_{\substack{\mathbf{k} \in \mathbb{N}_0^s \\ \mathbf{k}_j > m}} \boldsymbol{\nu}^{-\mathbf{k}} = \sum_{j=1}^s \left( \prod_{\substack{1 \leq \ell \leq s \\ l \neq j}} \left( \sum_{k=0}^\infty \boldsymbol{\nu}_\ell^{-k} \right) \cdot \sum_{k=m+1}^\infty \boldsymbol{\nu}_j^{-k} \right)$$

$$\leq \sum_{j=1}^s \left( \boldsymbol{\nu}_j^{-(m+1)} \prod_{1 \leq \ell \leq s} \left( \sum_{k=0}^\infty \boldsymbol{\nu}_\ell^{-k} \right) \right)$$

$$= \prod_{1 \leq \ell \leq s} \left( \sum_{k=0}^\infty \boldsymbol{\nu}_\ell^{-k} \right) \cdot \sum_{j=1}^s \boldsymbol{\nu}_j^{-(m+1)}$$

$$= \prod_{1 \leq \ell \leq s} \left( \frac{\boldsymbol{\nu}_\ell}{\boldsymbol{\nu}_\ell - 1} \right) \cdot s \cdot \nu^{-(m+1)}$$

and define $c(s, \boldsymbol{\nu}) := 2^s \cdot s \cdot \prod_{\ell=1}^s \frac{\boldsymbol{\nu}_j}{\boldsymbol{\nu}_j - 1} \cdot \nu^{-1}$. $\qquad \square$

To formulate the result for the approximation of holomorphically extendable functions using CVNNs we need to transfer the definition of $\mathcal{V}_s(\boldsymbol{\nu})$ to the complex setting. For $\boldsymbol{\nu} \in (1, \infty)^{2n}$ and with $\varphi_n$ as in Equation (A.1), we hence write

$$\mathcal{W}_n(\boldsymbol{\nu}) := \left\{ f : \Omega_n \to \mathbb{C} : f \circ \varphi_n \big|_{[-1,1]^{2n}} \in \mathcal{V}_{2n}(\boldsymbol{\nu}) \right\}$$

For $f \in \mathcal{W}_n(\boldsymbol{\nu})$ we define
$$\|f\|_{\mathcal{W}_n(\boldsymbol{\nu})} := \|f \circ \varphi_n\|_{\mathcal{V}_{2n}(\boldsymbol{\nu})}.$$

Thus, $\mathcal{W}_n(\boldsymbol{\nu})$ consists of all complex-valued functions defined on $\Omega_n$ that can be holomorphically extended onto some polyellipse in $\mathbb{C}^{2n}$, where $\Omega_n \subseteq \mathbb{C}^n$ is interpreted as a subset of $\mathbb{R}^{2n}$ and then as a subset of $\mathbb{C}^{2n}$. The final approximation result then reads as follows.

**Theorem D.5.** *Let $n \in \mathbb{N}$ and $\boldsymbol{\nu} \in (1, \infty)^{2n}$. Set*
$$\nu := \min_{1 \leq j \leq 2n} \boldsymbol{\nu}_j.$$

*Then there exists a constant $c = c(n, \boldsymbol{\nu}) > 0$ with the following property: For every function $\phi : \mathbb{C} \to \mathbb{C}$ that is smooth and non-polyharmonic on some open set $\varnothing \neq U \subset \mathbb{C}$ and for every $m \in \mathbb{N}$ there exists a first layer $\Phi \in \mathcal{F}_{n,m}^{\phi}$ with the property that for every $f \in \mathcal{W}_n(\boldsymbol{\nu})$ there exist coefficients $\sigma = \sigma(f) \in \mathbb{C}^m$ such that*
$$\|f - \sigma^T \Phi\|_{L^\infty(\Omega_n; \mathbb{C})} \leq c \cdot \nu^{-m^{1/(2n)}/17} \cdot \|f\|_{\mathcal{W}_n(\boldsymbol{\nu})}.$$

*Moreover, the map $f \mapsto \sigma(f)$ is a continuous linear functional with respect to the $L^\infty$-norm.*

*Proof.* Choose $M \in \mathbb{N}$ as the largest integer satisfying $(8M+1)^{2n} \leq m$, where we assume without loss of generality that $9^{2n} \leq m$. This can be done by choosing $\sigma = 0$ for $m < 9^{2n}$ at the cost of possibly enlarging $c$. Note that the maximality of $M$ implies $(8M+9)^{2n} > m$. By using $8M + 9 \leq 17M$, this gives us
$$M > \frac{1}{17} \cdot m^{1/(2n)}. \tag{D.1}$$
Choose the constant $c_1 = c_1(2n, \boldsymbol{\nu})$ according to Proposition D.4.

Fix $\mathbf{k} \in \mathbb{N}_0^{2n}$ with $\mathbf{k} \leq M$. Since $\widetilde{T_\mathbf{k}}$ is a polynomial of componentwise degree at most $M$, we have a representation
$$\left(\widetilde{T_\mathbf{k}} \circ \varphi_n^{-1}\right)(z) = \sum_{\substack{\boldsymbol{\ell}^1, \boldsymbol{\ell}^2 \in \mathbb{N}_0^n \\ \boldsymbol{\ell}^1, \boldsymbol{\ell}^2 \leq M}} a_{\boldsymbol{\ell}^1, \boldsymbol{\ell}^2}^\mathbf{k} \prod_{t=1}^n \operatorname{Re}(z_t)^{\boldsymbol{\ell}_t^1} \operatorname{Im}(z_t)^{\boldsymbol{\ell}_t^2}$$

with suitably chosen coefficients $a_{\boldsymbol{\ell}^1, \boldsymbol{\ell}^2}^\mathbf{k} \in \mathbb{C}$. By using the identities $\operatorname{Re}(z_t) = \frac{1}{2}(z_t + \overline{z_t})$ and also $\operatorname{Im}(z_t) = \frac{1}{2i}(z_t - \overline{z_t})$, we can rewrite $\widetilde{T_\mathbf{k}} \circ \varphi_n^{-1}$ into a complex polynomial in $z$ and $\overline{z}$, i.e.,
$$\left(\widetilde{T_\mathbf{k}} \circ \varphi_n^{-1}\right)(z) = \sum_{\substack{\boldsymbol{\ell}^1, \boldsymbol{\ell}^2 \in \mathbb{N}_0^n \\ \boldsymbol{\ell}^1, \boldsymbol{\ell}^2 \leq 2M}} b_{\boldsymbol{\ell}^1, \boldsymbol{\ell}^2}^\mathbf{k} z^{\boldsymbol{\ell}^1} \overline{z}^{\boldsymbol{\ell}^2}$$

with complex coefficients $b_{\boldsymbol{\ell}^1, \boldsymbol{\ell}^2}^\mathbf{k} \in \mathbb{C}$. Using Theorem 3.1, we choose $\rho_1, ..., \rho_m \in \mathbb{C}^n$ and $b \in \mathbb{C}$, such that for any polynomial $P \in \left\{\widetilde{T_\mathbf{k}} \circ \varphi_n^{-1} : \mathbf{k} \leq M\right\} \subseteq \mathcal{P}_{2M}^n$ there exist coefficients $\sigma_1(P), ..., \sigma_m(P) \in \mathbb{C}$, such that
$$\|g_P - P\|_{L^\infty(\Omega_n; \mathbb{C})} \leq \left(\sum_{\mathbf{k} \in \mathbb{N}_0^{2n}} \boldsymbol{\nu}^{-\mathbf{k}}\right)^{-1} \cdot \nu^{-(M+1)}, \tag{D.2}$$

where
$$g_P := \sum_{t=1}^m \sigma_t(P) \phi\left(\rho_t^T z + b\right).$$

Note that here we implicitly use the bound $(4 \cdot (2M) + 1)^{2n} \leq m$. We are now going to show that the first layer $\Phi \in \mathcal{F}_{n,m}^{\phi}$ defined using the $\rho_t$ and $b$ (i.e., $\Phi(z) = (\phi(\rho_t^T z + b))_{t=1}^m$) has the desired property.

To this end, take an arbitrary function $f \in \mathcal{W}_n(\boldsymbol{\nu})$. Proposition D.4 tells us that
$$\|f - P_M \circ \varphi_n^{-1}\|_{L^\infty(\Omega_n; \mathbb{C})} = \|f \circ \varphi_n - P_M\|_{L^\infty([-1,1]^{2n}; \mathbb{C})} \leq c_1 \cdot \nu^{-(M+1)} \cdot \|f \circ \varphi_n\|_{\mathcal{V}_{2n}(\boldsymbol{\nu})}, \tag{D.3}$$

where

$$P_M \circ \varphi_n^{-1} = \sum_{\substack{\mathbf{k} \in \mathbb{N}_0^{2n} \\ \mathbf{k} \leq M}} \langle f \circ \varphi_n, \widetilde{T_\mathbf{k}} \rangle_{\mu_{2n}} \cdot (\widetilde{T_\mathbf{k}} \circ \varphi_n^{-1}).$$

We then define the approximating network

$$g := \sum_{\substack{\mathbf{k} \in \mathbb{N}_0^{2n} \\ \mathbf{k} \leq M}} \langle f \circ \varphi_n, \widetilde{T_\mathbf{k}} \rangle_{\mu_{2n}} \cdot g_{\widetilde{T_\mathbf{k}} \circ \varphi_n^{-1}}.$$

From the definition of the functions $g_{\widetilde{T_\mathbf{k}} \circ \varphi_n^{-1}}$ it follows directly that $g$ is a shallow CVNN with first layer $\Phi \in \mathcal{F}_{n,m}^\phi$. Furthermore, it holds

$$\|P_M \circ \varphi_n^{-1} - g\|_{L^\infty(\Omega_n;\mathbb{C})} \leq \sum_{\substack{\mathbf{k} \in \mathbb{N}_0^{2n} \\ \mathbf{k} \leq M}} |\langle f \circ \varphi_n, \widetilde{T_\mathbf{k}} \rangle_{\mu_{2n}}| \cdot \|\widetilde{T_\mathbf{k}} \circ \varphi_n^{-1} - g_{\widetilde{T_\mathbf{k}} \circ \varphi_n^{-1}}\|_{L^\infty(\Omega_n;\mathbb{C})}$$

$$\overset{\text{(D.2),Lem. D.3}}{\leq} 2^n \cdot \|f \circ \varphi_n\|_{\mathcal{V}_{2n}(\boldsymbol{\nu})} \cdot \sum_{\substack{\mathbf{k} \in \mathbb{N}_0^{2n} \\ \mathbf{k} \leq M}} \boldsymbol{\nu}^{-\mathbf{k}} \cdot \left( \sum_{\mathbf{k} \in \mathbb{N}_0^n} \boldsymbol{\nu}^{-\mathbf{k}} \right)^{-1} \cdot \nu^{-(M+1)}$$

$$\leq 2^n \cdot \|f\|_{\mathcal{W}_n(\boldsymbol{\nu})} \cdot \nu^{-(M+1)}.$$

Combining this estimate with Equation (D.3) and applying the triangle inequality gives us

$$\|f - g\|_{L^\infty(\Omega_n;\mathbb{C})} \leq (c_1 + 2^n) \cdot \nu^{-(M+1)} \cdot \|f\|_{L^\infty(\mathcal{E}_{\boldsymbol{\nu}};\mathbb{C})}.$$

Hence, the claim follows taking $c := c_1 + 2^n$ and using (D.1).

The continuity of the linear map $f \mapsto \sigma(f)$ with respect to the $L^\infty$-norm follows directly from the fact that $f \mapsto \langle f \circ \varphi_n, \widetilde{T_\mathbf{k}} \rangle_{\mu_{2n}}$ is trivially continuous with respect to the $L^2_{\mu_{2n}}$-norm and hence also continuous with respect to the $L^\infty$-norm since $\mu_{2n}$ is a probability measure. $\square$

# E  Postponed proofs for the optimality results in the case of continuous weight selection

In this section we provide the proofs for the optimality results derived if a continuous weight selection is assumed. Specifically, we prove Theorems 4.1 and 5.1. The proofs of both results rely on a very general result about the approximation in normed spaces by subsets that are parametrizable with $m$ parameters [19]. We decided to include a detailed proof for this approximation result in this paper since the nature of the continuity assumption in [19] is not completely clear.

**Proposition E.1** ([19, Theorem 3.1]). *Let $(X, \|\cdot\|_X)$ be a normed space, $\varnothing \neq K \subseteq X$ a subset and $V \subseteq X$ a linear, not necessarily closed subspace of $X$ containing $K$. Let $m \in \mathbb{N}$, let $\overline{a} : K \to \mathbb{R}^m$ be a map which is continuous with respect to some norm $\|\cdot\|_V$ on $V$ and $M : \mathbb{R}^m \to X$ some arbitrary map. Let*

$$b_m(K)_X := \sup_{X_{m+1}} \sup \left\{ \varrho \geq 0 : U_\varrho(X_{m+1}) \subseteq K \right\}, \tag{E.1}$$

*where the first supremum is taken over all $(m+1)$-dimensional linear subspaces $X_{m+1}$ of $X$ and*

$$U_\varrho(X_{m+1}) := \{y \in X_{m+1} : \|y\|_X \leq \varrho\}.$$

*Further, we set $b_m(K)_X := 0$ if the supremum in (E.1) is not well-defined as a quantity in $[0, \infty]$. Then it holds*

$$\sup_{x \in K} \|x - M(\overline{a}(x))\|_X \geq b_m(K)_X.$$

*Proof.* The claim is trivial if $b_m(K)_X = 0$. Thus, assume $b_m(K)_X > 0$. Let $0 < \varrho \leq b_m(K)_X$ be any number such that there exists an $(m+1)$-dimensional subspace $X_{m+1}$ of $X$ with $U_\varrho(X_{m+1}) \subseteq K$. It follows $U_\varrho(X_{m+1}) \subseteq V$, hence $X_{m+1} \subseteq V$, so $\|\cdot\|_V$ defines a norm on $X_{m+1}$. Thus, the

restriction of $\bar{a}$ to $\partial U_\varrho(X_{m+1})$ is a continuous mapping to $\mathbb{R}^m$ with respect to $\|\cdot\|_V$. Since all norms are equivalent on the finite-dimensional space $X_{m+1}$, the Borsuk-Ulam-Theorem [18, Corollary 4.2] yields the existence of a point $x_0 \in \partial U_\varrho(X_{m+1})$ with $\bar{a}(x_0) = \bar{a}(-x_0)$. We then see

$$2\varrho = 2\|x_0\|_X \leq \|x_0 - M(\bar{a}(x_0))\|_X + \|x_0 + M(\bar{a}(-x_0))\|_X$$
$$= \|x_0 - M(\bar{a}(x_0))\|_X + \| - x_0 - M(\bar{a}(-x_0))\|_X,$$

and hence at least one of the two summands on the right has to be larger than or equal to $\varrho$. □

## E.1  Proof of Theorem 4.1

Using Proposition E.1, we can deduce our lower bound in the context of $C^k$-spaces. The proof is in fact almost identical to what is done in [19, Theorem 4.2]. However, we decided to include a detailed proof in this paper, since [19] considers Sobolev functions and not $C^k$-functions.

**Theorem E.2.** *Let $s, k \in \mathbb{N}$. Then there exists a constant $c = c(s, k) > 0$ with the following property: For any $m \in \mathbb{N}$ and any map $\bar{a} : C^k([-1,1]^s; \mathbb{R}) \to \mathbb{R}^m$ that is continuous with respect to some norm on $C^k([-1,1]^s; \mathbb{R})$ and any (possibly discontinuous) map $M : \mathbb{R}^m \to C([-1,1]^s; \mathbb{R})$, we have*

$$\sup_{\substack{f \in C^k([-1,1]^s;\mathbb{R}) \\ \|f\|_{C^k([-1,1]^s;\mathbb{R})} \leq 1}} \|f - M(\bar{a}(f))\|_{L^\infty([-1,1]^s;\mathbb{R})} \geq c \cdot m^{-k/s}.$$

*Proof.* The idea is to apply Proposition E.1 to $X := C([-1,1]^s; \mathbb{R})$, $V := C^k([-1,1]^s; \mathbb{R})$ and the set $K := \{f \in C^k([-1,1]^s; \mathbb{R}) : \|f\|_{C^k([-1,1]^s;\mathbb{R})} \leq 1\}$.

Assume in the beginning that $m = n^s$ with an integer $n > 1$. Pick $\phi \in C^\infty(\mathbb{R}^s)$ with $\phi \equiv 1$ on $[-3/4, 3/4]^s$ and $\phi \equiv 0$ outside of $[-1,1]^s$. Fix $c_0 = c_0(s,k) > 0$ with

$$1 \leq \|\phi\|_{C^k([-1,1]^s;\mathbb{R})} \leq c_0.$$

Let $Q_1, ..., Q_m$ be the partition (disjoint up to null-sets) of $[-1,1]^s$ into closed cubes of sidelength $2/n$. For every $j \in \{1, ..., m\}$ we write $Q_j = \bigtimes_{\ell=1}^s [a_\ell^{(j)} - 1/n, a_\ell^{(j)} + 1/n]$ with an appropriately chosen vector $a = (a_1^{(j)}, ..., a_s^{(j)}) \in [-1,1]^s$ and let

$$\phi_j(x) := \phi(n(x - a^{(j)})) \text{ for } x \in \mathbb{R}^s.$$

By choice of $\phi$, the maps $\phi_j$ are supported on a proper subset of $Q_j$ for every $j \in \{1, ..., m\}$ and an inductive argument shows

$$\partial^{\mathbf{k}}\phi_j(x) = n^{|\mathbf{k}|} \cdot (\partial^{\mathbf{k}}\phi)(n(x - a^{(j)})) \quad \text{for every } \mathbf{k} \in \mathbb{N}_0^s \text{ and } x \in \mathbb{R}^s$$

and hence in particular

$$\|\phi_j\|_{C^k([-1,1]^s;\mathbb{R})} \leq n^{|\mathbf{k}|} \cdot c_0. \tag{E.2}$$

Let $X_m := \text{span}\{\phi_1, ..., \phi_m\}$ and $S \in U_1(X_m) = \{f \in X_m : \|f\|_{L^\infty([-1,1]^s;\mathbb{R})} \leq 1\}$. Then we can write $S$ in the form $S = \sum_{j=1}^m c_j\phi_j$ with real numbers $c_1, ..., c_m \in \mathbb{R}$. Suppose there exists $j^* \in \{1, ..., m\}$ with $|c_{j^*}| > 1$. Then we have

$$\|S\|_{L^\infty([-1,1]^s;\mathbb{R})} \geq |S(a^{(j^*)})| \geq |c_{j^*}| > 1,$$

since the functions $\phi_j$ have disjoint support and $\phi_j(a^{(j)}) = 1$. This is a contradiction to $S \in U_1(X_m)$ and we can thus infer that $\max_j |c_j| \leq 1$. Furthermore, we see again because the functions $\phi_j$ have disjoint support that

$$\|\partial^{\mathbf{k}}S\|_{L^\infty([-1,1]^s;\mathbb{R})} \leq \max_j |c_j| \cdot \|\partial^{\mathbf{k}}\phi_j\|_{L^\infty([-1,1]^s;\mathbb{R})} \overset{(E.2)}{\leq} n^{|\mathbf{k}|} \cdot c_0 \leq c_0 \cdot n^k = c_0 \cdot m^{k/s}$$

for every $\mathbf{k} \in \mathbb{N}_0^s$ with $|\mathbf{k}| \leq k$ and hence

$$\|S\|_{C^k([-1,1]^s;\mathbb{R})} \leq c_0 \cdot m^{k/s}.$$

Thus, letting $\varrho := c_0^{-1} \cdot m^{-k/s}$ yields $U_\varrho(X_m) \subseteq K$, so we see by Proposition E.1 that

$$\sup_{f \in K} \|f - M_{m-1}(\overline{a}(f))\|_{L^\infty([-1,1]^s;\mathbb{R})} \geq \varrho = c_1 \cdot m^{-k/s}$$

with $c_1 = c_0^{-1}$ for every map $\overline{a} : X \to \mathbb{R}^{m-1}$ which is continuous with respect to some norm on $V$ and any map $M_{m-1} : \mathbb{R}^{m-1} \to X$. Using the inequality $m \leq 2(m-1)$ (note $m > 1$) we get

$$\sup_{f \in K} \|f - M_{m-1}(\overline{a}(f))\|_{L^\infty([-1,1]^s;\mathbb{R})} \geq c_1 \cdot m^{-k/s} \geq c_1 \cdot (2(m-1))^{-k/s} \geq c_2 \cdot (m-1)^{-k/s}$$

with $c_2 = c_1 \cdot 2^{-k/s}$. Hence, the claim has been shown for all numbers $m$ of the form $n^s - 1$ with an integer $n > 1$.

In the end, let $m \in \mathbb{N}$ be arbitrary and pick $n \in \mathbb{N}$ with $n^s \leq m < (n+1)^s$. For given maps $\overline{a} : V \to \mathbb{R}^m$ and $M : \mathbb{R}^m \to X$ with $\overline{a}$ continuous with respect to some norm on $V$, let

$$\tilde{a} : \quad V \to \mathbb{R}^{(n+1)^s - 1}, \quad f \mapsto (\overline{a}(f), 0) \quad \text{and}$$

$$M_{(n+1)^s-1} : \quad \mathbb{R}^{(n+1)^s - 1} \to X, \quad (x,y) \mapsto M(x),$$

where $x \in \mathbb{R}^m$, $y \in \mathbb{R}^{(n+1)^s - 1 - m}$. Then we get

$$\sup_{f \in K} \|f - M(\overline{a}(f))\|_{L^\infty([-1,1]^s;\mathbb{R})} = \sup_{f \in K} \|f - M_{(n+1)^s-1}(\tilde{a}(f))\|_{L^\infty([-1,1]^s;\mathbb{R})}$$

$$\geq c_2 \cdot ((n+1)^s - 1)^{-k/s} \geq c_2 \cdot (2^s n^s)^{-k/s} \geq c_3 \cdot m^{-k/s}$$

with $c_3 = c_2 \cdot 2^{-k}$. Here we used the bound $(n+1)^s - 1 \leq (2n)^s$. This proves the full claim. $\quad\square$

Using this theorem, we can now prove Theorem 4.1.

*Proof of Theorem 4.1.* Let $\overline{a} : C^k(\Omega_n; \mathbb{C}) \to \mathbb{C}^m$ be any map that is continuous with respect to some norm $\|\cdot\|_V$ on $C^k(\Omega_n; \mathbb{C})$, and let $M : \mathbb{C}^m \to C(\Omega_n; \mathbb{C})$ be arbitrary. With $\varphi_n, \varphi_m$ defined as in Equation (A.1), let

$$\tilde{a} : \quad C^k([-1,1]^{2n}; \mathbb{R}) \to \mathbb{R}^{2m}, \quad \tilde{f} \mapsto \varphi_m^{-1}\left(\overline{a}\left(\tilde{f} \circ \varphi_n^{-1}\big|_{\Omega_n}\right)\right).$$

Clearly, $\tilde{a}$ is continuous on $C^k([-1,1]^{2n}; \mathbb{R})$ with respect to the norm $\|\cdot\|_{\tilde{V}}$ on $C^k([-1,1]^{2n}; \mathbb{R})$ defined as

$$\|\tilde{f}\|_{\tilde{V}} := \left\|\tilde{f} \circ \varphi_n^{-1}\big|_{\Omega_n}\right\|_V \quad \text{for } \tilde{f} \in C^k([-1,1]^{2n}; \mathbb{R}).$$

Let

$$\widetilde{M} : \quad \mathbb{R}^{2m} \to C([-1,1]^{2n}; \mathbb{R}), \quad \widetilde{M}(x) := \mathrm{Re}(M(\varphi_m(x))) \circ \varphi_n\big|_{[-1,1]^{2n}}.$$

Then it holds

$$\sup_{\substack{f \in C^k(\Omega_n;\mathbb{C}) \\ \|f\|_{C^k(\Omega_n;\mathbb{C})} \leq 1}} \|f - M(\overline{a}(f))\|_{L^\infty(\Omega_n;\mathbb{C})}$$

$$\geq \sup_{\substack{f \in C^k(\Omega_n;\mathbb{R}) \\ \|f\|_{C^k(\Omega_n;\mathbb{R})} \leq 1}} \|f - \mathrm{Re}(M(\overline{a}(f)))\|_{L^\infty(\Omega_n;\mathbb{R})}$$

$$= \sup_{\substack{\tilde{f} \in C^k([-1,1]^{2n};\mathbb{R}) \\ \|\tilde{f}\|_{C^k([-1,1]^{2n};\mathbb{R})} \leq 1}} \left\|\tilde{f} \circ \varphi_n^{-1} - \mathrm{Re}\left(M\left(\overline{a}\left(\tilde{f} \circ \varphi_n^{-1}\big|_{\Omega_n}\right)\right)\right)\right\|_{L^\infty(\Omega_n;\mathbb{R})}$$

$$= \sup_{\substack{\tilde{f} \in C^k([-1,1]^{2n};\mathbb{R}) \\ \|\tilde{f}\|_{C^k([-1,1]^{2n};\mathbb{R})} \leq 1}} \left\|\tilde{f} - \mathrm{Re}\left(M\left(\varphi_m\left(\varphi_m^{-1}\left(\overline{a}\left(\tilde{f} \circ \varphi_n^{-1}\big|_{\Omega_n}\right)\right)\right)\right)\right) \circ \varphi_n\right\|_{L^\infty([-1,1]^{2n};\mathbb{R})}$$

$$= \sup_{\substack{\tilde{f} \in C^k([-1,1]^{2n};\mathbb{R}) \\ \|\tilde{f}\|_{C^k([-1,1]^{2n};\mathbb{R})} \leq 1}} \left\|\tilde{f} - \widetilde{M}(\tilde{a}(\tilde{f}))\right\|_{L^\infty([-1,1]^{2n};\mathbb{R})} \geq \tilde{c} \cdot (2m)^{-k/(2n)},$$

with a constant $\tilde{c} = \tilde{c}(n,k)$ provided by Theorem E.2. Hence, the claim follows by choosing $c = c(n,k) := 2^{-k/(2n)} \cdot \tilde{c}$. $\quad\square$

As a corollary, we formulate a special case of Theorem 4.1 for the case of shallow complex-valued neural networks.

**Corollary E.3.** *Let $n, k \in \mathbb{N}$. Then there exists a constant $c = c(n, k) > 0$ with the following property: For any $m \in \mathbb{N}$, $\phi \in C(\mathbb{C}; \mathbb{C})$ and any map*

$$\eta : \quad C^k(\Omega_n; \mathbb{C}) \to (\mathbb{C}^n)^m \times \mathbb{C}^m \times \mathbb{C}^m, \quad g \mapsto (\eta_1(g), \eta_2(g), \eta_3(g))$$

*which is continuous with respect to some norm on $C^k(\Omega_n; \mathbb{C})$, there exists $f \in C^k(\Omega_n; \mathbb{C})$ satisfying $\|f\|_{C^k(\Omega_n; \mathbb{C})} \leq 1$ and*

$$\|f - \Psi(f)\|_{L^\infty(\Omega_n; \mathbb{C})} \geq c \cdot m^{-k/(2n)},$$

*where $\Psi(f) \in C(\Omega_n; \mathbb{C})$ is given by*

$$\Psi(f)(z) := \sum_{j=1}^m (\eta_3(f))_j \, \phi\left([\eta_1(f)]_j^T z + (\eta_2(f))_j\right).$$

*Proof.* Using Theorem 4.1, we deduce that there exists $f \in C^k(\Omega_n; \mathbb{C})$ satisfying $\|f\|_{C^k(\Omega_n; \mathbb{C})} \leq 1$ and

$$\|f - \Psi(f)\|_{L^\infty(\Omega_n; \mathbb{C})} \geq c' \cdot (m(n+2))^{-k/(2n)}$$

for a constant $c' = c'(n, k) > 0$. Hence, the claim follows by letting $c := c' \cdot (n+2)^{-k/(2n)}$. $\quad\square$

## E.2  Proof of Theorem 5.1

We can use Proposition E.1 not only to show that the rate of convergence established in this paper is optimal (which is done in Appendix E.1) but also to show that the problem of approximating $C^k$-functions using a set of functions that can be parametrized with finitely many parameters is intractable in the sense that it suffers from the curse of dimensionality, provided that the map which assigns to each $C^k$-function the parameters of the approximating function is continuous. This is the subject of this section.

In [35] a certain space of polynomials was used to show the intractability in the case of *linear* approximation methods. We are also going to use this class of polynomials, but combine it with Proposition E.1 to infer intractability in the case of *continuous* approximation methods. We start with a lemma discussing an important property of this space of polynomials. This property is stated as part of a proof in [35], but no complete proof is provided.

**Lemma E.4.** *Let $s \in \mathbb{N}$ and consider a function $f \in C^\infty([-1, 1]^s; \mathbb{R})$ which is given via*

$$f(x) = \sum_{\boldsymbol{k} \in \{0,1\}^s} a_{\boldsymbol{k}} x^{\boldsymbol{k}} \tag{E.3}$$

*with coefficients $a_{\boldsymbol{k}} \in \mathbb{R}$ for every $\boldsymbol{k} \in \{0, 1\}^s$. Then it holds*

$$\|f\|_{C^k([-1,1]^s; \mathbb{R})} = \|f\|_{L^\infty([-1,1]^s; \mathbb{R})}$$

*for every $k \in \mathbb{N}$.*

*Proof.* The proof is by induction over $s$. We start with the case $s = 1$ and note that we can write $f(x) = ax + b$ with $a, b \in \mathbb{R}$ in that case. Switching to $-f$ if necessary, we can assume $a \geq 0$. Clearly, $\|f\|_{L^\infty([-1,1]; \mathbb{R})} \leq |a| + |b|$. Conversely, if $b \geq 0$ then $|f(1)| = |a + b| = |a| + |b|$. If otherwise $b < 0$ then $|f(-1)| = |b - a| = |a - b| = |a| + |b|$. Thus, $\|f\|_{L^\infty([-1,1]; \mathbb{R})} = |a| + |b|$. For the derivatives, we have $\|f'\|_{L^\infty([-1,1]; \mathbb{R})} = |a|$ and $\|f^{(k)}\|_{L^\infty([-1,1]; \mathbb{R})} = 0$ for $k \geq 2$. This proves the claim in the case $s = 1$.

We now assume that the claim holds for some arbitrary but fixed $s \in \mathbb{N}$. We further let $\alpha \in \mathbb{N}_0^{s+1}$ and fix a point $(x_1, ..., x_{s+1}) \in [-1, 1]^{s+1}$. We decompose $\alpha = (\alpha', \alpha_{s+1})$ with $\alpha' \in \mathbb{N}_0^s$. Let

$$\widetilde{f} : \quad [-1, 1] \to \mathbb{R}, \quad y_{s+1} \mapsto \partial^{(\alpha', 0)} f(x_1, ..., x_s, y_{s+1})$$

and note

$$\partial^\alpha f(x_1, ..., x_{s+1}) = \widetilde{f}^{(\alpha_{s+1})}(x_{s+1}).$$

Note that $f$ is affine-linear with respect to each variable (with all other variables hold fixed). Hence, $\widetilde{f}$ is an affine function and we can thus apply the case $s = 1$ to $\widetilde{f}$ and get

$$\|\widetilde{f}^{(\alpha_{s+1})}\|_{L^\infty([-1,1];\mathbb{R})} \le \|\widetilde{f}\|_{L^\infty([-1,1];\mathbb{R})}.$$

Putting this together, we infer

$$|\partial^\alpha f(x_1, ..., x_{s+1})| \le \|\widetilde{f}\|_{L^\infty([-1,1];\mathbb{R})} = \sup_{y_{s+1}\in[-1,1]} |\partial^{(\alpha',0)} f(x_1, ..., x_s, y_{s+1})|. \tag{E.4}$$

We now *fix* an arbitrary point $y_{s+1} \in [-1,1]$ and consider

$$\widehat{f}: \quad [-1,1]^s \to \mathbb{R}, \quad (y_1, ..., y_s) \mapsto f(y_1, ..., y_s, y_{s+1}).$$

Then it holds

$$\partial^{(\alpha',0)} f(x_1, ..., x_s, y_{s+1}) = \partial^{\alpha'} \widehat{f}(x_1, ..., x_s).$$

Applying the induction hypothesis to $\widehat{f}$ (which is easily seen to be of the form (E.3)) we get

$$|\partial^{\alpha'} \widehat{f}(x_1, ..., x_s)| \le \|\widehat{f}\|_{L^\infty([-1,1]^s;\mathbb{R})} \le \|f\|_{L^\infty([-1,1]^{s+1};\mathbb{R})}. \tag{E.5}$$

Combining (E.4) and (E.5) yields

$$|\partial^\alpha f(x_1, ..., x_{s+1})| \le \|f\|_{L^\infty([-1,1]^{s+1};\mathbb{R})}.$$

Since $\alpha \in \mathbb{N}_0^{s+1}$ was arbitrary, we get the claim by noting that

$$\|f\|_{C^k([-1,1]^s;\mathbb{R})} \ge \|f\|_{L^\infty([-1,1]^s;\mathbb{R})}$$

holds trivially for every $k \in \mathbb{N}$. $\qquad\qquad\qquad\qquad\qquad\qquad\qquad\qquad\qquad\qquad\qquad\quad\square$

Using the above lemma, we can now deduce that the approximation of smooth functions using continuous approximation methods is intractable in terms of the input dimension.

*Proof of Theorem 5.1.* We apply Proposition E.1 to $X := C([-1,1]^s;\mathbb{R})$, $V := C^{\infty,*,s}$ and to the set $K := \{f \in C^{\infty,*,s} : \|f\|_{C^\infty([-1,1]^s;\mathbb{R})} \le 1\}$ and $m := 2^s - 1$. The space

$$X_{m+1} := \left\{ [-1,1]^s \ni x \mapsto \sum_{\mathbf{k}\in\{0,1\}^s} a_{\mathbf{k}} x^{\mathbf{k}} : a_{\mathbf{k}} \in \mathbb{R} \right\}$$

consisting of all functions considered in the previous Lemma E.4 is an $(m+1)$-dimensional subspace of $C([-1,1]^s;\mathbb{R})$. For every $f \in X_{m+1}$ with $\|f\|_{L^\infty([-1,1]^s;\mathbb{R})} \le 1$, Lemma E.4 tells us $\|f\|_{C^\infty([-1,1]^s;\mathbb{R})} \le 1$. Hence, $U_1(X_{m+1}) \subseteq K$ and Proposition E.1 then yields the claim. $\quad\square$

*Remark* E.5. The statement of Theorem 5.1 also holds if the functions satisfy $\overline{a} : C^{\infty,*,s} \to \mathbb{R}^m$ and $M : \mathbb{R}^m \to C([-1,1]^s;\mathbb{R})$ with $m \le 2^s - 1$. This can be seen by defining

$$\tilde{a}: \quad C^{\infty,*,s} \to \mathbb{R}^{2^s-1}, \quad f \mapsto (\overline{a}(f), 0, ..., 0)$$

and

$$\widetilde{M}: \quad \mathbb{R}^{2^s-1} \to C([-1,1]^s;\mathbb{R}), \quad (a,b) \mapsto M(a)$$

with $a \in \mathbb{R}^m$ and $b \in \mathbb{R}^{2^s-1-m}$.

The following Corollary E.6 transfers Theorem 5.1 to the complex-valued setting.

**Corollary E.6.** *Let $n \in \mathbb{N}$. For any function $f \in C^\infty(\Omega_n;\mathbb{C})$ we write*

$$\|f\|_{C^\infty(\Omega_n;\mathbb{C})} := \sup_{k\in\mathbb{N}} \|f\|_{C^k(\Omega_n;\mathbb{C})}$$

*and let $C_\mathbb{C}^{\infty,*,n}$ denote the space consisting of all functions for which this expression is finite. Let $\overline{a} : C_\mathbb{C}^{\infty,*,n} \to \mathbb{C}^{2^{2n-1}-1}$ be continuous with respect to some norm on $C_\mathbb{C}^{\infty,*,n}$ and moreover, let $M : \mathbb{C}^{2^{2n-1}-1} \to C(\Omega_n;\mathbb{C})$ be an arbitrary map. Then it holds*

$$\sup_{\substack{f\in C_\mathbb{C}^{\infty,*,n} \\ \|f\|_{C^\infty(\Omega_n;\mathbb{C})}\le 1}} \|f - M(\overline{a}(f))\|_{L^\infty(\Omega_n;\mathbb{C})} \ge 1.$$

*Proof.* The transfer to the complex-valued setting works in the same manner as the proof of Theorem 4.1 (see Appendix E.1). We write $m := 2^{2n-1} - 1$ and note $2m = 2^{2n} - 2 \leq 2^{2n} - 1$. We define $\tilde{a} : C^{\infty,*,2n} \to \mathbb{R}^{2m}$ and $\widetilde{M} : \mathbb{R}^{2m} \to C([-1,1]^{2n}; \mathbb{R})$ in the same way as in the proof of Theorem 4.1. Using again the same technique as in the proof of Theorem 4.1, we get

$$\sup_{\substack{f \in C_{\mathbb{C}}^{\infty,*,n} \\ \|f\|_{C^\infty(\Omega_n;\mathbb{C})} \leq 1}} \|f - M(\overline{a}(f))\|_{L^\infty(\Omega_n;\mathbb{C})} \geq \sup_{\substack{\tilde{f} \in C^{\infty,*,2n} \\ \|\tilde{f}\|_{C^\infty([-1,1]^{2n};\mathbb{R})} \leq 1}} \left\| \tilde{f} - \widetilde{M}(\tilde{a}(\tilde{f})) \right\|_{L^\infty([-1,1]^{2n};\mathbb{R})} \geq 1,$$

applying Theorem 5.1 in the last inequality, using $2m \leq 2^{2n} - 1$. $\qquad\square$

We conclude this appendix by adding a note on the constant appearing in our main approximation bound.

**Corollary E.7.** *Let $n \in \mathbb{N}$ with $n \geq 2$ and $\alpha > 0$ and let $\phi \in C(\mathbb{C}; \mathbb{C})$. Let $\tilde{c} = \tilde{c}(n, \alpha) > 0$ be such that for every $m \in \mathbb{N}$ there exists a mapping*

$$\eta : \quad C_{\mathbb{C}}^{\infty,*,n} \to (\mathbb{C}^n)^m \times \mathbb{C}^m \times \mathbb{C}^m, \quad g \mapsto (\eta_1(g), \eta_2(g), \eta_3(g))$$

*that is continuous with respect to any norm on $C_{\mathbb{C}}^{\infty,*,n}$ and such that*

$$\|f - \Psi(f)\|_{L^\infty(\Omega_n;\mathbb{C})} \leq (\tilde{c} \cdot m)^{-\alpha} \cdot \|f\|_{C^\infty(\Omega_n;\mathbb{C})},$$

*for every $f \in C_{\mathbb{C}}^{\infty,*,n}$. Here, $\Psi(f) \in C(\Omega_n; \mathbb{C})$ is given by*

$$\Psi(f)(z) := \sum_{j=1}^{m} (\eta_3(f))_j \, \phi \left( [\eta_1(f)]_j^T z + (\eta_2(f))_j \right).$$

*Then it necessarily holds $\tilde{c} \leq 16 \cdot 2^{-n}$.*

*Proof.* We first assume $n \geq 4$. We take $m = \left\lfloor \frac{2^{2n-1}-1}{n+2} \right\rfloor$ and note that then $m(n+2) \leq 2^{2n-1} - 1$. Therefore, Corollary E.6 applies and we infer that for each $\varepsilon \in (0,1)$, there exists $f = f_\varepsilon \in C_{\mathbb{C}}^{\infty,*,n}$ with $\|f\|_{C^\infty(\Omega_n;\mathbb{C})} \leq 1$ and such that

$$1 - \varepsilon \leq \|f - \Psi(f)\|_{L^\infty(\Omega_n;\mathbb{C})} \leq (\tilde{c} \cdot m)^{-\alpha} \cdot \|f\|_{C^\infty(\Omega_n;\mathbb{C})} \leq (\tilde{c} \cdot m)^{-\alpha}.$$

This then necessarily implies $\tilde{c} \cdot m \leq 1$ or equivalently $\tilde{c} \leq 1/m$. It therefore suffices to derive a lower bound for $m$. Firstly, we note

$$2^{2n-1} = 2^{n-3} \cdot 2^{n+2} = 2^{n-3} \cdot (1+1)^{n+2} \geq 2^{n-3}(n+3),$$

where we applied Bernoulli's inequality. Because of $n \geq 4 \geq 3$, this yields

$$2^{2n-1} - 1 \geq 2^{n-3}(n+3) - 2^{n-3} = 2^{n-3}(n+2).$$

Hence, we get

$$m \geq \frac{2^{2n-1}-1}{n+2} - 1 \geq 2^{n-3} - 1 = 2^{n-3}(1 - 2^{3-n}) \geq 2^{n-4} = \frac{2^n}{16}.$$

Here, we used $n \geq 4$ in the last inequality. An explicit computation shows that the same bounds also holds in the cases $n = 2$ and $n = 3$. This proves the claim. $\qquad\square$

# F   Postponed proofs for the optimality results in the case of unrestricted weight selection

## F.1   Approximation using Ridge Functions

In this section we prove for $s \in \mathbb{N}_{\geq 2}$ that every function in $C^k([-1,1]^s; \mathbb{R})$ can be uniformly approximated with an error of the order $m^{-k/(s-1)}$ using a linear combination of $m$ so-called *ridge functions*. In fact, we only consider ridge *polynomials*, meaning functions of the form

$$\mathbb{R}^s \to \mathbb{R}, \quad x \mapsto p(a^T x)$$

for a fixed vector $a \in \mathbb{R}^s$ and a polynomial $p : \mathbb{R} \to \mathbb{R}$. Note that this result has already been obtained in a slightly different form in [30, Theorem 1]; namely, it is shown there that the rate of approximation $m^{-k/(s-1)}$ can be achieved by functions of the form $\sum_{j=1}^m f_j(a_j^T x)$ with $a_j \in \mathbb{R}^s$ and $f_j \in L^1_{\text{loc}}(\mathbb{R}^s)$. We will need the fact that the $f_j$ can actually be chosen as polynomials and that the vectors $a_1, ..., a_m$ can be chosen independently from the particular function $f$. This is shown in the proof of [30], but not stated explicitly. For this reason, and in order to clarify the proof itself and to make the paper more self-contained, we decided to present the proof in this appendix.

**Lemma F.1.** *Let $m, s \in \mathbb{N}$. Then we denote by*

$$
P_m^s := \left\{ \mathbb{R}^s \to \mathbb{R}, \quad x \mapsto \sum_{\substack{\boldsymbol{k} \in \mathbb{N}_0^s \\ |\boldsymbol{k}| \leq m}} a_{\boldsymbol{k}} x^{\boldsymbol{k}} : a_{\boldsymbol{k}} \in \mathbb{R} \right\}
$$

*the set of real polynomials of degree at most $m$. The subset of* homogeneous *polynomials of degree $m$ is defined as*

$$
H_m^s := \left\{ \mathbb{R}^s \to \mathbb{R}, \quad x \mapsto \sum_{\substack{\boldsymbol{k} \in \mathbb{N}_0^s \\ |\boldsymbol{k}| = m}} a_{\boldsymbol{k}} x^{\boldsymbol{k}} : a_{\boldsymbol{k}} \in \mathbb{R} \right\}.
$$

*Then there exists a constant $c = c(s) > 0$ satisfying*

$$
\dim(H_m^s) \leq c \cdot m^{s-1} \quad \forall m \in \mathbb{N}.
$$

*Proof.* It is immediate that the set

$$
\left\{ \mathbb{R}^s \to \mathbb{R}, \ x \mapsto x^{\mathbf{k}} : \mathbf{k} \in \mathbb{N}_0^s, \ |\mathbf{k}| = m \right\}
$$

forms a basis of $H_m^s$, hence

$$
\dim(H_m^s) = \# \left\{ \mathbf{k} \in \mathbb{N}_0^s : |\mathbf{k}| = m \right\}.
$$

This quantity clearly equals the number of possibilities for drawing $m$ times from a set with $s$ elements with replacement. Hence, we see

$$
\dim(H_m^s) = \binom{s + m - 1}{m},
$$

see for instance [12, Identity 143]. A further estimation shows

$$
\binom{s + m - 1}{m} = \prod_{j=1}^{s-1} \frac{m + j}{j} = \prod_{j=1}^{s-1} \left( 1 + \frac{m}{j} \right) \leq (1 + m)^{s-1} \leq 2^{s-1} \cdot m^{s-1}.
$$

Hence, the claim follows with $c(s) = 2^{s-1}$. $\qquad\qquad\square$

A combination of results from [36] together with the fact that it is possible to approximate $C^k$-functions using polynomials of degree at most $m$ with an error of the order $m^{-k}$, as shown in Theorem C.15, yields the desired result.

**Theorem F.2.** *Let $s, k \in \mathbb{N}$ with $s \geq 2$ and $r > 0$. Then there exists a constant $c = c(s, k) > 0$ with the following property: For every $m \in \mathbb{N}$ there exist $a_1, ..., a_m \in \mathbb{R}^s \setminus \{0\}$ with $\|a_j\|_2 = r$, such that for every function $f \in C^k([-1, 1]^s; \mathbb{R})$ there exist polynomials $p_1, ..., p_m \in P_m^1$ satisfying*

$$
\left\| f(x) - \sum_{j=1}^m p_j(a_j^T x) \right\|_{L^\infty([-1,1]^s;\mathbb{R})} \leq c \cdot m^{-k/(s-1)} \cdot \|f\|_{C^k([-1,1]^s;\mathbb{R})}.
$$

*Proof.* We first pick the constant $c_1 = c_1(s)$ according to Lemma F.1. Then we define the constant $c_2 = c_2(s) := (2s)^{s-1} \cdot c_1(s)$ and let $M \in \mathbb{N}$ be the largest integer satisfying

$$c_2 \cdot M^{s-1} \leq m.$$

Here, we assume without loss of generality that $m \geq c_2$, which can be justified by choosing $p_j = 0$ for every $j \in \{1, ..., m\}$ if $m < c_2$, at the cost of possibly enlarging $c$. Note that the choice of $M$ implies $c_2 \cdot (2M)^{s-1} \geq c_2 \cdot (M+1)^{s-1} > m$, and thus

$$M \geq \frac{1}{2} \cdot c_2^{-1/(s-1)} \cdot m^{1/(s-1)} = c_3 \cdot m^{1/(s-1)} \tag{F.1}$$

with $c_3 = c_3(s) := 1/2 \cdot c_2^{-1/(s-1)}$.

Using [36, Proposition 5.9] and Lemma F.1 we can pick $a_1, ..., a_m \in \mathbb{R}^s \setminus \{0\}$ satisfying

$$H^s_{s(2M-1)} = \text{span} \left\{ x \mapsto (a_j^T x)^{s(2M-1)} : j \in \{1, ..., m\} \right\}, \tag{F.2}$$

where we used that

$$c_1 \cdot (s(2M-1))^{s-1} \leq c_1 \cdot (2s)^{s-1} \cdot M^{s-1} = c_2 \cdot M^{s-1} \leq m.$$

Here we can assume $\|a_j\|_2 = r$ for every $j \in \{1, ..., m\}$ since multiplying each $a_j$ with a positive constant does not change the span in (F.2). From [36, Corollary 5.12] we infer that

$$P^s_{s(2M-1)} = \text{span} \left\{ x \mapsto (a_j^T x)^r : j \in \{1, ..., m\}, \ 0 \leq r \leq s(2M-1) \right\}. \tag{F.3}$$

Let $f \in C^k([-1,1]^s; \mathbb{R})$. Then, according to Theorem C.15, there exists a polynomial $P : \mathbb{R}^s \to \mathbb{R}$ of *coordinatewise* degree at most $2M - 1$ satisfying

$$\|f - P\|_{L^\infty([-1,1]^s;\mathbb{R})} \leq c_4 \cdot M^{-k} \cdot \|f\|_{C^k([-1,1]^s;\mathbb{R})},$$

where $c_4 = c_4(s, k) > 0$. Note that by construction it holds $P \in P^s_{s(2M-1)}$. Using (F.3) we deduce the existence of polynomials $p_1, ..., p_m : \mathbb{R} \to \mathbb{R}$ such that

$$P(x) = \sum_{j=1}^m p_j(a_j^T x) \quad \text{for all } x \in \mathbb{R}^s.$$

Combining the previously shown bounds, we get

$$\left\| f(x) - \sum_{j=1}^m p_j(a_j^T x) \right\|_{L^\infty([-1,1]^s;\mathbb{R})} = \|f(x) - P(x)\|_{L^\infty([-1,1]^s;\mathbb{R})} \leq c_4 \cdot M^{-k} \cdot \|f\|_{C^k([-1,1]^s;\mathbb{R})}$$

$$\overset{(F.1)}{\leq} c \cdot m^{-k/(s-1)} \cdot \|f\|_{C^k([-1,1]^s;\mathbb{R})},$$

as desired. Here, we defined $c = c(s, k) := c_4 \cdot c_3^{-k}$. $\qquad\square$

## F.2 Proof of Theorem 4.2

Using Theorem F.2, we can prove the following statement for complex-valued $C^k$-functions, which will play an important role in the proof of Theorem 4.2.

**Proposition F.3.** *Let $n, k \in \mathbb{N}$. Then there exists a constant $c = c(n, k) > 0$ with the following property: For any $m \in \mathbb{N}$ there exist complex vectors $b_1, ..., b_m \in \mathbb{C}^n$ with $\|b_j\|_2 = 1/\sqrt{2n}$ for $j = 1, ..., m$ and with the property that for any function $f \in C^k(\Omega_n; \mathbb{C})$ there exist functions $g_1, ..., g_m \in C(\Omega_1; \mathbb{C})$ such that*

$$\left\| f(z) - \sum_{j=1}^m g_j \left( b_j^T \cdot z \right) \right\|_{L^\infty(\Omega_n;\mathbb{C})} \leq c \cdot m^{-k/(2n-1)} \cdot \|f\|_{C^k(\Omega_n;\mathbb{C})} \cdot$$

*Note that the vectors $b_1, ...b_m$ can be chosen independently from the considered function $f$, whereas $g_1, ..., g_m$ do depend on $f$.*

*Proof.* Theorem F.2 yields the existence of a constant $c_1 = c_1(n, k) > 0$ with the property that for any $m \in \mathbb{N}$ there exist real vectors $a_1, ..., a_m \in \mathbb{R}^{2n}$ with $\|a_j\|_2 = 1/\sqrt{2n}$ such that for any function $\tilde{f} \in C^k\left([-1, 1]^{2n}; \mathbb{R}\right)$ there exist functions $\tilde{g}_1, ..., \tilde{g}_m \in C([-1, 1]; \mathbb{R})$ satisfying

$$\left\| \tilde{f}(x) - \sum_{j=1}^{m} \tilde{g}_j\left(a_j^T x\right) \right\|_{L^\infty([-1,1]^{2n};\mathbb{R})} \leq c_1 \cdot m^{-k/(2n-1)} \cdot \|\tilde{f}\|_{C^k([-1,1]^{2n};\mathbb{R})}.$$

We then define the vectors $b_1, ..., b_m \in \mathbb{C}^n$ componentwise via

$$(b_j)_\ell := (a_j)_\ell - i \cdot (a_j)_{n+\ell}, \quad \ell \in \{1, ..., n\}, \; j \in \{1, ..., m\}.$$

First we see $\|b_j\|_2 = \|a_j\|_2 = 1/\sqrt{2n}$. We first consider real-valued functions, i.e., $f \in C^k(\Omega_n; \mathbb{R})$. Let $\varphi_n$ be defined as in (A.1). By the choice of the constant $c_1$ we can find continuous functions $\tilde{g}_1, ..., \tilde{g}_m \in C([-1, 1]; \mathbb{R})$ such that

$$\left\| (f \circ \varphi_n)(x) - \sum_{j=1}^{m} \tilde{g}_j\left(a_j^T x\right) \right\|_{L^\infty([-1,1]^{2n})} \leq c_1 \cdot m^{-k/(2n-1)} \cdot \|f \circ \varphi_n\|_{C^k([-1,1]^{2n};\mathbb{R})}.$$

We then define $g_j \in C(\Omega_1; \mathbb{R})$ by $g_j(z) := \tilde{g}_j(\mathrm{Re}(z))$ for any $j \in \{1, ..., m\}$. For $z \in \Omega_n$ we then have

$$g_j\left((b_j)^T z\right) = \tilde{g}_j\left(\mathrm{Re}\left(\sum_{\ell=1}^{n} (b_j)_\ell \cdot z_\ell\right)\right)$$

$$= \tilde{g}_j\left(\mathrm{Re}\left(\sum_{\ell=1}^{n} \left((a_j)_\ell - i \cdot (a_j)_{n+\ell}\right)\left(\varphi_n^{-1}(z)_\ell + i \cdot \varphi_n^{-1}(z)_{n+\ell}\right)\right)\right)$$

$$= \tilde{g}_j\left(\sum_{\ell=1}^{n}\left[(a_j)_\ell\, \varphi_n^{-1}(z)_\ell + (a_j)_{n+\ell}\, \varphi_n^{-1}(z)_{n+\ell}\right]\right)$$

$$= \tilde{g}_j\left((a_j)^T \cdot \varphi_n^{-1}(z)\right). \tag{F.4}$$

Therefore,

$$\left\| f(z) - \sum_{j=1}^{m} g_j\left(b_j^T z\right) \right\|_{L^\infty(\Omega_n;\mathbb{R})} = \left\| (f \circ \varphi_n)(x) - \sum_{j=1}^{m} g_j\left(b_j^T \cdot \varphi_n(x)\right) \right\|_{L^\infty([-1,1]^{2n};\mathbb{R})}$$

$$\overset{(F.4)}{=} \left\| (f \circ \varphi_n)(x) - \sum_{j=1}^{m} \tilde{g}_j\left(a_j^T \cdot x\right) \right\|_{L^\infty([-1,1]^{2n};\mathbb{R})}$$

$$\leq c_1 \cdot m^{-k/(2n-1)} \cdot \|f \circ \varphi_n\|_{C^k([-1,1]^{2n};\mathbb{R})}.$$

By the above, for $f \in C^k(\Omega_n; \mathbb{C})$ we can pick functions $g_1^{\mathrm{Re}}, ..., g_m^{\mathrm{Re}}, g_1^{\mathrm{Im}}, ..., g_m^{\mathrm{Im}} \in C(\Omega_1; \mathbb{R})$ satisfying

$$\left\| \mathrm{Re}(f(z)) - \sum_{j=1}^{m} g_j^{\mathrm{Re}}\left(b_j^T z\right) \right\|_{L^\infty(\Omega_n;\mathbb{R})} \leq c_1 \cdot m^{-k/(2n-1)} \cdot \|\mathrm{Re}\left(f \circ \varphi_n\right)\|_{C^k([-1,1]^{2n};\mathbb{R})},$$

$$\left\| \mathrm{Im}(f(z)) - \sum_{j=1}^{m} g_j^{\mathrm{Im}}\left(b_j^T z\right) \right\|_{L^\infty(\Omega_n;\mathbb{R})} \leq c_1 \cdot m^{-k/(2n-1)} \cdot \|\mathrm{Im}\left(f \circ \varphi_n\right)\|_{C^k([-1,1]^{2n};\mathbb{R})}.$$

Defining $g_j := g_j^{\mathrm{Re}} + i \cdot g_j^{\mathrm{Im}}$ yields

$$\left\| f(z) - \sum_{j=1}^{m} g_j\left(b_j^T z\right) \right\|_{L^\infty(\Omega_n;\mathbb{C})} \leq c_1 \cdot \sqrt{2} \cdot m^{-k/(2n-1)} \cdot \|f\|_{C^k(\Omega_n;\mathbb{C})},$$

completing the proof. $\square$

The special activation function that yields the improved approximation rate of $m^{-k/(2n-1)}$ (see Theorem 4.2) is constructed in the following lemma.

**Lemma F.4.** *Let $\{u_\ell\}_{\ell=1}^\infty$ be an enumeration of the set of complex polynomials in $z$ and $\bar{z}$ with coefficients in $\mathbb{Q} + i\mathbb{Q}$. Then there exists a function $\phi \in C^\infty(\mathbb{C}; \mathbb{C})$ with the following properties:*

*1. For every $\ell \in \mathbb{N}$ and $z \in \Omega_1$ one has*

$$\phi(z + 3\ell) = u_\ell(z).$$

*2. $\phi$ is non-polyharmonic.*

*Proof.* Let $\psi \in C^\infty(\mathbb{C}; \mathbb{R})$ with $0 \le \psi \le 1$ and

$$\psi|_{\Omega_1} \equiv 1, \qquad \mathrm{supp}(\psi) \subseteq \widetilde{\Omega},$$

where $\widetilde{\Omega} := \left\{ z \in \mathbb{C} : |\mathrm{Re}\,(z)|, |\mathrm{Im}\,(z)| < \frac{3}{2} \right\}$. We then define

$$\phi := f \cdot \psi + \sum_{\ell=1}^\infty u_\ell(\bullet - 3\ell) \cdot \psi(\bullet - 3\ell),$$

where $f(z) = e^{\mathrm{Re}(z)}$. Note that $\phi$ is smooth since it is a locally finite sum of smooth functions. Furthermore, $\phi$ is non-polyharmonic on the interior of $\Omega_1$, since the calculation in the proof of Proposition A.2 shows for $z$ in the interior of $\Omega_1$ and $\rho : \mathbb{R} \to \mathbb{R}$, $t \mapsto e^t$ that

$$\left| \partial_{\mathrm{wirt}}^m \overline{\partial}_{\mathrm{wirt}}^\ell \phi(z) \right| = \left| \partial_{\mathrm{wirt}}^m \overline{\partial}_{\mathrm{wirt}}^\ell f(z) \right| = \frac{1}{2^{m+\ell}} \left| \rho^{(m+\ell)}(\mathrm{Re}(z)) \right| > 0$$

for arbitrary $m, \ell \in \mathbb{N}_0$. Finally, property (1) follows directly by construction of $\phi$ because

$$(\widetilde{\Omega} + 3\ell) \cap (\widetilde{\Omega} + 3\ell') = \varnothing$$

for $\ell \ne \ell'$. $\qquad\square$

Using the properties of the special activation function constructed in Lemma F.4 and applying the approximation result from Proposition F.3 we can now prove Theorem 4.2.

*Proof of Theorem 4.2.* Let $\phi$ be the activation function constructed in Lemma F.4. We choose the constant $c$ according to Proposition F.3. Let $m \in \mathbb{N}$ and $f \in C^k(\Omega_n; \mathbb{C})$. We can without loss of generality assume that $f \not\equiv 0$. Again, according to Proposition F.3, we can choose $\rho_1, ..., \rho_m \in \mathbb{C}^n$ with $\|_2\rho_j\| = 1/\sqrt{2n}$ and $g_1, ..., g_m \in C(\Omega; \mathbb{C})$ with the property

$$\left\| f(z) - \sum_{j=1}^m g_j \left( \rho_j^T z \right) \right\|_{L^\infty(\Omega_n)} \le c \cdot m^{-k/(2n-1)} \cdot \|f\|_{C^k(\Omega_n; \mathbb{C})}.$$

Recall from Lemma F.4 that $\{u_\ell\}_{\ell=1}^\infty$ is an enumeration of the set of complex polynomials in $z$ and $\bar{z}$. Hence, using the complex version of the Stone-Weierstraß-Theorem (see for instance [21, Theorem 4.51]), we can pick $\ell_1, ..., \ell_m \in \mathbb{N}$ such that

$$\left\| g_j - u_{\ell_j} \right\|_{L^\infty(\Omega_1; \mathbb{C})} \le m^{-1-k/(2n-1)} \cdot \|f\|_{C^k(\Omega_n; \mathbb{C})} \tag{F.5}$$

for every $j \in \{1, ..., m\}$. Since $\phi(\bullet + 3\ell) = u_\ell$ on $\Omega_1$ for each $\ell \in \mathbb{N}$, and since $\rho_j^T z \in \Omega_1$ for $j \in \{1, ..., m\}$ and $z \in \Omega_n$, we estimate

$$\left\| f(z) - \sum_{j=1}^m \phi\left(\rho_j^T z + 3\ell_j\right) \right\|_{L^\infty(\Omega_n; \mathbb{C})}$$

$$\leq \left\| f(z) - \sum_{j=1}^m g_j\left(\rho_j^T z\right) \right\|_{L^\infty(\Omega_n; \mathbb{C})} + \sum_{j=1}^m \left\| g_j\left(\rho_j^T z\right) - \phi\left(\rho_j^T \cdot z + 3\ell_j\right) \right\|_{L^\infty(\Omega_n; \mathbb{C})}$$

$$\leq c \cdot m^{-k/(2n-1)} \cdot \|f\|_{C^k(\Omega_n; \mathbb{C})} + \sum_{j=1}^m \left\| g_j(z) - u_{\ell_j}(z) \right\|_{L^\infty(\Omega_1; \mathbb{C})}$$

$$\overset{(\text{F.5})}{\leq} c \cdot m^{-k/(2n-1)} \cdot \|f\|_{C^k(\Omega_n; \mathbb{C})} + m^{-k/(2n-1)} \cdot \|f\|_{C^k(\Omega_n; \mathbb{C})}$$

$$= (c+1) \cdot m^{-k/(2n-1)} \cdot \|f\|_{C^k(\Omega_n; \mathbb{C})}. \qquad \qquad \square$$

## F.3  Proof of Theorem 4.3

As a preparation for the proof of Theorem 4.3, we first prove a similar result in the real-valued setting. We remark that the proof idea is inspired by the proof of [46, Theorem 4].

**Theorem F.5.** *Let $n, k \in \mathbb{N}$ and*

$$\phi: \quad \mathbb{R} \to \mathbb{R}, \quad \phi(x) := \frac{1}{1 + e^{-x}}$$

*be the sigmoid function. Then there exists a constant $c = c(n, k) > 0$ with the following property: If the numbers $\varepsilon \in (0, \frac{1}{2})$ and $m \in \mathbb{N}$ are such that for every function $f \in C^k\left([-1, 1]^n; \mathbb{R}\right)$ with $\|f\|_{C^k([-1,1]^n; \mathbb{R})} \leq 1$ there exist coefficients $\rho_1, ..., \rho_m \in \mathbb{R}^n$, $\eta_1, ..., \eta_m \in \mathbb{R}$ and $\sigma_1, ..., \sigma_m \in \mathbb{R}$ satisfying*

$$\left\| f(x) - \sum_{j=1}^m \sigma_j \cdot \phi\left(\rho_j^T x + \eta_j\right) \right\|_{L^\infty([-1,1]^n; \mathbb{R})} \leq \varepsilon,$$

*then necessarily*

$$m \geq c \cdot \frac{\varepsilon^{-n/k}}{\ln(1/\varepsilon)}.$$

*Proof.* We first pick a function $\psi \in C^\infty(\mathbb{R}^n; \mathbb{R})$ with the property that $\psi(0) = 1$ and $\psi(x) = 0$ for every $x \in \mathbb{R}^n$ with $\|x\|_2 > \frac{1}{4}$. We then choose

$$c_1 = c_1(n, k) := \left( \|\psi\|_{C^k([-1,1]^n; \mathbb{R})} \right)^{-1}.$$

Now, let $\varepsilon \in (0, \frac{1}{2})$ and $m \in \mathbb{N}$ be arbitrary with the property stated in the formulation of the theorem. If $\varepsilon > \frac{c_1}{2} \cdot \frac{1}{6^k}$, then $m \geq c \cdot \frac{\varepsilon^{-n/k}}{\ln(1/\varepsilon)}$ trivially holds (as long as $c = c(n, k) > 0$ is sufficiently small). Hence, we can assume that $\varepsilon \leq \frac{c_1}{2} \cdot \frac{1}{6^k}$. Now, let $N$ be the smallest integer with $N \geq 2$, for which

$$\frac{c_1}{2^{k+1}} \cdot N^{-k} \leq \varepsilon.$$

Note that this implies

$$N^k \geq \frac{c_1}{\varepsilon} \cdot \frac{1}{2^{k+1}} \geq \frac{c_1}{2^{k+1}} \cdot \frac{2}{c_1} \cdot 6^k = 3^k$$

and hence $N \geq 3$, whence $N - 1 \geq 2$. Therefore, by minimality of $N$, and since $\frac{N}{2} \leq N - 1$ because of $N \geq 2$, it follows that

$$\varepsilon < \frac{c_1}{2^{k+1}} \cdot (N-1)^{-k} \leq \frac{c_1}{2^{k+1}} 2^k \cdot N^{-k} = \frac{c_1}{2} \cdot N^{-k}. \tag{F.6}$$

Now, for every $\alpha \in \{-N, ..., N\}^n$ pick $z_\alpha \in \{0, 1\}$ arbitrary and let $y_\alpha := z_\alpha c_1 N^{-k}$. Define the function

$$f(x) := \sum_{\alpha \in \{-N,...,N\}^n} y_\alpha \cdot \psi(Nx - \alpha), \quad x \in \mathbb{R}^n.$$

Clearly, $f \in C^\infty(\mathbb{R}^n; \mathbb{R})$. Furthermore, since the supports of the functions $\psi(\bullet - \alpha)$, $\alpha \in \mathbb{Z}^n$ are pairwise disjoint, we see for any multi-index $\mathbf{k} \in \mathbb{N}_0^n$ with $|\mathbf{k}| \leq k$ that

$$\left\|\partial^{\mathbf{k}} f\right\|_{L^\infty([-1,1]^n;\mathbb{R})} \leq N^{|\mathbf{k}|} \cdot \max_\alpha |y_\alpha| \cdot \left\|\partial^{\mathbf{k}} \psi\right\|_{L^\infty([-1,1]^n;\mathbb{R})}$$

$$\leq N^k \cdot \max_\alpha |y_\alpha| \cdot \|\psi\|_{C^k([-1,1]^n;\mathbb{R})} \leq 1,$$

so we conclude that $\|f\|_{C^k([-1,1]^n;\mathbb{R})} \leq 1$. Additionally, for any fixed $\beta \in \{-N, ..., N\}^n$ we see

$$f\left(\frac{\beta}{N}\right) = y_\beta,$$

by choice of $\psi$. See also Figure 5 for an illustration of the function $f$.

By assumption, we can choose suitable coefficients $\rho_1, ..., \rho_m \in \mathbb{R}^n$, $\eta_1, ..., \eta_m \in \mathbb{R}$ and furthermore $\sigma_1, ..., \sigma_m \in \mathbb{R}$ such that

$$\|f - g\|_{L^\infty([-1,1]^n;\mathbb{R})} \leq \varepsilon$$

for

$$g := \sum_{j=1}^m \sigma_j \cdot \phi\left(\rho_j^T \cdot \bullet + \eta_j\right).$$

Letting

$$\tilde{g} := g(\bullet/N) = \sum_{j=1}^m \sigma_j \cdot \phi\left(\frac{\rho_j^T}{N} \cdot \bullet + \eta_j\right), \tag{F.7}$$

we see for every $\alpha \in \{-N, ..., N\}^n$ that

$$\tilde{g}(\alpha) = g\left(\frac{\alpha}{N}\right) \begin{cases} \geq y_\alpha - \varepsilon = c_1 N^{-k} - \varepsilon \overset{(F.6)}{>} (c_1/2) N^{-k}, & \text{if } z_\alpha = 1, \\ \leq y_\alpha + \varepsilon \overset{(F.6)}{<} (c_1/2) N^{-k}, & \text{if } z_\alpha = 0. \end{cases}$$

Therefore, we get $\mathbb{1}\left(\tilde{g} > (c_1/2)N^{-k}\right)(\alpha) = z_\alpha$ for any $\alpha \in \{-N, ..., N\}^n$. Since the choice of $z_\alpha$ has been arbitrary, it follows that the set

$$H := \left\{ \mathbb{1}\left(\tilde{g} > (c_1/2)N^{-k}\right)\big|_{\{-N,...,N\}^n} : \tilde{g} \text{ of form (F.7)} \right\}$$

shatters the whole set $\{-N, ..., N\}^n$. Therefore, we conclude that

$$\mathrm{VC}(H) \geq (2N + 1)^n \geq N^n.$$

On the other hand, [4, Theorem 8.11] shows that

$$\mathrm{VC}(H) \leq 2m(n + 2) \log_2(60n \cdot N) \leq c_3 \cdot m \cdot \ln(N)$$

with a suitably chosen constant $c_3 = c_3(n)$. Here we used that $N \geq 3$ so that $\ln(N) \geq \ln(3) > 0$. Combining those two inequalities yields

$$m \geq \frac{N^n}{c_3 \cdot \ln(N)}.$$

Using that $N \geq c_4 \cdot \varepsilon^{-1/k}$ with $c_4 := c_4(n, k) = \left(\frac{c_1}{2^{k+1}}\right)^{1/k}$ and $N \leq c_5 \cdot \varepsilon^{-1/k}$ with the definition $c_5 := c_5(n, k) = \left(\frac{c_1}{2}\right)^{1/k}$, we see that

$$m \geq \frac{c_4^n \cdot \varepsilon^{-n/k}}{c_3 \cdot \ln\left(c_5 \cdot \varepsilon^{-1/k}\right)} \geq c_6 \cdot \frac{\varepsilon^{-n/k}}{\ln(1/\varepsilon)}$$

with $c_6 = c_6(n, k) > 0$ chosen appropriately. $\qquad\square$

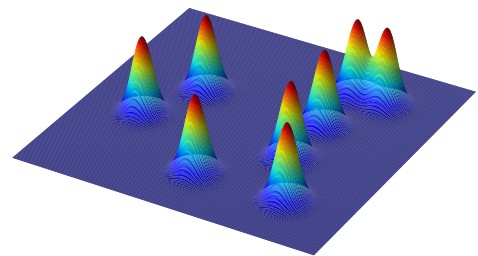

Figure 5: Illustration of the function $f$ considered in the proof of Theorem F.5.

As a corollary, we get a similar result for complex-valued neural networks.

**Corollary F.6.** *Let $n, k \in \mathbb{N}$ and*

$$\phi : \quad \mathbb{C} \to \mathbb{C}, \quad \phi(z) := \frac{1}{1 + e^{-\operatorname{Re}(z)}}.$$

*Then there exists a constant $c = c(n, k) > 0$ with the following property: If $\varepsilon \in (0, \frac{1}{2})$ and $m \in \mathbb{N}$ are such that for every function $f \in C^k (\Omega_n; \mathbb{C})$ with $\|f\|_{C^k(\Omega_n;\mathbb{C})} \le 1$ there exist coefficients $\rho_1, ..., \rho_m \in \mathbb{C}^n$, $\eta_1, ..., \eta_m \in \mathbb{C}$ and $\sigma_1, ..., \sigma_m \in \mathbb{C}$ satisfying*

$$\left\| f(z) - \sum_{j=1}^m \sigma_j \cdot \phi \left( \rho_j^T z + \eta_j \right) \right\|_{L^\infty(\Omega_n;\mathbb{C})} \le \varepsilon,$$

*then necessarily*

$$m \ge c \cdot \frac{\varepsilon^{-2n/k}}{\ln(1/\varepsilon)}.$$

*Proof.* We choose the constant $c = c(2n, k)$ according to the previous Theorem F.5 and let $\varphi_n$ as in (A.1). Then, let $\varepsilon \in (0, \frac{1}{2})$ and $m \in \mathbb{N}$ with the properties assumed in the statement of the corollary. If we then take an arbitrary function $f \in C^k \left( [-1, 1]^{2n}; \mathbb{R} \right)$ with $\|f\|_{C^k([-1,1]^{2n};\mathbb{R})} \le 1$, we deduce the existence of $\rho_1, ..., \rho_m \in \mathbb{C}^n$, $\eta_1, ..., \eta_m \in \mathbb{C}$ and $\sigma_1, ..., \sigma_m \in \mathbb{C}$, such that

$$\left\| f(x) - \operatorname{Re} \left( \sum_{j=1}^m \sigma_j \cdot \phi \left( \rho_j^T \cdot \varphi_n(x) + \eta_j \right) \right) \right\|_{L^\infty([-1,1]^{2n};\mathbb{R})}$$

$$\le \left\| (f \circ \varphi_n^{-1})(z) - \sum_{j=1}^m \sigma_j \cdot \phi \left( \rho_j^T z + \eta_j \right) \right\|_{L^\infty(\Omega_n;\mathbb{C})} \le \varepsilon.$$

In the next step, we show that

$$\mathbb{R}^{2n} \ni x \mapsto \operatorname{Re} \left( \sum_{j=1}^m \sigma_j \cdot \phi \left( \rho_j^T \cdot \varphi_n(x) + \eta_j \right) \right)$$

is a real-valued shallow neural network with $m$ neurons in the hidden layer and the real sigmoid function as activation function. Then the claim follows using Theorem F.5.

For every $j \in \{1, ..., m\}$ we pick a matrix $\tilde{\rho}_j \in \mathbb{R}^{2n \times 2}$ with the property that one has

$$\tilde{\rho}_j^{\,T} \cdot \varphi_n^{-1}(z) = \varphi_1^{-1} \left( \rho_j^T \cdot z \right)$$

for every $z \in \mathbb{C}^n$. This is possible, since this is equivalent to

$$\tilde{\rho}_j^{\,T} v = \varphi_1^{-1}(\rho_j^T \varphi_n(v))$$

for all $v \in \mathbb{R}^{2n}$, where the right-hand side is an $\mathbb{R}$-linear map $\mathbb{R}^{2n} \to \mathbb{R}^2$. Denoting the first column of $\tilde{\rho}_j$ by $\widehat{\rho}_j$, we get

$$\widehat{\rho}_j^{\,T} \cdot \varphi_n^{-1}(z) = \operatorname{Re}\left(\rho_j^T \cdot z\right) \quad \text{for all } z \in \mathbb{C}^n.$$

Writing $\gamma$ for the classical real-valued sigmoid function (i.e. $\gamma(x) = \frac{1}{1+e^{-x}}$), we deduce for arbitrary $x \in \mathbb{R}^{2n}$ that

$$\operatorname{Re}\left(\sum_{j=1}^m \sigma_j \cdot \phi\left(\rho_j^T \cdot \varphi_n(x) + \eta_j\right)\right) = \operatorname{Re}\left(\sum_{j=1}^m \sigma_j \cdot \gamma\left(\operatorname{Re}\left(\rho_j^T \cdot \varphi_n(x) + \eta_j\right)\right)\right)$$

$$= \operatorname{Re}\left(\sum_{j=1}^m \sigma_j \cdot \gamma\left(\widehat{\rho}_j^{\,T} x + \operatorname{Re}(\eta_j)\right)\right)$$

$$= \sum_{j=1}^m \operatorname{Re}(\sigma_j) \cdot \gamma\left(\widehat{\rho}_j^{\,T} x + \operatorname{Re}(\eta_j)\right),$$

where in the last step we used that $\gamma$ is real-valued. As noted above, this completes the proof. $\qquad\square$

Now, we can finally prove Theorem 4.3.

*Proof of Theorem 4.3.* Let $\alpha = \frac{2n}{k}$ and choose $c_2 = c_2(\alpha) = c_2(n,k) > 0$ such that the inequality $\ln(x) \le c_2 \cdot x^{\alpha/2}$ holds for all $x \ge 1$. Furthermore, let $c_1 = c_1(n,k) > 0$ as in Corollary F.6. By choosing $c = c(n,k) > 0$ sufficiently small , we can ensure that $\varepsilon_m := c \cdot (m \cdot \ln(m))^{-k/(2n)}$ satisfies

$$\ln\left(\frac{c_1}{c_2}\right) + \frac{\alpha}{2} \ln(1/\varepsilon_m) \ge \frac{\alpha}{4} \cdot \ln(1/\varepsilon_m) \quad \text{for all } m \in \mathbb{N}_{\ge 2}.$$

By further shrinking $c = c(n,k) > 0$ if necessary, we may assume

$$c \cdot (2 \cdot \ln(2))^{-k/(2n)} < \frac{1}{2}$$

and hence $c \cdot (m \cdot \ln(m))^{-k/(2n)} < \frac{1}{2}$ for all $m \in \mathbb{N}_{\ge 2}$. Finally, setting $c_3 := \frac{\alpha}{4}$ and shrinking $c$ even further, we can arrange that $c^\alpha < c_1 \cdot c_3$. Now, assume towards a contradiction that for every $f \in C^k(\Omega_n; \mathbb{C})$ with $\|f\|_{C^k(\Omega_n;\mathbb{C})} \le 1$ there are coefficients $\rho_1, ..., \rho_m \in \mathbb{C}^n, \sigma_1, ..., \sigma_m \in \mathbb{C}$ and $\eta_1, ..., \eta_m \in \mathbb{C}$ such that

$$\left\| f(z) - \sum_{j=1}^m \sigma_j \cdot \phi\left(\rho_j^T z + \eta_j\right) \right\|_{L^\infty(\Omega_n;\mathbb{C})} < c \cdot (m \cdot \ln(m))^{-k/(2n)}.$$

Applying Corollary F.6, we then get

$$m \ge c_1 \cdot \frac{\varepsilon^{-2n/k}}{\ln(1/\varepsilon)}$$

with $\varepsilon = c \cdot (m \cdot \ln m)^{-k/(2n)} \in (0, \frac{1}{2})$ and $c_1 = c_1(n,k) > 0$. Recall from the beginning of the proof that $\alpha := 2n/k$ and that $\ln(x) \le c_2 x^{\alpha/2}$ for every $x \ge 1$. We observe

$$m \ge c_1 \cdot \frac{\varepsilon^{-\alpha}}{\ln(1/\varepsilon)} \ge \frac{c_1}{c_2} \varepsilon^{-\alpha/2},$$

which implies

$$\ln(m) \ge \ln\left(\frac{c_1}{c_2}\right) + \frac{\alpha}{2} \cdot \ln(1/\varepsilon) \ge \frac{\alpha}{4} \cdot \ln(1/\varepsilon) = c_3 \cdot \ln(1/\varepsilon).$$

Overall we then get

$$m \ge c_1 \cdot \frac{\varepsilon^{-\alpha}}{\ln(1/\varepsilon)} = c_1 \cdot c^{-\alpha} \cdot \frac{m \cdot \ln(m)}{\ln(1/\varepsilon)} \ge c_1 \cdot c_3 \cdot c^{-\alpha} \cdot \frac{m \cdot \ln(m)}{\ln(m)} = c_1 \cdot c_3 \cdot c^{-\alpha} \cdot m.$$

Since we chose $c$ such that $c^\alpha < c_1 \cdot c_3$ we get the desired contradiction. $\qquad\square$

