# OpenReview forum: "Optimal approximation using complex-valued neural networks"
_NeurIPS.cc/2023/Conference — NeurIPS 2023 poster_

### Official Review · Reviewer_6LvE · 2023-07-06

**Soundness:** 3 good
**Presentation:** 4 excellent
**Contribution:** 3 good
**Rating:** 7
**Confidence:** 4

**Summary:**

The paper studies approximation rates of shallow complex-valued neural networks (CVNN) with general non-polyharmonic activation functions. First, the paper establishes upper bounds for the error of approximation of polynomial and general smooth functions by CVNNs. Then, various aspects of the optimality of these bounds are discussed. First, it is shown that these bounds are tight assuming continuous weight selection. Next, a special activation function is presented for which the bound can be improved. Finally, it is shown that for a standard sigmoid-type activation function the rate bound is tight.

**Strengths:**

**Contribution.** The paper seems to be the first paper that establishes optimal approximation rates for CVNNs with general non-polyharmonic activation functions. Previous results were either limited to specific activations or only established the universal approximation property, without convergence rates. Moreover, the paper comprehensively analyzes the optimality of its convergence rates. It proves their tightness assuming a continuous weight selection and shows that faster rates can be achieved if this assumption is dropped and a specially designed activation is used. It should be noted, however, that the CVNN model considered in the paper does not seem to be very important, and that most results of the paper are analogs of existing results for usual real-valued neural networks - see Weaknesses below.

**Quality and clarity.** The paper is very well written. The theorems are clearly stated, sketches of key ideas of the proofs are provided in the main text. Proof details are provided in a large appendix and also seem to be carefully written, though I did not study all of them closely.

**Weaknesses:**

**Questionable significance of the CVNN model.** The usual real-valued neural networks are important both mathematically and practically. They represent simple and natural non-linear models whose significance is well-established. In contrast, CVNNs do not seem to be significant from either perspective. Mathematically, they mix holomorphic linear operations with generic non-linear operations. This combination does not seem to have interesting analytic properties (except for the original observation by Voigtlaender that the condition of non-polynomiality for RVNN gets replaced by non-polyharmonicity for CVNN) or obvious practical or computational meaning. The paper does not explain why CVNNs are important, instead referring to a small number of papers from 2016-2018 where a similar structure was applied.

**Limited conceptual novelty.** The paper generally adapts existing methods and results from the real-valued to the complex-valued setting - admittedly with many extra new twists. The established convergence rate is the same as for the respective real network with doubled real dimension.

**High technicality for a conference.** This is quite a technical paper with 40 pages of proofs in the appendix. It is unlikely that any NeurIPS reviewer properly checks all these, so the paper might be more suitable for a journal with a more comprehensive review process. However, this is only a minor point, since the paper makes a good effort to present key ideas and sketches of proofs already in the main text.



**Questions:**

Why specifically is the particular computational structure used in CVNNs (a layer of holomorphic linear operations + a layer of pointwise non-holomorphic non-linear operations + another layer of holomorphic linear operations) important?

**Limitations:**

See above

---

> ### Author Rebuttal · Authors · 2023-08-08
>
> Thanks for your constructive feedback and in particular for the positive comments on the quality and clarity of the paper.
>
> In the following, we individually answer your comments:
>
> > The usual real-valued neural networks are important both mathematically and practically. They represent simple and natural non-linear models whose significance is well-established. In contrast, CVNNs do not seem to be significant from either perspective. Mathematically, they mix holomorphic linear operations with generic non-linear operations. This combination does not seem to have interesting analytic properties (except for the original observation by Voigtlaender that the condition of non-polynomiality for RVNN gets replaced by non-polyharmonicity for CVNN) or obvious practical or computational meaning. The paper does not explain why CVNNs are important, instead referring to a small number of papers from 2016-2018 where a similar structure was applied.
>
> > Why specifically is the particular computational structure used in CVNNs (a layer of holomorphic linear operations + a layer of pointwise non-holomorphic non-linear operations + another layer of holomorphic linear operations) important?
>
> We agree that compared to RVNNs CVNNs are less used. One reason for this is certainly the limited availability of wide-spread libraries for implementing and training CVNNs. Nevertheless, there has been sufficient interest in CVNNs such that the effort of implementing such a library has been invested (https://github.com/NEGU93/cvnn). We would also like to mention the following more recent papers that find CVNNs to be advantageous for tasks that naturally involve complex numbers. We will include them in the final version of our paper:
>
> - https://doi.org/10.1109/ICASSP39728.2021.9413814
> - https://doi.org/10.1007/s11265-022-01793-0
> - https://doi.org/10.1002/mrm.28733
> - https://doi.org/10.1002/nbm.4312
>
>
> The intuitive reasons for why CVNNs perform better than RVNNs in some application areas are the following: In applications where complex-valued inputs naturally occur, it is natural to use complex arithmetic in applied machine learning models as well. In particular, the use of phase-preserving complex activation functions such as the complex cardioid or the modReLU should be mentioned. Note that phase preservation cannot be achieved when non-trivial real activation functions are applied separately to real and imaginary parts of complex-valued neurons. Likewise, the holomorphic nature of the linearities (i.e., the linearities are linear over $\Bbb{C}$) is crucial for a faithful handling of the phase.
>
>
> As you have already pointed out, CVNNs exhibit quite interesting mathematical properties, in particular regarding the interplay of the properties of the activation function and the expressivity of the resulting CVNNs. The universal approximation theorem states that shallow CVNNs are universal if and only if the activation function is non-polyharmonic, but it is a priori not clear at all that the property of not being polyharmonic is enough to guarantee *optimal quantitative* approximation rates. Our paper shows that this is the case.
>
> > The paper generally adapts existing methods and results from the real-valued to the complex-valued setting - admittedly with many extra new twists. The established convergence rate is the same as for the respective real network with doubled real dimension.
>
> Our proof indeed required "many extra new twists" compared to the proof of the real-valued case.
> Indeed, there are several novel results and ideas in our paper. In particular noteworthy is Prop. 2.1 which connects the notion of non-polyharmonicity with the non-vanishing of Wirtinger derivatives at a single point. This is non-trivial and central to our proof. Also the other proofs are not simply a matter of "replacing $\Bbb{R}$ by $\Bbb{C}$" but rather require substantial changes to the proofs, mostly because real-valued activation functions are 1D objects, whereas complex activation functions are multi-dimensional (cf. the differences between ODEs and PDEs).
>
> The established convergence rate is indeed the same as for RVNNs with doubled real dimension. Note, however, that we prove this bound to be optimal (assuming a continuous weight selection) so that a better bound is not to be expected anyways.
>
> > High technicality for a conference. This is quite a technical paper with 40 pages of proofs in the appendix. It is unlikely that any NeurIPS reviewer properly checks all these, so the paper might be more suitable for a journal with a more comprehensive review process. However, this is only a minor point, since the paper makes a good effort to present key ideas and sketches of proofs already in the main text.
>
> Including such a long paper at NeurIPS is not completely unusual. See for instance https://papers.nips.cc/paper_files/paper/2021/hash/c82836ed448c41094025b4a872c5341e-Abstract.html
> We are happy that you appreciate our efforts to present the key ideas and sketches of proofs in the main text.

---

> > ### Comment · Reviewer_6LvE · 2023-08-12
> >
> >
> > Thank you for your answers. Your arguments look convincing. I'm increasing my score.
> >
> > I still have an impression, though, that your setting is a special case of a more general setting that would be more natural. The usual shallow real networks are constructed from layers that obey two kinds of constraints: linear layers obey the linearity constraint, while the pointwise nonlinear layers obey the constraint that the action is pointwise. My understanding is that your setting extends this setup in the following way. First, you group the variables in blocks of two (real/imaginary parts), and then you impose Cauchy-Riemann conditions on the linear layers. In the linear case the Cauchy-Riemann conditions reduce to a pair of linear algebraic conditions in each $2\times 2$ block of the respective matrix. You also replace pointwise nonlinearity by blockwise nonlinearity. Now, my understanding is that your results show that these modifications do not break the classical approximation rate. This suggests that none of the specific modifications that you do (blocking, Cauchy-Riemann, blockwise activation) actually has any effect on the approximation rate. So if you divided the variables into blocks of say three variables rather than two, and imposed some blockwise linear algebraic constraints other than Cauchy-Riemann, and used a suitably non-degenerate blockwise activation, would this also preserve the classical rate?

---

> > > ### Author Response · Authors · 2023-08-15
> > >
> > > Thanks for your positive feedback and for increasing the score! We greatly appreciate it.
> > >
> > > It is indeed an interesting question whether our results can be embedded into an even more general setting, where arbitrarily big groups of neurons are considered. And indeed, some older works (such as "Quaternionic Neural Networks for Associative Memories" by Teijiro Isokawa, Haruhiko Nishimura, and Nobuyuki Matsui) have for instance considered quaternion-valued neural networks or even so-called Clifford-algebra-valued neural networks (although we are not aware of significant real-world applications of these).
> > >
> > > However, in general going beyond the setting of complex numbers and quaternions, the considered structures get much less canonical, i.e., it is not quite clear which algebraic constraints one should impose in the case of, e.g., blocks of three neurons. This is mostly since it is not possible to endow $\mathbb{R}^3$ with an algebraic structure that represents a sensible multiplication (allowing for division). In fact, it is not possible to endow $\mathbb{R}^s$ with such a structure whenever $s \notin \{1,2,4\}$ (which is exactly the case of the usual real numbers, the complex numbers and quaternions). This is known as Frobenius' Theorem for real division algebras.
> > >
> > > Finally, the set of "good" blockwise activation functions would heavily depend on the chosen algebraic constraint. Thus, one should probably first settle the question of universality in these general settings, and then study the question of approximation rates.
> > >
> > > We would expect that the reachable approximation rate (under suitable assumptions such as continuity) again agrees with the "expected" rate, but this is not clear at all and would have to be carefully investigated.

---

### Official Review · Reviewer_ve6Y · 2023-07-07

**Soundness:** 4 excellent
**Presentation:** 3 good
**Contribution:** 3 good
**Rating:** 7
**Confidence:** 3

**Summary:**

The paper studies approximation error bounds for complex-valued neural networks based. The authors relied on several techniques from the work by Mhaskar (1996) and proved several theorems in the paper. I will summarize two important results here.
1. Given a function from $C^k$, an optimal error bound is proved under some conditions on the activation function and the hypothesis of continuous weight selection. Furthermore, the complex-valued neural network achieving this bound has a universal first layer.
2. When the hypothesis of continuous weight selection is dropped, the authors proved that there exists an activation function allowing the complex-valued neural network to achieve a better bound. No optimality is claimed under this case. On the other hand, there is an activation function such that the bound cannot be improved up to logarithmic factors.

**Strengths:**

The novelty of this paper is clear. The paper provides error bounds for complex-valued neural networks using more general activation functions and shows that the optimal bound can be achieved under the hypothesis of continuous weight selection. The results in this paper generalize the results in the previous work [9]. The problem is well-motivated, and the theorems proved are sound. Overall, the paper is well-written, and I enjoy reading the paper.

**Weaknesses:**

The hypothesis of continuous weight selection is not a practical assumption. The optimal choice of the network weights is discontinuous in general (see the reference below). It would be better if the authors could make this clear in the paper.

Kainen, Paul C., Věra Kůrková, and Andrew Vogt. "Approximation by neural networks is not continuous." Neurocomputing 29, no. 1-3 (1999): 47-56.

Under this assumption, the error bound can be proved to be optimal. However, no optimal results are provided if this assumption is dropped. Regarding this, it would be more convincing for the paper if the authors could provide insights into difficulties that arose in proving the statements and potential workarounds.

The results provided are similar to [9] so it would be more interesting if the authors could describe what difficulties they have overcome to prove these results. Given the high similarity of the results, I would expect more discussions in the related work section or in the descriptions of the proof sketch.

**Questions:**

Line 28: Would it be more precise if referencing the work by Cybenko in 1989?

Line 140: What is your definition of a smooth function? Please make it precise.

From Theorem 3.1, the complexity of an approximating complex-value network is established for a polynomial. Would it be possible to use the Stone-Weierstrass theorem to establish an approximation bound? This seems to be a natural step to apply.

**Limitations:**

The authors have sufficiently addressed the limitations of the paper in Section 5. In my view, the greatest limitation of this work is the use of continuous weight selection for guaranteeing the optimality of the error bound. It would be more convincing if the authors can give some insights into this in Section 5.

---

> ### Author Rebuttal · Authors · 2023-08-04
>
> We are glad that you enjoyed reading our paper!
> In the following we address each of the points that you raised.
>
> # Point 1
> > The hypothesis of continuous weight selection is not a practical assumption. The optimal choice of the network weights is discontinuous in general (see the reference below). It would be better if the authors could make this clear in the paper.
>
> Thanks for suggesting that "Approximation by neural networks is not continuous" might be a relevant reference for our paper. We will include it in the final version and point out that selecting the **best approximating NN** of a given size for a given function is discontinuous.
>
> In addition:
>
> 1. The continuity assumption is satisfied in many practical scenarios. In practice NNs are trained by variants of SGD. Here the training algorithm does not have direct access to the (unknown) ground-truth function $f$, but can access $f$ only through the training samples $(x_i,y_i)$, where $y_i =f(x_i)$ or $y_i =f(x_i)+\mathrm{noise}$. One can show that each gradient update depends continuously on the training samples and therefore also on the ground-truth function $f$. Strictly speaking, this continuity is only guaranteed to hold if the activation function is continuously differentiable. Therefore, for any continuously differentiable activation function the continuity assumption is satisfied in practice.
>
> 2. The paper mentioned above shows that selecting **the best** approximating NN for each function $f$ is a discontinuous operation. This, however, is not the setting that we consider. We show that for $\varepsilon=c\cdot m^{-k/(2n)}$ each function in the $C^k$-unit ball can be approximated using a shallow CVNN with $m$ neurons up to error $\varepsilon$, but it could be that several functions in the $C^k$ unit ball can in fact be approximated much better. Thus, the mentioned paper does not fully apply to the setting we consider.
>
>  # Point 2
> > (...) it would be more convincing for the paper if the authors could provide insights into difficulties that arose in proving the statements and potential workarounds.
>
> The following are the main difficulties:
>
> 1. Our Theorems 4.2 and 4.3 show that, if one drops the continuity assumption, there is no single approximation rate that is the optimal rate jointly for all smooth activation functions. This means that one would need to perform a separate analysis for different (classes of) activation functions.
>
> 2. For deep NNs (with more than two hidden layers) with general smooth activation function it is not possible to derive any non-trivial lower bounds without assuming continuity, since there exists an activation function with the property that NNs of constant size using this activation function can approximate any continuous function to arbitrary precision (see [22, Theorem 4]). Note that [22] considers real-valued NNs, but the results can be transferred to CVNNs with a suitable choice of the activation function. Hence, a lower bound in the case of unrestricted weight selection can, if at all, only be derived for shallow NNs.
>
> 3. In the real-valued case, fully general lower bounds for the approximation capabilities of shallow NNs have been derived by using results from [15] regarding the approximation properties of so-called ridge functions, i.e., functions of the form $\sum_{j=1}^m \phi_j(\langle a_j, x\rangle)$ with $a_j \in \Bbb{R}^d$ and $\phi_j: \Bbb{R} \to \Bbb{R}$. We are interested in generalizing these results to higher-dimensional ridge functions of the form $\sum_{j=1}^m \phi_j (A_j x)$, where each $\phi_j : \Bbb{R}^s \to \Bbb{R}$ and $A_j \in \Bbb{R}^{s \times d}$. This would imply lower bounds for shallow CVNNs. However, such a generalization seems to be highly non-trivial and is outside the scope of our paper.
>
> We will include a brief discussion of these points in the final manuscript.
>
> # Point 3
> > The results provided are similar to [9] so it would be more interesting if the authors could describe what difficulties they have overcome to prove these results. Given the high similarity of the results, I would expect more discussions in the related work section or in the descriptions of the proof sketch.
>
> 1. As outlined in Section 1.1.2 our results differ *significantly* from [9]: we consider general activation functions, we show that the rate proved in [9] for deep NNs can in fact already be achieved using shallow NNs, and can be improved by a log factor.
>
> 2. Our proof techniques differ significantly from those in [9]. While in [9] the target function is approximated by many polynomials of low degree, combined with a partition of unity, we approximate the target function by a single polynomial of high degree. For approximating the polynomial using a NN, [9] uses a construction specific to ReLU which was pioneered by Yarotsky [34]. In contrast, we use Wirtinger derivatives and divided differences which can be done for quite general activation functions.
>
> We will include a discussion of the proof techniques in the final manuscript.
>
> # Questions
>
> > Line 28: Would it be more precise if referencing the work by Cybenko in 1989?
>
> We will cite the work by Cybenko in the final version. The reason why we cite [19] is that it considers very general activation functions.
>
> > Line 140: What is your definition of a smooth function? Please make it precise.
>
> Thanks for noting that the definition is missing. By "smooth" we mean $C^\infty$-functions in the sense of real variables. We will include that in the final version of the paper.
>
> > From Theorem 3.1, the complexity of an approximating CVNN is established for a polynomial. Would it be possible to use the Stone-Weierstrass theorem to establish an approximation bound? This seems to be a natural step to apply.
>
> Using the Stone-Weierstrass theorem, one could establish qualitative results. However, this theorem is not quantitative. Theorem J.15 can be seen as a quantitative version of the Stone-Weierstrass Theorem for approximating $C^k$-functions.

---

> > ### Comment · Reviewer_ve6Y · 2023-08-21
> >
> > I would like to thank the authors for their detailed responses. My concerns are fully addressed, and in light of this, I have increased my rating by 1. I hope the authors keep their promises and deliver extra analyses and clarifications in their final version. One of the concerns raised by Reviewer GqHo is about the constant $c(n,k)$. I think this is not that problematic given the independence of $\epsilon$. However, I believe it is helpful for the reader to know the upper bound of the constant and know why the constant is huge and perhaps some potential approaches to improve it. I also agree with the authors that the curse of dimensionality cannot be avoided without making strong assumptions about the function family.

---

### Official Review · Reviewer_yApE · 2023-07-07

**Soundness:** 3 good
**Presentation:** 3 good
**Contribution:** 3 good
**Rating:** 5
**Confidence:** 3

**Summary:**

This paper studies the approximation power of complex-valued neural networks (CVNNs). They derive that the approximation error is with the order $m^{-k/2n}$, where m is the number of neurons, k is the smoothness of the target function, and n is the input dimension of the neural network. They also show that this approximation error is optimal under some mild assumptions.

**Strengths:**

1. This paper derives the approximation bounds of CVNNs for any continuous activation functions.
2. Furthermore, they show that the approximation bounds they obtain are optimal with a natural continuity assumption.
3. For several specific activation functions, the authors derive the upper bounds without the assumption of the continuity of weight selection.

**Weaknesses:**

1. For general activation functions, the authors need to assume the continuity of weight selection to derive the optimal approximation rates. It will be more convincing if they can show results for general activation functions without the assumption of the continuity of weight selection.

**Questions:**

Please see the weaknesses.

**Limitations:**

No.

---

> ### Author Rebuttal · Authors · 2023-08-04
>
> Thanks for your constructive feedback for our paper.
>
> Your main criticism of our paper is that
>
> > It will be more convincing if they can show results for general activation functions without the assumption of the continuity of weight selection.
>
> We agree that it would be interesting to establish lower bounds also in the case of discontinuous weight selection, as we also mention in the Limitations section of our paper. However, in full generality this is a highly non-trivial problem a full resolution of which is outside the scope of this paper. In particular we would like to mention the following points:
>
> 1. The continuity assumption is natural: In practice neural networks are trained by variants of stochastic gradient descent. Here the training algorithm does not have direct access to the (unknown) ground-truth function $f$, but can access $f$ only through the training samples $(x_i, y_i)$, where $y_i = f(x_i)$ or $y_i = f(x_i) + \mathrm{noise}$. One can show that each gradient update depends continuously on the training samples and therefore also on the ground-truth function $f$. Strictly speaking, this continuity is only guaranteed to hold if the activation function is continuously differentiable. Therefore, for any continuously differentiable activation function the continuity assumption is satisfied in practice.
>
> 2. For *deep* neural networks (with more than two hidden layers) with general smooth activation function it is *not* possible to derive any non-trivial lower bounds without assuming continuity, since there exists an activation function with the property that neural networks of constant size using this activation function can approximate any continuous function to *arbitrary* precision (see [22, Theorem 4]). Note that [22] considers the real-valued setting but the results can be transferred to complex-valued networks with a suitable choice of the activation function. Hence, a lower bound in the case of unrestricted weight selection can, if at all, only be derived in the case of shallow networks.
>
> 3. In the real-valued case, fully general lower bounds for the approximation capabilities of shallow neural networks have been derived by appealing to the results from [15] regarding the approximation properties of so-called ridge functions. Here, a ridge function is of the form $\sum_{j=1}^m \phi_j (\langle a_j, x\rangle)$ with $a_j \in \mathbb{R}^d$ and each $\phi_j: \mathbb{R} \to \mathbb{R}$. We are interested in generalizing these results to higher-dimensional ridge functions, i.e., functions of the form $\sum_{j=1}^m \phi_j (A_j x)$, where each $\phi_j : \mathbb{R}^s \to \mathbb{R}$ and $A_j \in \mathbb{R}^{s \times d}$. This would imply lower bounds for shallow complex-valued neural networks. However, such a generalization seems to be highly non-trivial and is outside the scope of our paper.
>
> 4. We already discuss the issue of possibly discontinuous weight selection to some depth in Theorems 4.2 and 4.3. Theorem 4.2 shows that there is a smooth activation function for which one can even achieve the rate $\mathcal{O}(m^{-k/(2n-1)})$ which shows that without the continuity assumption and for this special activation function the upper bound of $\mathcal{O}(m^{-k/(2n)})$ is not sharp. In contrast, Theorem 4.3 shows that for some special activation function the upper bound of $\mathcal{O}(m^{-k/(2n)})$ is actually sharp (up to logarithmic factors), even in the case of possibly discontinuous weight selection.
>
>      In combination, these theorems show that in the case of unrestricted weight selection one cannot derive a sharp approximation rate holding simultaneously for all smooth activation functions. Specifically, Theorem 4.2 shows that such a bound would have to be smaller than $m^{-k/(2n-1)}$ while Theorem 4.3 shows that it would have to be greater than $(m \cdot \ln (m))^{-k/(2n)}$. This means that for the case of discontinuous weight assignment one has to perform an individual analysis for different activation functions.

---

### Official Review · Reviewer_GqHo · 2023-07-10

**Soundness:** 3 good
**Presentation:** 2 fair
**Contribution:** 2 fair
**Rating:** 3
**Confidence:** 4

**Summary:**

The paper studies the expressive power of complex-valued neural networks. It is shown that depth 2 networks with non-polyharmonic (the complex equivalent of non-polynomial) activations can approximate any continuous function on a compact domain to arbitrary accuracy which decays polynomially with the width and exponentially with the smoothness of the target function, but nevertheless suffers from the curse of dimensionality. This upper bound is also shown to be optimal in general, and that furthermore, there are more tailored examples where one can slightly improve the approximation rate.


Post-rebuttal:

I still find the technical contribution of the paper rather fair (as my original score indicates). My main concern with the paper is that its impact feels somewhat limited as it provides a guarantee which suffers from the curse of dimensionality. While I understand that this cannot be evaded in general, obtaining a result which is tight in the worst case where the worst case is intractable is a clear limitation. While I appreciate the fact that there are other older results in the literature which are highly cited and obtain such results for the real setting, this paper's merits should be considered compared to what is already known, and from this perspective, obtaining results in the complex setting that are analogous to the real setting is of interest but not groundbreaking. I have therefore decided to keep my original score.

**Strengths:**

- The problem feels overall well-motivated, and CVNNs seem like an interesting model to study.

- The analogies and differences between real and complex neural networks highlighted by the results in this paper are interesting.

**Weaknesses:**

- On a technical perspective, it seems like the main technical contributions here are to adapt existing technique to the CVNN setting. Most results seem to take existing proofs and just apply them to the complex setting, which doesn't feel novel enough.

- The exact dependence on the rate of approximation is unclear. There's a `constant' $c(n,k)$ which hides some dependence on the input dimension and smoothness parameters, which could be potentially huge (see questions below).

- The separation established in Section 4 by Theorem 4.2 and Theorem 4.3 is not particularly strong, and is only significant in cases where the input dimension $n$ is very small.

- I didn't find the main approximation result in Theorem 3.2 very surprising. This is an analogous result for something which is already known for real-valued networks, so obtaining this result for complex-valued functions feels incremental. In particular, it would be of greater interest, in my opinion, to study cases where complex-valued networks attain an \emph{efficient} approximation in the input dimension $n$, since these have stronger practical implications.

**Questions:**

Questions:

- "complex-valued networks behave significantly different from real-valued networks": I don't see how this is `significantly different'. The second property does indicate a difference, but the first characteristic is analogous (non-polynomial and non-polyhedral activations).

- There's no clear motivation for the very technical choice of the $C^k$-norm in the paper. Can you provide a more intuitive explanation for it and why it is chosen? In particular, why is this choice not stylized for obtaining the main result?

- "It is crucial that the size of the networks considered in Theorem 3.1 is independent of the approximation quality $\varepsilon$": Why is this important?

- What is the quantity $c(n,k)$ which appears in many of the theorem? The bounds in the paper are interesting since they allow us to study the dependence of the accuracy attained as a function of, say, $n$ for some fixed $k$; but the dependence of $c(n,k)$ on $n$ isn't clear. I tried to better understand it by looking at the proofs in the appendix but it's not made explicit there as well. $c(n,k)$ could, potentially, have magnitude $m^{k/(2n)}$ which would render the upper bound vacuous. Why is this not the case? Is this the same quantity in both the upper bound in Theorem 3.2 and the lower bound in Theorem 4.3? If not, it could make the either bound very loose despite appearing to provide a tight result.


Comments:

- Abstract: "the real-valued case is supported by a firm mathematical foundation" -- I wouldn't say our mathematical understanding of deep learning is firm.

- The abstract doesn't explicitly state what are the functions the main result applies to.

- Line 27: The universal approximation theorem dates back to 1989. See "Approximation by Superpositions of a Sigmoidal Function" by G. Cybenko

- "CVNNs have the same excellent approximation properties as real-valued networks.": Why is this conclusion reached? The results in the paper hold for a very broad class of smooth function and therefore suffer from the curse of dimensionality in the worst case.

**Limitations:**

- I think that the lack of clarity in the theorem statements regarding $c(n,k)$ is problematic, yet this is not discussed in the paper. I urge the authors to clarify this and discuss this more clearly.

- The limitations of the provided results which suffer from the curse of dimensionality should be clearly discussed.

---

> ### Author Rebuttal · Authors · 2023-08-04
>
> Thanks for your constructive feedback. We agree that CVNNs are an interesting model to study.
>
> # Weaknesses
>
> > Most results seem to take existing proofs and just apply them to the complex setting.
>
> We do not think that this is a fair assessment of our work. There are several novel results and ideas in our paper. In particular noteworthy is Prop. 2.1 which connects the notion of non-polyharmonicity with the non-vanishing of Wirtinger derivatives at a single point. This is non-trivial and central to our proof. Also the other proofs are not simply a matter of "replacing $\Bbb{R}$ by $\Bbb{C}$" but rather require substantial changes to the proofs, mostly because real-valued activation functions are 1D objects, whereas complex activation functions are multi-dimensional (cf. the differences between ODEs and PDEs).
>
> > I didn't find the main approximation result in Theorem 3.2 very surprising. [...]
>
> We don't think this result is "boring": It is highly non-trivial (and maybe surprising) that the correct complex generalization of "non-polynomial" (for our problem) is "non-polyharmonic". [33] shows that this is the correct generalization for *universality* of **shallow** NNs, but already in the case of NNs with two hidden layers this stops being true. Therefore, it is a priori not clear that the notion of "non-polyharmonic activation function" is the correct one for deriving *quantitative* approximation bounds such as studied in our paper.
> > it would be of greater interest [...] to study cases where CVNNs attain an *efficient* approximation in the input dimension $n$
>
> In this paper, we study $C^k$-functions, for which the curse of dimensionality is inevitable. We agree that studying other function classes (such as generalizations of the Barron class to CVNNs) is interesting, and plan to do this in the future.
> > The exact dependence on the rate of approximation is unclear. [...]
>
> See below.
> > The separation established in Section 4 by Theorems 4.2 and 4.3 is not particularly strong [...]
>
> The purpose of these two theorems is to show that it is impossible to derive upper and lower bounds with a *matching* approximation rate for all smooth activation functions and a possibly discontinuous weight assignment. We just care that the two rates are **distinct**, not that there is a strong separation.
>
> # Questions
>
> > "CVNNs behave significantly different from RVNNs": I don't see how this is `significantly different'. [...]
>
> The difference that we emphasize here is that, to obtain universality, there are more admissible activation functions for *deep* NNs than for *shallow* NNs in the complex case, which is not the case for RVNNs.
>
> > There's no clear motivation for the very technical choice of the $C^k$-norm in the paper.
>
> The $C^k$-space and the norm that we use are standard and widely used in the literature. Maybe you refer to the definition below l.116, where we wrote "$\partial^{\mathbf{k}}$" instead of  "$\partial^{\mathbf{k}} f$". We will fix this in the final version.
>
> > "It is crucial that the size of the NNs considered in Theorem 3.1 is independent of the approximation quality ": Why is this important?
>
> It is important to obtain the final approximation result regarding the equation in l.184-185. Moreover, it is noteworthy that for polynomials the network size is independent of the error, which is not the case for our other results.
>
> > Abstract: [...] I wouldn't say our mathematical understanding of deep learning is firm.
>
> We agree this formulation should be changed. We propose "is supported by a growing mathematical foundation".
>
> > The abstract doesn't explicitly state what functions the main result applies to.
>
> Thanks for noting that this is not entirely clear. In the final version we will make this more explicit.
>
> > Line 27: The universal approximation theorem dates back to 1989. [...]
>
> We agree that the work by Cybenko should be cited and we will do so in the final version. We cited [19] since it is the first work to consider *arbitrary, non-polynomial* activations.
>
> > "CVNNs have the same excellent approximation properties as RVNNs.": Why is this conclusion reached? The results in the paper [...]  suffer from the curse of dimensionality [...]
>
> We mean "excellent" in the sense that CVNNs attain the same approximation rate as RVNNs for $C^k$-functions. This is the *optimal* rate that can be achieved with $m$ (continuously selected) parameters (Theorem 4.1). You are correct in noting that even though the results are optimal they suffer from the curse of dimensionality (COD). We plan to study alternative function classes for which CVNNs do not suffer from the COD in future work.
>
> > What is the quantity $c(n,k)$ which appears in many of the theorem? [...] $c(n,k)$ could, potentially, have magnitude $m^{k/(2n)}$ which would render the upper bound vacuous. [...] Is this the same quantity in both Theorem 3.2 and Theorem 4.3? [...]
>
> The constant $c(n,k)$ only depends on $n$ and $k$ and is **independent** of the number of neurons $m$. Therefore, it cannot grow like $m^{k/(2n)}$. Roughly, $n$ and $k$ are taken as fixed and we ask how quickly the approximation error decays (asymptotically) as $m\to\infty$.
>
> We don't compute $c(n,k)$ explicitly but note that the dependence is indeed exponential in $n$, even for large $k$ (e.g., $k = n$). However, this is not a shortcoming of our proof but a fact of life, as follows from "Approximation of infinitely differentiable multivariate functions is intractable" by Novak and Woźniakowski, Journal of Complexity (2009).
>
> The constants that appear in theorems 3.2 and 4.3 are not the same, but this is standard in the literature for similar results. Precisely studying the constant $c(n,k)$ is extremely difficult and has only been done in a few very special cases (see the paper by Novak et al. above).
>
> > The limitations of the provided results which suffer from the curse of dimensionality should be clearly discussed.
>
> The final version will discuss this in greater detail.

---

> > ### Comment · Reviewer_GqHo · 2023-08-18
> > **Post-rebuttal response**
> >
> > Dear authors,
> >
> > Thank you for your detailed response which addresses the questions I raised. I appreciate your intensions to incorporate some of my suggestions into the paper.
> >
> > I want to point out that a setting where $n$ is taken as fixed is limiting. While such results are mainly of interest when $n$ is moderate, the resulting approximations quickly become inefficient as $n$ grows. I understand that this is the best that can be done in the worst case, but this ignores potential "average case" results and might be very loose in many instances. Such a limitation should be clearly discussed in the paper. Moreover, even though $c(n,k)$ is independent of $m$, in cases where it behaves as $k^n$ for example still make the upper bound weak.

---

> > > ### Author Response · Authors · 2023-08-19
> > >
> > > Thanks for your reply. We understand that you are dissatisfied with the dependence of our approximation bounds on the input dimension, which makes them subject to the curse of dimensionality, and we agree that it is important to study alternative function classes for which neural networks can avoid the curse. We strongly disagree, however, with the conclusion that this makes our results uninteresting, or not worth publishing.
> > >
> > >
> > > In the following, we try one final time to make our point.
> > >
> > >
> > > 1. As we already pointed out in an earlier response, for the $C^k$ function class that we consider, the curse of dimensionality is an inescapable fact of life and cannot be avoided. This holds in a worst-case sense, but quite likely also in an "average case" sense. Indeed, the paper "Phase Transitions in Rate Distortion Theory and Deep Learning" by Grohs, Klotz, and Voigtlaender shows this optimality in an average sense in a slightly, but closely related setting. We are strongly convinced that this result also extends to our setting. Verifying this, however, is outside the scope of this paper.
> > >
> > >
> > > 2. In the real-valued setting, there are several highly influential (well published and highly cited) works that are subject to the same limitations as our result. As selected examples we mention the following:
> > >
> > >     - "Neural networks for optimal approximation of smooth and analytic functions" by Mhaskar
> > >     - "Error bounds for approximations with deep ReLU networks" by Yarotsky
> > >     - "Optimal approximation of piecewise smooth functions using deep ReLU neural networks" by Petersen and Voigtlaender
> > >     - "Optimal approximation of continuous functions by very deep ReLU networks" by Yarotsky
> > >     - "The phase diagram of approximation rates for deep neural networks" by Yarotsky and Zhevnerchuk
> > >     - "Deep network approximation characterized by number of neurons" by Shen, Yang, and Zhang.
> > >
> > >     This underlines that such results are of high interest in the community.
> > >
> > >
> > > 3. Our results are not strictly limited to the class of $C^k$ functions. As an important auxiliary result (which might be of independent interest), we show that CVNNs can well approximate algebraic polynomials. There are natural and widely studied classes of functions that can be very well approximated by polynomials; for instance this holds for certain classes of holomorphic functions; see e.g. Example 2 in the paper "Approximation of smooth functionals using deep ReLU networks" by Song, Liu, Fan, and Zhou. For instance, for this class, our results on the approximation of polynomials using CVNNs would imply that using CVNNs with $N$ neurons, one can obtain an approximation error bound of $C \cdot \rho^{- N^{1/(2n)} / 5}$, where $C > 0$ and $\rho > 1$ only depend on the size of the polyellipse on which the considered functions are holomorphic, but not on the dimension. This bound is still subject to the curse of dimensionality, but much less than our bound for $C^k$ functions. This shows that our results and proof techniques can be useful to tackle the question of alternative function classes for which the curse can be avoided. We emphasize that there are many other function classes that can be well approximated by polynomials; for these, our results will thus be helpful.
> > >
> > >     We will be happy to add a brief discussion of these points to the final version of the paper.

---

### Decision · Program_Chairs · 2023-09-21

**Decision:**

Accept (poster)

**Comment:**

The targeted problem --- function approximation of complex valued neural networks --- possesses its own value and is largely overlooked in literature, given fruitful results about universal approximation using real valued neural networks. The obtained approximation rates and implications are clearly discussed, and the technical proofs are sound.

The paper has some weaknesses, and the major one is a lack of new analytical framework. As pointed out, the proof is largely adapted from existing works with modifications to the complex setting. Yet these twists allowing an extension of approximation theory to complex valued neural networks are still fair contributions.

There is an issue about the fixed input dimension and the approximation rate suffers from the curse of dimensionality. In the context of universal approximation, the input dimension is usually taken as fixed, and the obtained rate suffers from the curse of dimensionality without further assumptions. That being said, the approximation rate obtained in the paper is in par with older ones, confirming the technical soundness. However, it should be noted (Reviewer GqHo, 6LvE) that translating results from the real to the complex setting is of interest but not groundbreaking. Therefore, the recommendation is acceptance (poster).